# RUNTIME LEARNING MACHINE

## ABSTRACT

This paper proposes the **Runtime Learning Machine** for safety-critical autonomous systems. The learning machine has three interactive components: a high-performance (HP)-Student, a high-assurance (HA)-Teacher, and a Coordinator. The HP-Student is a high-performance but not fully verified Phy-DRL (physics-regulated deep reinforcement learning) agent that performs safe runtime learning in real plants, using real-time sensor data from real-time physical environments. On the other hand, HA-Teacher is a verified but simplified design, focusing on safety-critical functions. As a complementary, HA-Teacher's novelty lies in real-time patch for two missions: i) correcting unsafe learning of HP-Student, and ii) backing up safety. The Coordinator manages the interaction between HP-Student and HA-Teacher. Powered by the three interactive components, the runtime learning machine notably features i) assuring lifetime safety (i.e., safety guarantee in any runtime learning stage), ii) tolerating unknown unknowns, iii) addressing Sim2Real gap, and iv) automatic hierarchy learning (i.e., safety-first learning, and then high-performance learning). Experiments involving a cart-pole system, two quadruped robots, and a 2D quadrotor, as well as comparisons with state-of-the-art safe DRL, fault-tolerant DRL, and approaches for addressing Sim2Real gap, demonstrate the machine's effectiveness and unique features.

## 1 INTRODUCTION

Deep reinforcement learning (DRL) has been incorporated into numerous autonomous systems and has shown significant advancements in making sequential and complex decisions in various fields, such as autonomous driving Kendall et al. (2019); Kiran et al. (2021), chemical processes Savage et al. (2021); He et al. (2021), and robot locomotion Ibarz et al. (2021); Levine et al. (2016). These DRL-enabled systems have the potential to revolutionize many processes across different industries, leading to tangible economic impacts Tolentino (2019). However, the public-facing AI Incident database in AID has revealed that machine learning (ML) techniques, including DRL, can achieve remarkable performance without ensuring safety Zachary & Helen (2021). For instance, a report by the National Highway Traffic Safety Administration highlighted 351 car crashes related to advanced driver assistance systems from July 2023 to March 2024 in the US alone NHTSA. Therefore, ensuring high-performance DRL with verifiable safety is even more crucial today, aligning well with the market's demand for safe ML techniques.

### 1.1 SAFETY CHALLENGES AND OPEN PROBLEMS

Our considered safety challenges are rooted in the unknown unknowns and the Sim2Real gap.

**Challenge 1: Unknown Unknowns.** The unknown unknowns generally refer to outcomes, events, circumstances, or consequences that are not known in advance and cannot be predicted in time and distributions Bartz-Beielstein (2019). The dynamics of many safety-critical autonomous systems (e.g., autonomous vehicles Rajamani (2011), airplanes Roskam (1995), and quadrupedal robots Bledt et al. (2018)) are governed by a combination of known knowns (e.g., Newton's laws of motion), known unknowns (e.g., Gaussian noise without knowing to mean and variance), and unknown unknowns. The unknown unknowns are due to, for example, unforeseen operating environments and DNN's colossal parameter space, intractable activation, and hard-to-verify. The safety assurance also requires resilience to unknown unknowns, which is very challenging. The reasons stem from characteristics of unknown unknowns: there is almost zero historical data, unpredictable timing and distributions, resulting in the unavailability of models for scientific discoveries and understanding.

**Challenge 2: Sim2Real Gap.** The prevalent DRL involves training a policy within a simulator using synthetic data and deploying it to physical platforms. However, the difference between the simulated environment and the real world creates a gap known as the Sim2Real gap. This gap causes a drop in performance when using pre-trained DRL in real physical environments. Numerous approaches have been developed to address the Sim2Real gap Peng et al. (2018); Nagabandi et al. (2019); Tan et al. (2018); Yu et al. (2017); Cao et al. (2022); Imai et al. (2022); Du et al. (2021); Vuong et al. (2019); Yang et al. (2022a). These methods aim to improve the realism of the simulator and can mitigate the Sim2Real and domain gaps to varying degrees. Nevertheless, undisclosed gaps and missing dynamics continue to hinder the safety assurance of real plants.

To address Challenges 1 and 2, the most appealing solution is provided in Prospect 1.1 below.

**Prospect 1.1.** *Runtime learning for a high-performance action policy in **real** plants – using **real**-time sensor data generated from **real**-time physical environments while **prioritizing safety**.*

However, two open problems arise about bringing Prospect 1.1 into reality.

**Problem 1.2.** *If the DRL agent's actions lead to a safety violation, how can we correct his unsafe learning and back up the safety of real plants in a timely manner?*

**Problem 1.3.** *How to tolerate and also teach the DRL agent to tolerate unknown unknowns and Sim2Real gap for assuring safety of real plants?*

## 1.2 RELATED WORK

Significant efforts have been devoted to DRL safety by developing safe DRL and fault-tolerant DRL.

**Safe DRL.** One research focus of safe DRL is the safety-embedded reward, as a DRL agent must learn a high-performance action policy with verifiable safety. The control Lyapunov function (CLF) proposed in Perkins & Barto (2002); Berkenkamp et al. (2017); Chang & Gao (2021); Zhao et al. (2023) is a candidate. Meanwhile, seminal work in Westenbroek et al. (2022) revealed that a CLF-like reward could enable DRL with verifiable stability. At the same time, enabling verifiable safety is achievable by extending CLF-like rewards with given safety regulations. However, systematic guidance for constructing such CLF-like rewards remains open. The residual action policy is another shift in safe DRL, which integrates data-driven action policy and physics-model-based action policy. The existing residual diagrams focus on stability guarantee Rana et al.; Li et al.; Cheng et al. (2019b); Johannink et al. (2019), with the exception being Cheng et al. (2019a) on safety guarantee. However, the physics models considered are nonlinear and intractable, which thwarts delivering a verifiable safety guarantee or assurance, if not impossible. The recently developed Phy-DRL (physics-regulated DRL) framework Cao et al. (2024; 2023) can satisfactorily address the open problems of safe DRL. Summarily, Phy-DRL permits simplifying the model of nonlinear dynamics to an analyzable and tractable linear one. This linear model can then be a model-based guide for constructing the safety-embedded (CLF-like) reward and residual action policy. Meanwhile, the Phy-DRL exhibits verifiable safety. However, it is only mathematically or theoretically possible due to the underlying assumptions of manageable Sim2Real gap and unknown unknowns. In other words, Phy-DRL cannot offer verifiable safety for real plants in the face of unknown unknowns and the Sim2Real gap.

**Fault-tolerant DRL.** Fault-tolerant DRL is another direction for DRL safety in real plants. Recent approaches include neural Simplex Phan et al. (2020), runtime assurance Brat & Pai (2023); Sifakis & Harel (2023); Chen et al. (2022), and model predictive shielding Bastani (2021); Banerjee et al. (2024). They treat the DRL agent as a high-performance module (HPM) but a black box that runs in parallel with a verified high-assurance module (HAM). Normally, HPM controls the real plants. HAM takes over once safety violation occurs. These architectures can ensure the safe running of DRL in real plants under the assumption that Challenges 1 and 2 do not cause HAM to fail, which is not practical for systems whose operating environments are dynamic and unpredictable. Furthermore, they are not solutions to Problem 1.2 and Problem 1.3. Specifically, in all these architectures, HAM and HPM are independent, that is, HPM cannot learn from HAM, and HAM cannot teach HPM how to be safe. Meanwhile, HAM is the static model-based controller, and its action will be unreliable if the real-time unknown unknowns and Sim2Real gap create a significant model mismatch.

## 1.3 CONTRIBUTION: RUNTIME LEARNING MACHINE: FROM THEORY TO IMPLEMENTATION

To address Problem 1.2 and Problem 1.3 for delivering Prospect 1.1, we propose the **Runtime**

**Learning Machine**, whose framework is shown in Figure 1. The machine constitutes high-performance (HP)-Student, high-assurance (HA)-Teacher, and Coordinator. HP-Student is a Phy-DRL agent that can be pre-trained and continue to learn in real plants that operate in real-time physical environments. HA-Teacher is a verified and physics-based design, with its functionality being reduced to a safety-critical level. Coordinator manages interactions between HP-Student and HA-Teacher. As a metaphor, HP-Student's runtime learning in our machine is like a student's journey. First, he learns from teachers in middle school, high school, college, etc., who have verified domain knowledge in subjects like physics and mathematics, to gain essential knowledge. Then, he delves deeper into specific areas during graduate studies to acquire expertise in those fields. Summarily, our runtime machine learning has following three distinct characteristics.

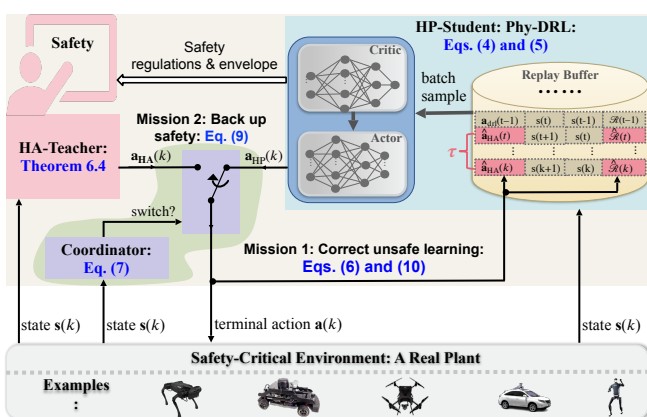

Figure 1: Runtime learning machine framework.

**Characteristic 1: Automatic Hierarchy Learning Mechanism**. HP-Student's growth in our runtime learning machine is an automatic hierarchical learning mechanism that respects safety-first principles for safety-critical autonomous systems without compromising mission performance. As depicted in Figure 7 in Appendix A, HP-Student undergoes a two-stage learning process:

- **Stage 1: Safety-first Learning**. HP-Student first learns from HA-Teacher how to be safe (i.e., constraining the system states of real plants into a safety set). Meanwhile, Figure 7 illustrates that prioritizing safety does not compromise mission performance. In other words, violating safety protocols results in decreased mission performance.
- **Stage 2: Self High-performance Learning.** After HP-Student has learned how to control system states within safety envelopes, Coordinator rarely activates HA-Teacher. Consequently, HP-Student engages in self-learning within the safety envelope for a high-performance action policy, such as the car closely following the planned blue path in stage 2 in Figure 7 in Appendix A.

**Characteristic 2: Assuring Safety by Tolerating Unknown Unknowns and Sim2Real Gap**. HA-Teacher's real-time patch, enabled by a real-time model, real-time action mission, and real-time model-based policy computation, aim to ensure lifetime safety. This means guaranteeing the safety of real plants during any runtime learning stages, regardless of HP-Student's failures, and in the face of real-time unknown unknowns and the Sim2Real gap.

**Characteristic 3: Highly Interactive HP-Student and HA-Teacher** The interactions between HP-Student and HA-Teacher in the runtime learning machine occur in two dimensions:

- **HP-Student $\longrightarrow$ HA-Teacher**: HP-Student shares his safety regulations and envelope with HA-Teacher for his real-time patch design.
- **HP-Student $\longleftarrow$ HA-Teacher**: Showing in Figure 1, HA-Teacher has two missions: i) correct unsafe learning of HP-Student and ii) back up the safety of the real plants, in the face of unknown unknowns and Sim2Real gap.

**Note**: Appendix B summarizes notations used throughout the paper.

## 2 PRELIMINARIES: DEFINITIONS OF SAFETY AND HIGH PERFORMANCE

We introduce the dynamics model of a DRL-enabled real plant:

$$\mathbf{s}(k+1) = \mathbf{A}(\mathbf{s}(k)) \cdot \mathbf{s}(k) + \mathbf{B}(\mathbf{s}(k)) \cdot \mathbf{a}(k) + \mathbf{f}(\mathbf{s}(k)), \ k \in \mathbb{N} \tag{1}$$

whose equilibrium point is $\mathbf{s}^* = \mathbf{0}$. In Equation (1), $\mathbf{f}(\mathbf{s}(k)) \in \mathbb{R}^n$ is the model mismatch, $\mathbf{A}(\mathbf{s}(k)) \in \mathbb{R}^{n \times n}$ and $\mathbf{B}(\mathbf{s}(k)) \in \mathbb{R}^{n \times m}$ denote system matrix and control structure matrix, respectively, $\mathbf{s}(k) \in \mathbb{R}^n$ is real plant's state in real-time, $\mathbf{a}(k) \in \mathbb{R}^m$ is the action command in real-time.

The safety issue arises from the system's state $\mathbf{s}(k)$ and the associated safety regulations, defining the permissible state space for the system:

$$\text{Safety set: } \mathbb{X} \triangleq \left\{ \mathbf{s} \in \mathbb{R}^n | \underline{\mathbf{v}} \leq \mathbf{D} \cdot \mathbf{s} \leq \overline{\mathbf{v}}, \quad \text{with } \mathbf{D} \in \mathbb{R}^{h \times n}, \overline{\mathbf{v}}, \underline{\mathbf{v}} \in \mathbb{R}^h \right\}. \tag{2}$$

where $\mathbf{D}$, $\overline{\mathbf{v}}$ and $\underline{\mathbf{v}}$ are given in advance for formulating $h \in \mathbb{N}$ safety regulations. Inequalities in Equation (2) are generic, as they can cover many safety regulations, such as speed regulation, collision avoidance, lane keeping, and tracking for autonomous vehicles. Building on the safety set, the lifetime safety is formally defined below.

**Definition 2.1** (Lifetime Safety). Consider the safety set $\mathbb{X}$ in Equation (2). The real plant in Equation (1) is said to have lifetime safety, if given any $\mathbf{s}(1) \in \mathbb{X}$, the $\mathbf{s}(k) \in \mathbb{X}$ holds at any time $k \in \mathbb{N}$, regardless of HP-Student's failure.

**Definition: High Performance.** 'High Performance' in this paper has two-dimensional definition: 1) mission performance (measured by, for example, tracking errors in the lane-tracking and path-following tasks) and 2) operation performance (measured by, for example, jerky movements for customers' comfort in autonomous vehicles). In the learning machine, HP-Student's reward encodes safety regulations, mission, and operation for learning a high-performance action policy with a safety guarantee. On the other hand, HA-Teacher's function is reduced to be safety-critical only, and his performance consideration is only about the operation regulations.

## 3  DESIGN OVERVIEW

Our proposed runtime learning machine aims to address Problem 1.2 and Problem 1.3 to deliver Prospect 1.1. To do so, showing in Figure 1, it is designed to have three interactive components:

- **HP-Student** builds on Phy-DRL agent, which can be pre-trained in a simulator or another domain but performs runtime learning in a real plant to tolerate unknown unknowns and address the Sim2real gap.
- **HA-Teacher** is a verified safety-only design whose novelty lies in real-time patches with two missions: timely correcting unsafe learning and backing up safety.
- **Coordinator** is responsible for monitoring the real-time safety status and facilitating interactions between HP-Student and HA-Teacher. Specifically, when the real-time safety status of the plant being controlled by HP-Student approaches the safety boundary, Coordinator prompts HA-Teacher to intervene and assure the safety of real plant, and correct unsafe learning of HP-Student. When the real-time states return to a safe region, Coordinator triggers the switch back to HP-Student and terminates the learning correction.

Next, we will describe the designs of the three interactive components in Sections 4 to 6, respectively.

## 4  RUNTIME LEARNING MACHINE: HP-STUDENT COMPONENT

### 4.1  HP-STUDENT CANDIDATES

We acknowledge that DRL is unable to directly embed high-dimensional or many safety regulations (represented by $h \in \mathbb{N}$ in Equation (2)) into the reward function due to the reward being a one-dimensional real value, creating a dimension gap. To bridge this dimension gap, the literature Cao et al. (2024; 2023) introduces the concept of a safety envelope, which has the one-dimensional condition and can be designed as a subset of the safety set $\mathbb{X}$ (refer to Figures 2 and 7 for visualization).

$$\text{Safety envelope: } \Omega \triangleq \left\{ \mathbf{s} \in \mathbb{R}^n | \mathbf{s}^\top \cdot \mathbf{P} \cdot \mathbf{s} \leq 1, \mathbf{P} \succ 0 \right\}. \tag{3}$$

So, our HP-Student candidates are those whose rewards can successfully embed the safety envelope in Equation (3), and such safety-embedded rewards can be shared with HA-Teacher for his real-time patch design. Along with this direction, DRL with CLF-like reward proposed in Westenbroek et al. (2022) and Phy-DRL (physics-regulated DRL) proposed in Cao et al. (2024; 2023) are two preferred candidates, as they can successfully embed the safety envelope into their rewards. Finally, HP-Student adopts Phy-DRL because Phy-DRL also features fast training theoretically and experimentally, which is desirable for runtime learning in real plants. Next, we will review the HP-Student design.

## 4.2 HP-Student: Phy-DRL: Residual Action Policy and Safety-embedded Reward

Recalling Phy-DRL in Cao et al. (2024; 2023), HP-Student has residual action policy formula:

$$\mathbf{a}_{\text{HP}}(k) = \underbrace{\mathbf{a}_{\text{drl}}(k)}_{\text{data-driven}} + \underbrace{\mathbf{a}_{\text{phy}}(k)}_{\text{model-based}} (= \mathbf{F} \cdot \mathbf{s}(k)), \tag{4}$$

where $\mathbf{a}_{\text{drl}}(k)$ denotes a date-driven action from DRL, while $\mathbf{a}_{\text{phy}}(k)$ is a model-based action. Referring to safety envelope in Equation (3), HP-Student's safety-embedded reward is

$$\mathcal{R}(\mathbf{s}(k), \mathbf{a}_{\text{drl}}(k)) = \underbrace{\mathbf{s}^{\top}(k) \cdot \mathbf{P} \cdot \mathbf{s}(k) - \mathbf{s}^{\top}(k+1) \cdot \mathbf{P} \cdot \mathbf{s}(k+1)}_{\triangleq\, r(\mathbf{s}(k),\, \mathbf{s}(k+1))} + w(\mathbf{s}(k), \mathbf{a}_{\text{HP}}(k)), \tag{5}$$

where the sub-reward $r(\mathbf{s}(k), \mathbf{s}(k + 1))$ is safety-embedded, while the sub-reward $w(\mathbf{s}(k), \mathbf{a}(k))$ aims at high operation performance (e.g., minimizing energy consumption of resource-limited robots and avoiding jerks for customers' comfort in autonomous vehicles). The matrices $\mathbf{F}$ in Equation (4) and $\mathbf{P}$ in Equation (3) and Equation (5) are the design variables. Their automatic computation by the CVXPY toolbox is detailed in Cao et al. (2024).

*Remark* 4.1 (**Safety- And Also Mission-Embedded**). The equilibrium $\mathbf{s}^* = \mathbf{0}$ means that the system described in Equation (1) can be interpreted as the dynamics of mission-tracking error. For instance, in a path-following task, the path represents the mission goal, while $\mathbf{s}(k)$ denotes the real-time tracking error of the path. Additionally, as indicated in Equation (3), the center of the safety envelope is the $\mathbf{s}^* = \mathbf{0}$. Based on this, we can conclude that the sub-reward $r(\mathbf{s}(k), \mathbf{s}(k + 1))$ defined in Equation (5) encompasses both safety and mission considerations, and HP-Student's learning encourages actions that increase $r(\mathbf{s}(k), \mathbf{s}(k + 1))$ over time. Furthermore, an increase in $r(\mathbf{s}(k), \mathbf{s}(k + 1))$ signifies progress towards both the envelope center and the mission goal. This also explains Figure 7, where prioritizing safety does not compromise mission performance (violating safety protocols results in decreased mission performance).

## 4.3 HP-Student: Correction of Unsafe Runtime Learning

HP-Student can be pre-trained in a simulator or another domain, and then he performs runtime learning in real plants within a real-time physical environment. HP-Student utilizes the actor-critic architecture-based DRL such as those outlined in Lillicrap et al. (2016) and Haarnoja et al. (2018) for runtime learning, in order to learn a safe data-driven policy that maximizes the expected return. HP-Student consists of an action policy and an action-value function.

Sampling efficiency is crucial for runtime learning. Experience replay (ER) Andrychowicz et al. (2017) enables off-policy algorithms to reuse past experiences, significantly improving sampling efficiency and preventing forgetting of learned knowledge Khetarpal et al. (2022). ER also helps break the correlation between adjacent transitions to avoid sampling bias for a stable learning process, which is important when online data is limited due to the expensive interaction with physical systems. During online inference, we continuously store real transitions resulting from the actions of HP-Student and corrected unsafe actions by HA-Teacher in the replay buffer. As shown in Figure 1, if the action $\mathbf{a}_{\text{HP}}(k)$ from HP-Student leads to unsafe behavior of a real plant, HA-Teacher takes control to ensure safety of real plant, and corrects the unsafe data-driven action to $\widehat{\mathbf{a}}_{\text{HA}}(k)$ and the corresponding reward to $\widehat{\mathcal{R}}(k)$, according to

$$\mathbf{a}_{\text{drl}}(k) \leftarrow \widehat{\mathbf{a}}_{\text{HA}}(k) \triangleq \mathbf{a}_{\text{HA}}(k) - \mathbf{a}_{\text{phy}}(k), \qquad \mathcal{R}(\mathbf{s}(k), \mathbf{a}_{\text{drl}}(k)) \leftarrow \widehat{\mathcal{R}}(k) \triangleq \mathcal{R}(\mathbf{s}(k), \widehat{\mathbf{a}}_{\text{HA}}(k)), \tag{6}$$

where $\mathbf{a}_{\text{phy}}(k)$ is HP-Student's model-based action in Equation (4), and $\mathbf{a}_{\text{HA}}(k)$ is the action from HA-Teacher, whose design is presented in Section 6. In the meantime, during runtime learning, a minibatch of transitions is uniformly sampled for training or learning Fujimoto et al. (2018).

*Remark* 4.2. Equation (6) states that according to HP-Student's residual action policy in Equation (4), the action correction is only applied to the data-driven $\mathbf{a}_{\text{drl}}(k)$, as the model-based action policy $\mathbf{a}_{\text{phy}}(k) = \mathbf{F} \cdot \mathbf{s}(k)$ is invariant.

## 5 Runtime Learning Machine: Coordinator Component

Coordinator manages interactions between HP-Student and HA-Teacher according to

$$\text{Triggering condition: } \mathbf{s}^{\top}(k - 1) \cdot \mathbf{P} \cdot \mathbf{s}(k - 1) \leq 1 \text{ and } \mathbf{s}^{\top}(k) \cdot \mathbf{P} \cdot \mathbf{s}(k) > 1, \tag{7}$$

coupled with which, we introduce the active time phase of HA-Teacher:

$$\text{HA-Teacher's active phase: } \mathbb{T}_{\sigma(k)} \triangleq \{k, \, k+1, \, \dots, \, k+\tau\}, \ \ \tau \in \mathbb{N} \tag{8}$$

where $\sigma(k)$ represents a piece-wise signal for notation. For instance, $\sigma(k) = i$ for $k \in \mathbb{T}_{\sigma(k)}$ signifies the $i$-th time that HA-Teacher is triggered, and its active phase this time is $\mathbb{T}_i$. The switching logic of actions applied to a real plant for backing up safety is as follows:

$$\mathbf{a}(t) \leftarrow \begin{cases} \mathbf{a}_{\text{HA}}(t), & \text{if triggering condition (7) holds at } k \text{ and } t \in \mathbb{T}_{\sigma(k)} \\ \mathbf{a}_{\text{HP}}(t), & \text{otherwise} \end{cases} \tag{9}$$

synchronizing with which is the correcting logic of HP-Student's unsafe action and reward:

$$\mathbf{a}_{\text{drl}}(t) \leftarrow \begin{cases} \widehat{\mathbf{a}}_{\text{HA}}(t), & \text{if triggering condition (7) holds at } k \text{ and } t \in \mathbb{T}_{\sigma(k)} \\ \mathbf{a}_{\text{drl}}(k), & \text{otherwise} \end{cases} \tag{10a}$$

$$\mathcal{R}(t) \leftarrow \begin{cases} \widehat{\mathcal{R}}(t), & \text{if triggering condition (7) holds at } k \text{ and } t \in \mathbb{T}_{\sigma(k)} \\ \mathcal{R}(\mathbf{s}(t), \mathbf{a}_{\text{drl}}(t)), & \text{otherwise} \end{cases} \tag{10b}$$

where $\widehat{\mathbf{a}}_{\text{HA}}(t)$ and $\widehat{\mathcal{R}}(t)$ are the corrected action and reward by HA-Teacher, defined in Equation (6).

*Remark* 5.1 (**Enabling Automatic Hierarchy Learning**). Operating within the safety envelope, Coordinator activates HA-Teacher, if the real-time states of the real plant move outside the safety envelope. Once the active phase ends, control transitions back to HP-Student, and HA-Teacher's correction of unsafe learning concludes. If condition (7) is no longer met, HP-Student will have successfully learned to control the real plant within safety envelope, and continual runtime learning will then focus on achieving high mission and operation performance.

*Remark* 5.2 (**Active Phase**). Referring to Equations (8) to (10), the symbol $\tau$ represents the correction horizon for unsafe action and reward of HP-Student and the dwell time of HA-Teacher. Its allowable minimum value is one. However, if the value of $\tau$ is very small, the patch center may not sufficiently attract system states to the envelope inside, and HA-Teacher will dominate the learning machine, only ensuring safety. Corollary E.1 in Appendix E guides determining the appropriate value for $\tau$.

## 6 RUNTIME LEARNING MACHINE: HA-TEACHER COMPONENT

Enabling runtime learning in real plants is straightforward in addressing the Sim2Real gap, but not so for unknown unknowns, because unknown unknowns lack historical data and cannot be predicted in time and distribution. When an unknown unknown creates safety issues in a time-critical environment, it is crucial to update the dynamics models, action plans, and mission goals promptly to ensure safe and effective responses in real time. The insight inspires us to develop the real-time patch as the HA-Teacher. Its model knowledge, action policy, and mission goal are dynamic and real-time. The mathematical formula for a real-time patch is

$$\Psi_{\sigma(k)} \triangleq \{ \mathbf{s} \mid (\mathbf{s} - \chi \cdot \widehat{\mathbf{s}}_{\sigma(k)})^\top \cdot \widehat{\mathbf{P}}_{\sigma(k)} \cdot (\mathbf{s} - \chi \cdot \widehat{\mathbf{s}}_{\sigma(k)}) \leq (1 - \chi)^2 \cdot \widehat{\mathbf{s}}_{\sigma(k)}^\top \cdot \widehat{\mathbf{P}}_{\sigma(k)} \cdot \widehat{\mathbf{s}}_{\sigma(t)} \}, \tag{11}$$

coupled with which is the real-time action policy:

$$\mathbf{a}_{\text{HA}}(k) = \widehat{\mathbf{F}}_{\sigma(k)} \cdot (\mathbf{s}(k) - \chi \cdot \widehat{\mathbf{s}}_{\sigma(k)}), \quad \text{with } \chi \in (0, 1) \text{ such that } \chi^2 \cdot \widehat{\mathbf{s}}_{\sigma(k)}^\top \cdot \mathbf{P} \cdot \widehat{\mathbf{s}}_{\sigma(k)} < 1, \tag{12}$$

where $\widehat{\mathbf{P}}_{\sigma(k)} \succ 0$, the $\chi \cdot \widehat{\mathbf{s}}_{\sigma(k)}$ represents the patch center (i.e., the yellow dots in Figure 2), and the $\widehat{\mathbf{s}}_{\sigma(k)}$ denotes the real-time state that triggers HA-Teacher and remains constant for defining patch center during HA-Teacher's active phase $\mathbb{T}_{\sigma(k)}$ (defined in Equation (8)), i.e.,

$$\widehat{\mathbf{s}}_{\sigma(t)} = \mathbf{s}(k) \ \text{ for } \ t \in \mathbb{T}_{\sigma(k)}, \text{ with } \mathbf{s}(k) \text{ satisfying triggering condition (7).} \tag{13}$$

*Remark* 6.1 (**Why called patch?**). In today's world, there are two approaches to achieving the same control task: a high-dimensional data-driven DRL and a low-dimensional physics-model-based controller. The data-driven DRL provides superior performance but is challenging to verify (due to DNN's huge parameter, nonlinear activation, etc.). On the other hand, the physics-model-based approach offers analyzable and verifiable behavior but has limited performance (due to model mismatch). This explains why the set in Equation (11) follows a very similar safety envelope formula in Equation (3), but it is referred to as a patch: the envelope represents a DRL design, while the patch represents a physics-model-based design with a small verifiable-safety region.

When a plant under the control of HP-Student experiences a safety violation at time $k$ (as indicated by the condition in Equation (7)), Coordinator activates HA-Teacher. HA-Teacher then utilizes real-time sensor data $\widehat{\mathbf{s}}_{\sigma(k)}$ to update the physics-model knowledge $(\mathbf{A}(\widehat{\mathbf{s}}_{\sigma(k)}), \mathbf{B}(\widehat{\mathbf{s}}_{\sigma(k)}))$. This update is used to compute the real-time patch in Equation (11) and the coupled action policy in Equation (12). The real-time patch and its coupled action policy will empower HA-Teacher to achieve backing up safety and correcting unsafe learning of HP-Student. However, to deliver the targeted capabilities, real-time patch must meet following three requirements.

**Requirement 1: Attracting Toward Safety Envelope**. The center of the patch must be within the safety envelope. If it's not, as shown by the patch $\Psi_4$ in Figure 2, the system's state can get stuck in the patch. This can lead to HA-Teacher dominating the machine during runtime learning, and HP-Student being unable to self-learn for a high-performance action policy.

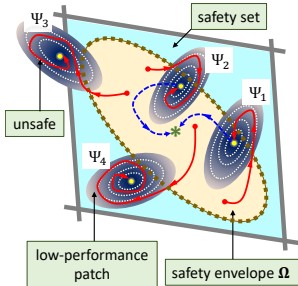

Figure 2: System behavior.

**Requirement 2: Conformity with Safety Regulations**. The real-time patches must be subsets of the safety set in Equation (2). If not, the patch will not be able to ensure safety, as shown by patch $\Psi_3$ in Figure 2, where the system states leave the safety set.

**Requirement 3: Conformity with Operation Regulations**. It is necessary to confine the real-time action $\mathbf{a}_{\text{HA}}(k)$ within a physically-feasible bounded action space:

$$\mathbb{A} \triangleq \left\{ \mathbf{a}_{\text{HA}} \in \mathbb{R}^m \,\middle|\, \underline{\mathbf{z}} \leq \mathbf{C} \cdot \mathbf{a}_{\text{HA}} \leq \overline{\mathbf{z}}, \text{ with } \mathbf{C} \in \mathbb{R}^{g \times m}, \, \underline{\mathbf{z}}, \, \overline{\mathbf{z}} \in \mathbb{R}^m \right\}, \tag{14}$$

where $\mathbf{C}, \overline{\mathbf{z}}$ and $\underline{\mathbf{z}}$ are given in advance for formulating operation regulations.

The $\widehat{\mathbf{F}}_{\sigma(k)}$ and $\widehat{\mathbf{P}}_{\sigma(k)}$ in Equation (12) and Equation (11) are our design variables for delivering the real-time patch and coupled action policy. Our design focus is on how $\widehat{\mathbf{F}}_{\sigma(k)}$ and $\widehat{\mathbf{P}}_{\sigma(k)}$ can meet Requirements 1–3. We observe from Equations (11) and (12) that the patch center $\chi \cdot \widehat{\mathbf{s}}_{\sigma(k)}$ meets Requirement 1 because it is located inside the safety envelope (due to $\chi^2 \cdot \widehat{\mathbf{s}}_{\sigma(k)}^\top \cdot \mathbf{P} \cdot \widehat{\mathbf{s}}_{\sigma(k)} < 1$) to attract systems toward the envelope. So, the remaining task is to follow Requirements 2 and 3 to design $\widehat{\mathbf{F}}_{\sigma(k)}$ and $\widehat{\mathbf{P}}_{\sigma(k)}$, which relies on a tracking-error dynamics model obtained from Equation (1):

$$\mathbf{e}(k+1) = \mathbf{A}(\widehat{\mathbf{s}}_{\sigma(k)}) \cdot \mathbf{e}(k) + \mathbf{B}(\widehat{\mathbf{s}}_{\sigma(k)}) \cdot \mathbf{a}_{\text{HA}}(k) + \mathbf{h}(\mathbf{e}(k)), \text{ with } \mathbf{e}(k) \triangleq \mathbf{s}(k) - \chi \cdot \widehat{\mathbf{s}}_{\sigma(k)}. \tag{15}$$

*Remark* 6.2. The design of HA-Teacher needs the knowledge of system dynamics model, denoted as $(\mathbf{A}(\widehat{\mathbf{s}}_{\sigma(k)}), \mathbf{B}(\widehat{\mathbf{s}}_{\sigma(k)}))$. The dynamics of safety-critical autonomous systems have been extensively studied, allowing us to access the dynamics models of many autonomous systems. For instance, dynamics models of quadruped robots, drones, and autonomous vehicles can be found in the works of Di Carlo et al. (2018), Yuan et al. (2022), and Rajamani (2011), respectively.

Next, we present a practical and common assumptions regarding the model mismatch for the design.

**Assumption 6.3.** The model mismatch in $\mathbf{h}(\cdot)$ in Equation (15) is locally Lipschitz in $\Psi_{\sigma(k)}$, i.e.,

$$(\mathbf{h}(\mathbf{e}_1) - \mathbf{h}(\mathbf{e}_2))^\top \cdot \widehat{\mathbf{P}}_{\sigma(k)} \cdot (\mathbf{h}(\mathbf{e}_1) - \mathbf{h}(\mathbf{e}_2)) \leq \kappa \cdot (\mathbf{e}_1 - \mathbf{e}_2)^\top \cdot \widehat{\mathbf{P}}_{\sigma(k)} \cdot (\mathbf{e}_1 - \mathbf{e}_2), \, \forall \mathbf{e}_1, \mathbf{e}_2 \in \Psi_{\sigma(k)}.$$

We also assume that the computing hardware, mechanical components, sensors, and operating systems function correctly.

Theorem 6.4 presents the design, meeting Requirements 2 and 3; its proof is in Appendix D.2.

**Theorem 6.4** (**Real-time Patch Design**). *Consider the HA-Teacher's action policy in Equation* (12), *the patch* $\Psi_{\sigma(k)}$ *in Equation* (11), *and the action space* $\mathbb{A}$ *in Equation* (14), *where the matrices* $\widehat{\mathbf{F}}_{\sigma(k)}$ *and* $\widehat{\mathbf{P}}_{\sigma(k)}$ *are computed according to*

$$\widehat{\mathbf{F}}_{\sigma(k)} = \widehat{\mathbf{R}}_{\sigma(k)} \cdot \widehat{\mathbf{Q}}_{\sigma(k)}^{-1}, \quad \widehat{\mathbf{P}}_{\sigma(k)} = \widehat{\mathbf{Q}}_{\sigma(k)}^{-1}, \tag{16}$$

*with* $\widehat{\mathbf{R}}_{\sigma(k)}$ *and* $\widehat{\mathbf{Q}}_{\sigma(k)}$ *satisfying the conditions in Equations* (18), (23) *and* (27) *to* (30). *Under Assumption 6.3, the following properties hold, where* $0 < \alpha < 1$ *and* $\mathbb{T}_{\sigma(k)}$ *is defined in Equation* (8).

    *1. The real-time patch* $\Psi_{\sigma(k)} \subseteq \mathbb{X}$ *holds for any time* $k$.

    *2. The* $\mathbf{e}^\top(t+1) \cdot \widehat{\mathbf{P}}_{\sigma(t)} \cdot \mathbf{e}(t+1) \leq \alpha \cdot \mathbf{e}^\top(t) \cdot \widehat{\mathbf{P}}_{\sigma(t)} \cdot \mathbf{e}(t)$ *holds for any time* $t \in \mathbb{T}_{\sigma(k)}$.

    3. *The HA-Teacher's real-time action satisfies* $\mathbf{a}_{HA}(t) \in \mathbb{A}$ *for any time* $t \in \mathbb{T}_{\sigma(k)}$.

*Remark* 6.5 (**Fast Computation Time**). The $\widehat{\mathbf{F}}_{\sigma(k)}$ and $\widehat{\mathbf{P}}_{\sigma(k)}$ are automatically computed from Equations (16), (18), (23) and (27) to (30), using the LMI toolbox Gahinet et al. (1994); Boyd et al. (1994). The computation time is quite short (0.01–0.04 seconds); its impact can be disregarded.

*Remark* 6.6 (**Safety Knowledge from HP-Student**). The safety regulations and envelope provided by HP-Student are applied in Equations (18) and (27) for the patch design. The resulting properties in Items 1 and 3 of Theorem 6.4 show that the designed patch meets Requirements 2 and 3. The property in Item 2 is used to develop guidance (i.e., Corollary E.1 in Appendix E) for determining $\tau$, which is the dwell time and correction horizon of HA-Teacher.

# 7 EXPERIMENT

The experiment involved comprehensive comparisons, a cart-pole system, and a real A1 quadruped robot. **Note**: Appendix H.5.2 presents the experiment of two additional benchmarks: a Go2 quadruped robot and a 2D quadrotor.

## 7.1 CART-POLE SYSTEM

We pre-train HP-Student using the OpenAI Gym Brockman et al. (2016). The pre-training process includes domain randomization Sadeghi & Levine (2017); Nagabandi et al. (2019) to bridge the Sim2Real gap, through introducing random force disturbances and randomizing the friction force. We use the simulator to mimic the real plant. The Sim2Real gap is intentionally created by inducing a friction force that is out of the distribution of those in pre-training. Unknown unknowns are disturbances applied to HP-Student's action commands, generated by a randomized Beta distribution. Appendix F explains why the randomized Beta distribution can be one kind of unknown unknown.

The system's state consists of the pendulum angle $\theta$, the cart position $x$, and their respective velocities $\omega = \dot{\theta}$ and $v = \dot{x}$. The goal of HP-Student is to stabilize system at the equilibrium $\mathbf{s}^* = [0, 0, 0, 0]^\top$, while keeping the system states within the safety set $\mathbb{X} = \{\, \mathbf{s} \mid |x| \leq 1, \ |\theta| < 0.8 \,\}$. The action space of HA-Teacher is $\mathbb{A} = \{\, \mathbf{a}_{HA} \in \mathbb{R}| \, |\mathbf{a}_{HA}| \leq 40 \,\}$. The designs for HP-Student and HA-Teacher are presented in Appendix G.3 and Appendix G.4, respectively. Additionally, Appendix G.1 presents the pre-training and runtime learning configurations.

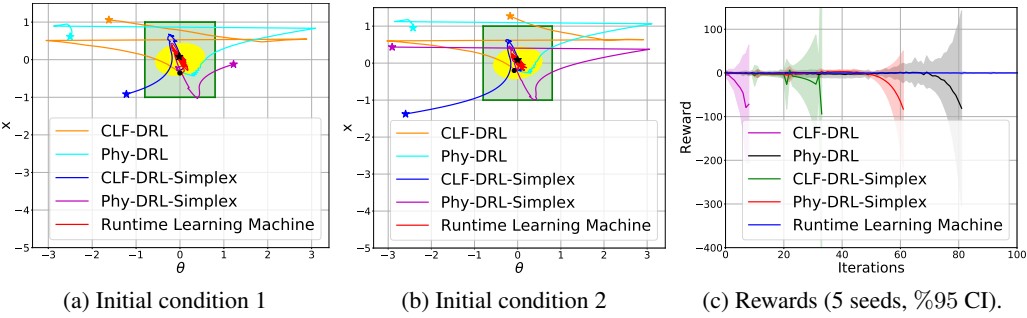

    (a) Initial condition 1         (b) Initial condition 2         (c) Rewards (5 seeds, $\%95$ CI).

Figure 3: (a) and (b): Phase plots in **Episode 1** under two initial conditions, where the black dot and star denote the initial condition and final location, respectively. Green and yellow areas denote safety set and envelope, respectively. (c): Reward trajectories over five random seeds.

When we disable HA-Teacher's real-time patch and unsafe learning correction, our runtime learning machine degrades to the recently developed fault-tolerant DRL: runtime assurance Chen et al. (2022) and neural Simplex Phan et al. (2020). Since runtime assurance is an extension of Simplex Sha (2001), we refer to the two compared models as 'CLF-DRL-Simplex' and 'Phy-DRL-Simplex', with their high-performance components being the newly developed Phy-DRL Cao et al. (2024) and CLF-DRL Westenbroek et al. (2022), respectively. When HA-Teacher is completely disabled, they further degrade to pure Phy-DRL and CLF-DRL. Therefore, we now have five models for comparison. The phase plots of position and angle, as well as the trajectories of learning reward, are presented in Figure 3. Additionally, Figures 9 to 12 in Appendix G.5.1 include phase plots in episodes 5, 10,

15, and 20, respectively. It is shown in Figure 3 (a) and (b) and Figures 9 to 12 that our runtime learning machine can assure lifetime safety in the face of unknown unknowns and the Sim2Real gap, as system states (magenta curves) never leave the safety set (green area) in any learning stage. In contrast, current fault-tolerant DRL and safe DRL cannot achieve this. Meanwhile, as seen in Figure 3 (c), our runtime learning machine provides remarkably stable and fast agent learning.

We next showcase the automatic hierarchy learning mechanism. To do this, we disable HA-Teacher in episodes 5 and 20 and observe the system's behavior under the control of the sole HP-Student. The phase plot and trajectories of system with ten random initial conditions (each runs for 2000 steps) are displayed in Figures 4 and 13. Upon observing Figure 4 (a), we can conclude that HP-Student has successfully learned from the HA-Teacher how to ensure safety in episode 5: his action policy can confine the system states to the safety set (green area). HP-Student will automatically become independent of HA-Teacher and self-learn for a high-performance action policy. This is evident in Figure 4 (b) together with trajectories of HA-Teacher's activation ratio in Figure 14, where in episode 20, HP-Student consistently confines the system within her safety envelope (yellow area), and HA-Teacher is seldom triggered by the condition in Equation (7). Additionally, the action policy of HP-Student in episode 20 demonstrates higher mission performance: faster clustering and much closer proximity to the mission goal, as observed in Figure 4 (b) and Figure 13. Additional experiment on behavior or activation ratio of HA-Teacher is presented in Appendix G.5.3.

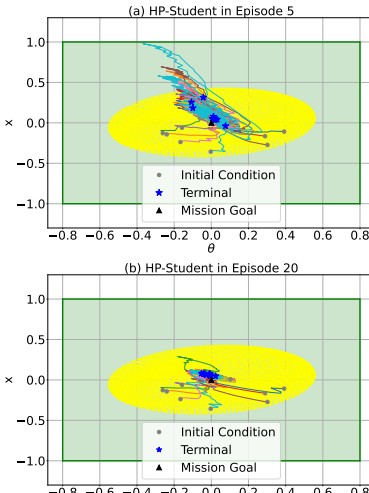

Figure 4: Demonstrating automatic hierarchy learning.

Finally, the experimental results in Figure 15 in Appendix G.5.4 emphasize HA-Teacher's unsafe learning correction in contributing to HP-Student's fast and stable learning with larger reward values.

## 7.2 REAL QUADRUPED ROBOT

The action policy's mission is to track the robot's center of mass (CoM) height, CoM x-velocity, and other states to the corresponding commands $r_{v_x}, r_h$, and zeros, constraining system states to a safety set $\mathbb{X} = \{ \mathbf{s} \mid |\text{CoM x-velocity} - r_{v_x}| \leq 0.3 \text{ m/s}, |\text{CoM z-height} - r_h| \leq 0.15 \text{ m} \}$. HA-Teacher's action space is $\mathbb{A} = \{ \mathbf{a}_{HA} \in \mathbb{R}^6 \mid |\mathbf{a}_{HA}| \leq [30, 30, 30, 60, 60, 60]^\top \}$. The designs of HP-Student and HA-Teacher are presented in Appendix H.3 and Appendix H.4, respectively. During pre-training in the simulator, we set $r_{v_x} = 0.6$ m/s and $r_h = 0.24$ m. To better demonstrate the runtime learning machine, the real robot's velocity command is $r_{v_x} = 0.35$ m/s, which is different from the one in simulator. For the runtime learning, one episode is defined as "*running robot for 15 seconds.*"

We compared the runtime learning machine with existing approaches to address the Sim2Real gap in training HP-Student in the simulator. The approach we compared is called 'delay + domain,' which involves concurrent delay randomization Imai et al. (2022) and domain randomization Sadeghi & Levine (2017) (by randomizing friction force). This approach resulted in two comparison models. 1) 'Continual Phy-DRL: delay + domain,' represents a well-trained Phy-DRL using the 'delay + domain' approach in the simulator, which performed continual learning in the real robot to fine-tune the action policy. 2) 'Phy-DRL: delay + domain,' represents the well-trained Phy-DRL policy directly deployed to the real robot. The comparison video for episode 1 is available at comparison video link [anonymous hosting and browsing] and the trajectories of the robot's CoM height and CoM-x velocity in episode 1 are shown in Figure 5. Additional

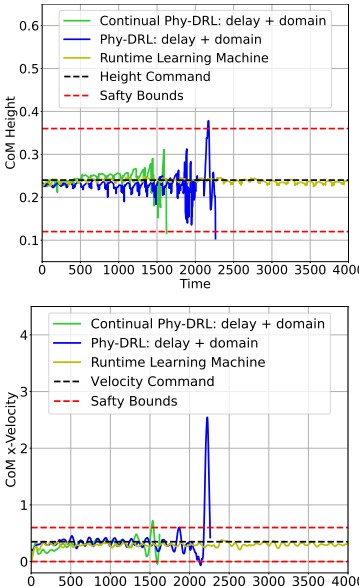

Figure 5: Trajectories.

trajectories for episodes 5, 10, 15, and 20 can be found in Figure 16 in Appendix H.5.1. After watching the comparison video and observing Figures 5 and 16, we concluded that a well-trained Phy-DRL in the simulator cannot guarantee the safety of the real robot due to the Sim2Real gap and unknown unknowns that the delay randomization and force randomization failed to capture. In contrast, our runtime learning machine can provide safety guarantee in any sampled learning episode.

We continue the comparison with 'Continual Phy-DRL: delay + domain.' It is a well-trained Phy-DRL in the simulator and performs continual learning in the real robot for 20 episodes. Figure 6 presents the trajectories of learning reward in terms of iteration steps and the episode-average reward. This demonstrates that the runtime learning machine features stable, fast, and safe learning in real plants. This notable feature is attributed to HA-Teacher's real-time patch for correcting unsafe learning and backing up safety. In addition, HA-Teacher enables HP-Student's safety-first learning from him in the learning machine. To verify this, we deactivate HA-Teacher in episodes 1 and 20, and compare system behavior. The demonstration video is available at safety-first-learning video link [anonymous hosting and browsing], which illustrates that HP-Student quickly learned from HA-Teacher to be safe, within 20 episodes (i.e., 300 seconds).

Finally, we showcase the learning machine's ability to tolerate various unknown unknowns. In addition to inherent unknowns, our experiment includes five unknown unknowns that have never occurred in HP-Student's historical training and learning. They are 1) **Beta**: Disturbances injected into HP-Student's actions, generated by a randomized Beta distribution (see Appendix F for

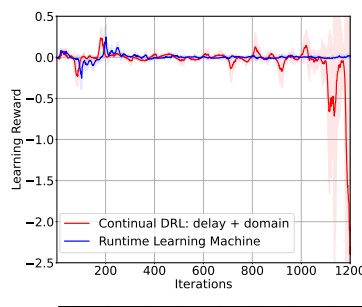

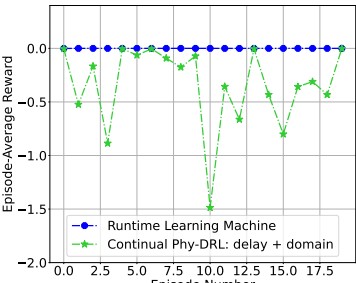

Figure 6: Rewards.

an explanation of its representation of unknown unknowns); 2) **PD**: Random and sudden payload (around 4 lbs) drops on the robot's back; 3) **Kick**: Random and sudden kick by a human; 4) **DoS**: A real denial-of-service fault of the platform, which can be caused by task scheduling latency, communication delay, communication block, etc., but is unknown to us; and 5) **SP**: A sudden side push. We consider three combinations of these unknown unknowns applied to the runtime learning stage: i) '**Beta + PD**,' ii) '**Beta + DoS + Kick**,' and iii) '**Beta + SP**.' The demonstration video is available at unknown-unknown video link [anonymous hosting and browsing]. Meanwhile, Figure 17 presents the corresponding trajectories. The demonstration video and Figure 17 demonstrate that our learning machine successfully ensures the safety of the real plant by tolerating such unknowns.

## 8 CONCLUSION AND DISCUSSION

This paper presents a runtime learning machine designed for safety-critical autonomous systems. The learning machine consists of the interactive HP-Student, HA-Teacher, and Coordinator. The machine's goal is to facilitate runtime learning for a high-performance action policy with verified safety in real plants, using real-time sensor data from real-time physical environments. The learning machine ensures lifetime safety by accommodating unknown unknowns and addressing the Sim2Real gap. The runtime learning machine also serves as an automatic hierarchy learning mechanism for HP-Student. Hierarchically, HP-Student first learns from the HA-Teacher to prioritize safety. After mastering safety-first learning, HP-Student autonomously self-learns to develop a high-performance action policy with a safety guarantee. Our runtime learning machine has shown outstanding features compared to state-of-the-art safe DRL and fault-tolerant DRL, with approaches to addressing the Sim2Real gap. These were demonstrated through comprehensive experiments on a cart-pole system, two quadruped robots, and one 2D quadrotor.

Incorporating an early warning function into our runtime learning machine for the Coordinator's management of interaction between HP-Student and HA-Teacher constitutes our future research. Reachability through worst-case dynamics Anderson et al. (2020) could be a solution. Another future research is to enhance the robustness of our runtime learning machine in handling unknown unknowns and bridging the Sim2Real gap by utilizing concepts from differentiable simulation Song et al. (2024) and robust adaptive control Hovakimyan & Cao (2010).

ETHICS STATEMENT

This paper does not have ethics issues because its applications focus on learning-enabled autonomous systems and do not involve human subjects, animals, privacy, or social security.

REPRODUCIBILITY STATEMENT

The code to reproduce our experimental results has been uploaded with this paper as Supplementary Material. If accepted, the code will be open source on GitHub. Meanwhile, the paper has disclosed all the information needed to reproduce the experimental results. Please refer to Appendices G.3, G.4, H.3, H.4 and H.5.2 for design details of experiments and Appendices G.1, H.1, I.2.4 and K for the configurations of training and runtime learning, the computation resources, and the implementation in real robot.

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

# Appendices

# A ILLUSTRATION: HP-STUDENT'S HIERARCHY LEARNING

Figure 7 illustrates HP-Student's two learning stages in the hierarchy learning mechanism.

- **Stage 1:** If he cannot guarantee safety, he first learn from HA-Teacher for safety-first learning to prioritize safety.
- **Stage 2:** As he masters the capability of safety guarantee (i.e., constraining system states into his safety envelope), his continuous runtime learning rarely activates the HA-Teacher, allowing him to automatically self-learn a high-performance action policy.

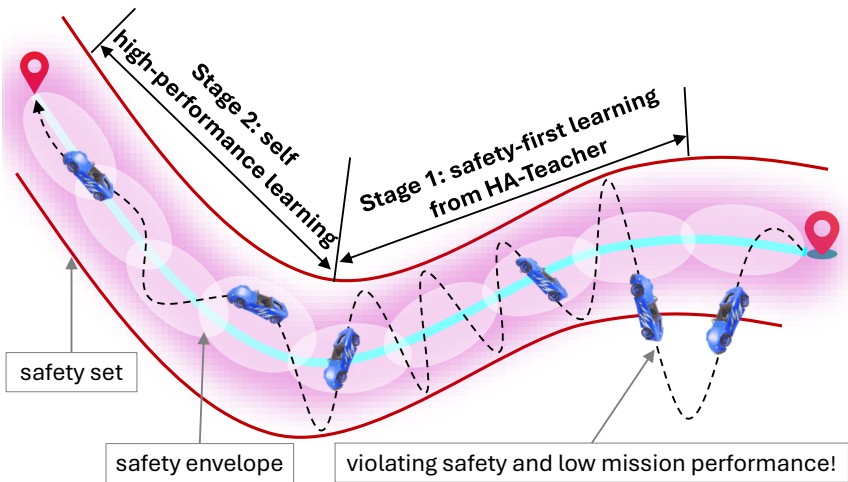

Figure 7: HP-Student's two learning stages.

## B  NOTATIONS

**Notations throughout Paper**

| | |
|---|---|
| $a$ | A scalar (integer or real) |
| $\mathbf{a}$ | A vector |
| $\mathbf{A}$ | A matrix |
| $\mathbb{A}$ | A set |
| $\mathbb{R}^n$ | Set of $n$-dimensional real vectors |
| $\mathbb{N}$ | Set of natural numbers |
| $[\mathbf{x}]_i$ | The $i$-th entry of vector $\mathbf{x}$ |
| $[\mathbf{W}]_{i,:}$ | The $i$-th row of matrix $\mathbf{W}$ |
| $[\mathbf{W}]_{:,j}$ | The $j$-th column of matrix $\mathbf{W}$ |
| $[\mathbf{W}]_{i,j}$ | Matrix $\mathbf{W}$'s element at row $i$ and column $j$ |
| $\mathbf{P} \succ (\prec) 0$ | Matrix $\mathbf{P}$ is positive (negative) definite |
| $\top$ | Transposition of a matrix or vector |
| $\lceil \cdot \rceil$ | Ceiling function |
| $\mathbf{P}^{-1}$ | Inverse of matrix $\mathbf{P}$ |
| $\ln(a)$ | Natural logarithm of the number $a > 0$ |
| $\cdot$ | Matrix multiplication |

## C    AUXILIARY LEMMAS

This section introduces the auxiliary lemmas used to establish the theoretical framework for our proposed runtime learning machine.

**Lemma C.1** (Schur Complement Zhang (2006)). *For any symmetric matrix* $\mathbf{M} = \begin{bmatrix} \mathbf{A} & \mathbf{B} \\ \mathbf{B}^\top & \mathbf{C} \end{bmatrix}$, *then*

$\mathbf{M} \succ 0$ *holds if and only if* $\mathbf{C} \succ 0$ *and* $\mathbf{A} - \mathbf{B}\mathbf{C}^{-1}\mathbf{B}^\top \succ 0$.

**Lemma C.2** (Cao et al. (2024)). *Consider the safety set* $\mathbb{X}$ *defined in Equation* (2) *and define a set*

$$\Omega_{\sigma(k)} \triangleq \{\, \mathbf{s} \mid \mathbf{s}^\top \cdot \widehat{\mathbf{Q}}_{\sigma(k)}^{-1} \cdot \mathbf{s} \le 1,\ \ \widehat{\mathbf{P}}_{\sigma(k)} \succ 0\}. \tag{17}$$

*We have* $\Omega_{\sigma(k)} \subseteq \mathbb{X}$ *if*

$$[\underline{\mathbf{D}}]_{i,:} \cdot \widehat{\mathbf{Q}}_{\sigma(k)} \cdot [\underline{\mathbf{D}}^\top]_{:,i} = \begin{cases} \ge 1, & [\mathbf{d}]_i = 1 \\ \le 1, & [\mathbf{d}]_i = -1 \end{cases}, \ \text{ and } \ [\overline{\mathbf{D}}]_{i,:} \cdot \widehat{\mathbf{Q}}_{\sigma(k)} \cdot [\overline{\mathbf{D}}^\top]_{:,i} \le 1,\ i \in \{1, \ldots, h\} \tag{18}$$

*where* $\overline{\mathbf{D}} = \frac{\mathbf{D}}{\overline{\Lambda}}$, $\underline{\mathbf{D}} = \frac{\mathbf{D}}{\underline{\Lambda}}$, *and for* $i, j \in \{1, \ldots, h\}$,

$$[\mathbf{d}]_i \triangleq \begin{cases} 1, & [\underline{\mathbf{v}}]_i > 0 \\ 1, & [\overline{\mathbf{v}}]_i < 0 \\ -1, & otherwise \end{cases}, \tag{19}$$

$$[\overline{\Lambda}]_{i,j} \triangleq \begin{cases} 0, & i \ne j \\ [\overline{\mathbf{v}}]_i, & [\underline{\mathbf{v}}]_i > 0 \\ [\underline{\mathbf{v}}]_i, & [\overline{\mathbf{v}}]_i < 0 \\ [\overline{\mathbf{v}}]_i, & otherwise \end{cases}, \tag{20}$$

$$[\underline{\Lambda}]_{i,j} \triangleq \begin{cases} 0, & i \ne j \\ [\underline{\mathbf{v}}]_i, & [\underline{\mathbf{v}}]_i > 0 \\ [\overline{\mathbf{v}}]_i, & [\overline{\mathbf{v}}]_i < 0 \\ -[\underline{\mathbf{v}}]_i, & otherwise \end{cases}. \tag{21}$$

**Lemma C.3.** *Consider the action set* $\mathbb{A}$ *defined in Equation* (14), *and*

$$\Xi \triangleq \{\, \mathbf{a}_{HA} \in \mathbb{R}^m \mid \mathbf{a}_{HA}^\top \cdot \mathbf{T}^{-1} \cdot \mathbf{a}_{HA} \le 1,\ \mathbf{V} \succ 0 \}. \tag{22}$$

*We have* $\Xi \subseteq \mathbb{A}$, *if*

$$[\underline{\mathbf{C}}]_{i,:} \cdot \mathbf{T} \cdot [\underline{\mathbf{C}}^\top]_{:,i} = \begin{cases} \ge 1, & [\mathbf{c}]_i = 1 \\ \le 1, & [\mathbf{c}]_i = -1 \end{cases}, \ \text{ and } \ [\overline{\mathbf{C}}]_{i,:} \cdot \mathbf{T} \cdot [\overline{\mathbf{C}}^\top]_{:,i} \le 1, \quad i \in \{1, \ldots, m\} \tag{23}$$

*where* $\overline{\mathbf{C}} = \frac{\mathbf{C}}{\overline{\Delta}}$ *and* $\underline{\mathbf{C}} = \frac{\mathbf{C}}{\underline{\Delta}}$, *and for* $i, j \in \{1, \ldots, m\}$:

$$[\mathbf{c}]_i \triangleq \begin{cases} 1, & [\underline{\mathbf{z}}]_i > 0 \\ 1, & [\overline{\mathbf{z}}]_i < 0 \\ -1, & otherwise \end{cases}, \tag{24}$$

$$[\overline{\Delta}]_{i,j} \triangleq \begin{cases} 0, & i \ne j \\ [\overline{\mathbf{z}}]_i, & [\underline{\mathbf{z}}]_i > 0 \\ [\underline{\mathbf{z}}]_i, & [\overline{\mathbf{z}}]_i < 0 \\ [\overline{\mathbf{z}}]_i, & otherwise \end{cases}, \tag{25}$$

$$[\underline{\Delta}]_{i,j} \triangleq \begin{cases} 0, & i \ne j \\ [\underline{\mathbf{z}}]_i, & [\underline{\mathbf{z}}]_i > 0 \\ [\overline{\mathbf{z}}]_i, & [\overline{\mathbf{z}}]_i < 0 \\ -[\underline{\mathbf{z}}]_i, & otherwise \end{cases}. \tag{26}$$

*Proof.* Lemma C.3's proof path is exactly the same as the proof of Lemma B.2 in Cao et al. (2024), so it is omitted here. □

**Lemma C.4.** *For two vectors* $\mathbf{x} \in \mathbb{R}^n$, $\mathbf{y} \in \mathbb{R}^n$, *and a matrix* $\mathbf{P} \succ 0$, *we have*

$$2 \cdot \mathbf{x}^\top \cdot \mathbf{P} \cdot \mathbf{y} \leq \gamma \cdot \mathbf{x}^\top \cdot \mathbf{P} \cdot \mathbf{x} + \frac{1}{\gamma} \cdot \mathbf{y}^\top \cdot \mathbf{P} \cdot \mathbf{y}, \ \ with \ \gamma > 0.$$

*Proof.* The proof is straightforward when we consider $\mathbf{P} \succ 0$ and recall the following inequality:

$$(\sqrt{\gamma} \cdot \mathbf{x} - \frac{1}{\sqrt{\gamma}} \cdot \mathbf{y})^\top \cdot \mathbf{P} \cdot (\sqrt{\gamma} \cdot \mathbf{x} - \frac{1}{\sqrt{\gamma}} \cdot \mathbf{y}) = \gamma \cdot \mathbf{x}^\top \cdot \mathbf{P} \cdot \mathbf{x} + \frac{1}{\gamma} \cdot \mathbf{y}^\top \cdot \mathbf{P} \cdot \mathbf{y} - 2 \cdot \mathbf{x}^\top \cdot \mathbf{P} \cdot \mathbf{y} \geq 0.$$

□

# D  THEOREM 6.4: CONDITIONS AND PROOF

## D.1  CONDITIONS OF REAL-TIME PATCH DESIGN IN THEOREM 6.4

The conditions for designing the real-time path are presented in Equations (18) and (23), and the remaining are given below.

$$\widehat{\mathbf{Q}}_{\sigma(k)} - \mu \cdot \mathbf{P}^{-1} \succ 0, \quad with\ \mu > 0 \tag{27}$$

$$(1 - \chi \cdot \gamma_1) \cdot \mu \geq 1 - 2 \cdot \chi + \frac{\chi}{\gamma_1} > 0, \tag{28}$$

$$\begin{bmatrix} \widehat{\mathbf{Q}}_{\sigma(k)} & \widehat{\mathbf{R}}_{\sigma(k)}^{\top} \\ \widehat{\mathbf{R}}_{\sigma(k)} & \mathbf{T} \end{bmatrix} \succ 0, \tag{29}$$

$$\begin{bmatrix} \left(\alpha - \kappa \cdot \left(1 + \frac{1}{\gamma_2}\right)\right) \cdot \widehat{\mathbf{Q}}_{\sigma(k)} & \widehat{\mathbf{Q}}_{\sigma(k)} \cdot \mathbf{A}^{\top}(\widehat{\mathbf{s}}_{\sigma(k)}) + \widehat{\mathbf{R}}_{\sigma(k)}^{\top} \cdot \mathbf{B}^{\top}(\widehat{\mathbf{s}}_{\sigma(k)}) \\ \mathbf{A}(\widehat{\mathbf{s}}_{\sigma(k)}) \cdot \widehat{\mathbf{Q}}_{\sigma(k)} + \mathbf{B}(\widehat{\mathbf{s}}_{\sigma(k)}) \cdot \widehat{\mathbf{R}}_{\sigma(k)} & \frac{\widehat{\mathbf{Q}}_{\sigma(k)}}{1+\gamma_2} \end{bmatrix} \succ 0, \tag{30}$$

where the $\widehat{\mathbf{Q}}_{\sigma(k)}, \widehat{\mathbf{R}}_{\sigma(k)}, \mathbf{T}$, and $\mu$ are variables, while the $\mathbf{P}, 0 < \chi < 1, \gamma_1 > 0, \gamma_2 > 0, 0 < \alpha < 1$, $\mathbf{A}(\widehat{\mathbf{s}}_{\sigma(k)})$, and $\mathbf{B}(\widehat{\mathbf{s}}_{\sigma(k)})$ are known and given in advance. The variables $\widehat{\mathbf{Q}}_{\sigma(k)}, \widehat{\mathbf{R}}_{\sigma(k)}, \mathbf{T}$, and $\mu$ can be automatically computed from Equations (18), (23) and (27) to (30) by using the available LMI toolbox Gahinet et al. (1994); Boyd et al. (1994).

## D.2  PROOF OF THEOREM 6.4

The three statements in Theorem 6.4 are proved separately.

### D.2.1  PROOF OF STATEMENT IN ITEM 1

The envelope patch in Equation (11) can be equivalently rewritten as:

$$\Psi_{\sigma(k)} = \left\{ \mathbf{s} \mid \mathbf{s}^{\top} \cdot \widehat{\mathbf{P}}_{\sigma(k)} \cdot \mathbf{s} \leq (1-\chi)^2 \cdot \mathbf{s}^{\top}(k) \cdot \widehat{\mathbf{P}}_{\sigma(k)} \cdot \mathbf{s}(k) + 2 \cdot \chi \cdot \mathbf{s}^{\top} \cdot \widehat{\mathbf{P}}_{\sigma(k)} \cdot \widehat{\mathbf{s}}_{\sigma(k)} \right.$$
$$\left. - \chi^2 \cdot \widehat{\mathbf{s}}_{\sigma(k)}^{\top} \cdot \widehat{\mathbf{P}}_{\sigma(k)} \cdot \widehat{\mathbf{s}}_{\sigma(k)} \right\}, \tag{31}$$

which, in light of Equation (13), equivalently transforms to

$$\Psi_{\sigma(k)} = \left\{ \mathbf{s} \mid \mathbf{s}^{\top} \cdot \widehat{\mathbf{P}}_{\sigma(k)} \cdot \mathbf{s} \leq (1 - 2 \cdot \chi) \cdot \mathbf{s}^{\top}(k) \cdot \widehat{\mathbf{P}}_{\sigma(k)} \cdot \mathbf{s}(k) + 2 \cdot \chi \cdot \mathbf{s}^{\top} \cdot \widehat{\mathbf{P}}_{\sigma(k)} \cdot \widehat{\mathbf{s}}(k) \right\}. \tag{32}$$

In light of Lemma C.4 in Appendix C, we have

$$2 \cdot \mathbf{s}^{\top} \cdot \widehat{\mathbf{P}}_{\sigma(k)} \cdot \mathbf{s}(k) \leq \gamma_1 \cdot \mathbf{s}^{\top} \cdot \widehat{\mathbf{P}}_{\sigma(k)} \cdot \mathbf{s} + \frac{1}{\gamma_1} \cdot \mathbf{s}^{\top}(k) \cdot \widehat{\mathbf{P}}_{\sigma(k)} \cdot \mathbf{s}(k), \ with\ \gamma_1 > 0$$

substituting which into the inequality in Equation (32) and considering $0 < \chi < 1$ yields

$$\mathbf{s}^{\top} \cdot \widehat{\mathbf{P}}_{\sigma(k)} \cdot \mathbf{s} \leq (1 - 2 \cdot \chi) \cdot \mathbf{s}^{\top}(k) \cdot \widehat{\mathbf{P}}_{\sigma(k)} \cdot \mathbf{s}(k) + 2 \cdot \chi \cdot \mathbf{s}^{\top} \cdot \widehat{\mathbf{P}}_{\sigma(k)} \cdot \widehat{\mathbf{s}}(k)$$
$$\leq (1 - 2 \cdot \chi + \frac{\chi}{\gamma_1}) \cdot \mathbf{s}^{\top}(k) \cdot \widehat{\mathbf{P}}_{\sigma(k)} \cdot \mathbf{s}(k) + \chi \cdot \gamma_1 \cdot \mathbf{s}^{\top} \cdot \widehat{\mathbf{P}}_{\sigma(k)} \cdot \mathbf{s}, \tag{33}$$

which leads to

$$(1 - \chi \cdot \gamma_1) \cdot \mathbf{s}^{\top} \cdot \widehat{\mathbf{P}}_{\sigma(k)} \cdot \mathbf{s} \leq (1 - 2 \cdot \chi + \frac{\chi}{\gamma_1}) \cdot \mathbf{s}^{\top}(k) \cdot \widehat{\mathbf{P}}_{\sigma(k)} \cdot \mathbf{s}(k). \tag{34}$$

We conclude from Equations (32) to (34) that if the inequality for defining the envelope patch $\Psi_{\sigma(k)}$ in Equation (32) holds, the inequality in Equation (34) holds as well. Therefore, we can define the first auxiliary set:

$$\Theta_1 = \left\{ \mathbf{s} \mid (1 - \chi \cdot \gamma_1) \cdot \mathbf{s}^{\top} \cdot \widehat{\mathbf{P}}_{\sigma(k)} \cdot \mathbf{s} \leq (1 - 2 \cdot \chi + \frac{\chi}{\gamma_1}) \cdot \mathbf{s}^{\top}(k) \cdot \widehat{\mathbf{P}}_{\sigma(k)} \cdot \mathbf{s}(k) \right\}, \tag{35}$$

and it satisfies

$$\Psi_{\sigma(k)} \subseteq \Theta_1. \tag{36}$$

Considering $\mu > 0$, we can conclude from Equation (28) that $1 - \chi \cdot \gamma > 0$. Therefore, the set in Equation (35) can be equivalently transformed to

$$\Theta_1 = \left\{ \mathbf{s} \mid \mathbf{s}^\top \cdot \widehat{\mathbf{P}}_{\sigma(k)} \cdot \mathbf{s} \le \frac{1 - 2 \cdot \chi + \frac{\chi}{\gamma_1}}{1 - \chi \cdot \gamma_1} \cdot \mathbf{s}^\top(k) \cdot \widehat{\mathbf{P}}_{\sigma(k)} \cdot \mathbf{s}(k) \right\}. \tag{37}$$

Considering $\widehat{\mathbf{P}}_{\sigma(k)} = \widehat{\mathbf{Q}}_{\sigma(k)}^{-1}$ and $\mu > 0$, the condition in Equation (27) is equivalent to

$$\frac{1}{\mu} \cdot \mathbf{P} \succ \widehat{\mathbf{P}}_{\sigma(k)},$$

substituting which into the inequality in Equation (37) results in

$$\mathbf{s}^\top \cdot \widehat{\mathbf{P}}_{\sigma(k)} \cdot \mathbf{s} \le \frac{1 - 2 \cdot \chi + \frac{\chi}{\gamma_1}}{1 - \chi \cdot \gamma_1} \cdot \mathbf{s}^\top(k) \cdot \widehat{\mathbf{P}}_{\sigma(k)} \cdot \mathbf{s}(k)$$

$$\le \frac{1 - 2 \cdot \chi + \frac{\chi}{\gamma_1}}{1 - \chi \cdot \gamma_1} \cdot \frac{1}{\mu} \cdot \mathbf{s}^\top(k) \cdot \mathbf{P} \cdot \mathbf{s}(k) = \frac{1 - 2 \cdot \chi + \frac{\chi}{\gamma_1}}{1 - \chi \cdot \gamma_1} \cdot \frac{1}{\mu}, \tag{38}$$

where last equality is obtained because $\mathbf{s}(k)$ approaches the boundary of the safety envelope, i.e., $\mathbf{s}^\top(k) \cdot \mathbf{P} \cdot \mathbf{s}(k) = 1$. From this, we can conclude that if the inequality defining the set $\Theta_1$ in Equation (37) holds, the inequality in Equation (38) holds as well. Therefore, we can define the second auxiliary set as:

$$\Theta_2 = \left\{ \mathbf{s} \mid \mathbf{s}^\top \cdot \widehat{\mathbf{P}}_{\sigma(k)} \cdot \mathbf{s} \le \frac{1 - 2 \cdot \chi + \frac{\chi}{\gamma_1}}{1 - \chi \cdot \gamma_1} \cdot \frac{1}{\mu} \right\}, \tag{39}$$

and it satisfies

$$\Theta_1 \subseteq \Theta_2. \tag{40}$$

Moving forward, we note that the condition in Equation (28) is equivalent to $0 < \frac{1 - 2\chi + \frac{\chi}{\gamma_1}}{1 - \chi \cdot \gamma_1} \cdot \frac{1}{\mu} < 1$. Therefore, we can define the third auxiliary set as

$$\Theta_3 = \left\{ \mathbf{s} \mid \mathbf{s}^\top \cdot \widehat{\mathbf{P}}_{\sigma(k)} \cdot \mathbf{s} \le 1, \ \widehat{\mathbf{P}}_{\sigma(k)} \succ 0 \right\}, \tag{41}$$

and referring to Equation (39), it satisfies

$$\Theta_2 \subseteq \Theta_3. \tag{42}$$

At the moment, we can draw conclusions from Equations (36), (40) and (43):

$$\Psi_{\sigma(k)} \subseteq \Theta_1 \subseteq \Theta_2 \subseteq \Theta_3. \tag{43}$$

Observing Equations (17) and (41), we have $\Theta_3 = \Omega_{\sigma(k)}$. Then, applying Lemma C.2 in Appendix C, we have $\Theta_3 = \Omega_{\sigma(k)} \subseteq \mathbb{X}$, which, in light of Equation (43), results in $\Psi_{\sigma(k)} \subseteq \mathbb{X}$. We thus complete the proof of the statement in Item 1.

### D.2.2 PROOF OF STATEMENT IN ITEM 2

We define a Lyapunov candidate for the tracking-error dynamics described in Equation (15) as:

$$V(t) = \mathbf{e}^\top(t) \cdot \widehat{\mathbf{P}}_{\sigma(t)} \cdot \mathbf{e}(t), \tag{44}$$

which, combined with the dynamics in Equation (15) and the action policy in Equation (12), leads to

$$V(t+1) - \alpha \cdot V(t)$$

$$= \mathbf{e}^\top(t+1) \cdot \widehat{\mathbf{P}}_{\sigma(t)} \cdot \mathbf{e}(t+1) - \alpha \cdot \mathbf{e}^\top(t) \cdot \widehat{\mathbf{P}}_{\sigma(t)} \cdot \mathbf{e}(t)$$

$$= \mathbf{e}^\top(t) \cdot \left( \overline{\mathbf{A}}^\top(\widehat{\mathbf{s}}_{\sigma(t)}) \cdot \widehat{\mathbf{P}}_{\sigma(t)} \cdot \overline{\mathbf{A}}(\widehat{\mathbf{s}}_{\sigma(t)}) - \alpha \cdot \widehat{\mathbf{P}}_{\sigma(t)} \right) \cdot \mathbf{e}(t) + \mathbf{h}^\top(\mathbf{e}(t)) \cdot \widehat{\mathbf{P}}_{\sigma(t)} \cdot \mathbf{h}(\mathbf{e}(t))$$

$$+ 2 \cdot \mathbf{e}^\top(t) \cdot \left( \overline{\mathbf{A}}(\widehat{\mathbf{s}}_{\sigma(t)}) \cdot \widehat{\mathbf{P}}_{\sigma(t)} \right) \cdot \mathbf{h}(\mathbf{e}(t)), \tag{45}$$

where we define:

$$\overline{\mathbf{A}}(\widehat{\mathbf{s}}_{\sigma(t)}) \overset{\triangle}{=} \mathbf{A}(\widehat{\mathbf{s}}_{\sigma(t)}) + \mathbf{B}(\widehat{\mathbf{s}}_{\sigma(t)}) \cdot \widehat{\mathbf{F}}_{\sigma(t)}. \tag{46}$$

After applying Lemma C.4 in Appendix C, we have:

$$2\mathbf{e}^\top(t) \cdot \left( \overline{\mathbf{A}}(\widehat{\mathbf{s}}_{\sigma(t)}) \cdot \widehat{\mathbf{P}}_{\sigma(t)} \right) \cdot \mathbf{h}(\mathbf{e}(t))$$

$$\leq \gamma_2 \cdot \mathbf{e}^\top(t) \cdot \overline{\mathbf{A}}^\top(\widehat{\mathbf{s}}_{\sigma(t)}) \cdot \widehat{\mathbf{P}}_{\sigma(t)} \cdot \overline{\mathbf{A}}(\widehat{\mathbf{s}}_{\sigma(t)}) \cdot \mathbf{e}(t) + \frac{1}{\gamma_2} \cdot \mathbf{h}^\top(\mathbf{e}(t)) \cdot \widehat{\mathbf{P}}_{\sigma(t)} \cdot \mathbf{h}(\mathbf{e}(t)), \tag{47}$$

where $\gamma_2 > 0$.

We note that Assumption 6.3 implies:

$$\mathbf{h}^\top(\mathbf{e}(t)) \cdot \widehat{\mathbf{P}}_{\sigma(t)} \cdot \mathbf{h}(\mathbf{e}(t)) \leq \kappa \cdot \mathbf{e}^\top(t) \cdot \widehat{\mathbf{P}}_{\sigma(t)} \cdot \mathbf{e}(t). \tag{48}$$

Substituting inequalities in Equations (47) and (48) into Equation (45) yields:

$$V(t+1) - \alpha \cdot V(t)$$

$$\leq \mathbf{e}^\top(t) \cdot \left( (1+\gamma_2) \cdot \overline{\mathbf{A}}^\top(\widehat{\mathbf{s}}_{\sigma(t)}) \cdot \widehat{\mathbf{P}}_{\sigma(t)} \cdot \overline{\mathbf{A}}(\widehat{\mathbf{s}}_{\sigma(t)}) - (\alpha - \kappa \cdot (1 + \frac{1}{\gamma_2})) \cdot \widehat{\mathbf{P}}_{\sigma(t)} \right) \cdot \mathbf{e}(t). \tag{49}$$

Recalling the Schur Complement in Lemma C.1 of Appendix C and considering $\widehat{\mathbf{P}}_{\sigma(t)} \succ 0$, we conclude that the inequality in Equation (30) is equivalent to

$$(\alpha - \kappa \cdot (1 + \frac{1}{\gamma_2})) \cdot \widehat{\mathbf{Q}}_{\sigma(t)} - (1 + \gamma_2) \cdot (\widehat{\mathbf{Q}}_{\sigma(t)} \cdot \mathbf{A}^\top(\widehat{\mathbf{s}}_{\sigma(t)}) + \widehat{\mathbf{R}}_{\sigma(t)}^\top \cdot \mathbf{B}^\top(\widehat{\mathbf{s}}_{\sigma(t)}) \cdot \widehat{\mathbf{Q}}_{\sigma(t)}^{-1}$$

$$\cdot (\mathbf{A}(\widehat{\mathbf{s}}_{\sigma(t)}) \cdot \widehat{\mathbf{Q}}_{\sigma(t)} + \mathbf{B}(\widehat{\mathbf{s}}_{\sigma(t)}) \cdot \widehat{\mathbf{R}}_{\sigma(t)}) \succ 0,$$

multiplying both the left-hand side and the right-hand side of which by $\widehat{\mathbf{Q}}^{-1}$ yields:

$$(\alpha - \kappa \cdot (1 + \frac{1}{\gamma_2})) \cdot \widehat{\mathbf{Q}}_{\sigma(t)}^{-1} - (1 + \gamma_2) \cdot (\mathbf{A}^\top(\widehat{\mathbf{s}}_{\sigma(t)}) + \widehat{\mathbf{Q}}_{\sigma(t)}^{-1} \cdot \widehat{\mathbf{R}}_{\sigma(t)}^\top \cdot \mathbf{B}^\top(\widehat{\mathbf{s}}_{\sigma(t)})) \cdot \widehat{\mathbf{Q}}_{\sigma(t)}^{-1}$$

$$\cdot (\mathbf{A}(\widehat{\mathbf{s}}_{\sigma(k)}) + \mathbf{B}(\widehat{\mathbf{s}}_{\sigma(t)}) \cdot \widehat{\mathbf{R}}_{\sigma(t)} \cdot \mathbf{Q}_{\sigma(t)}^{-1}) \succ 0,$$

Substituting the definitions in Equation (16) into which, we arrive at

$$(\alpha - \kappa \cdot (1 + \frac{1}{\gamma_2})) \cdot \widehat{\mathbf{P}}_{\sigma(t)} - (1 + \gamma_2) \cdot (\mathbf{A}^\top(\widehat{\mathbf{s}}_{\sigma(t)}) + \widehat{\mathbf{F}}_{\sigma(t)}^\top \cdot \mathbf{B}^\top(\widehat{\mathbf{s}}_{\sigma(t)}) \cdot \widehat{\mathbf{P}}_{\sigma(t)}$$

$$\cdot (\mathbf{A}(\widehat{\mathbf{s}}_{\sigma(t)}) + \mathbf{B}(\widehat{\mathbf{s}}_{\sigma(t)}) \cdot \widehat{\mathbf{F}}_{\sigma(t)}) \succ 0. \tag{50}$$

Recalling Equation (46), the inequality in Equation (50) is equivalent to the following:

$$(1 + \gamma_2) \cdot \overline{\mathbf{A}}^\top(\widehat{\mathbf{s}}_{\sigma(t)}) \cdot \widehat{\mathbf{P}}_{\sigma(t)} \cdot \overline{\mathbf{A}}(\widehat{\mathbf{s}}_{\sigma(t)}) - (\alpha - \kappa \cdot (1 + \frac{1}{\gamma_2})) \cdot \widehat{\mathbf{P}}_{\sigma(t)} \prec 0,$$

which, in conjunction with Equation (49), leads to $V(t+1) - \alpha \cdot V(t) \leq 0$, i.e., $\mathbf{e}^\top(t+1) \cdot \widehat{\mathbf{P}}_{\sigma(t)} \cdot \mathbf{e}(t+1) \leq \alpha \cdot \mathbf{e}^\top(t) \cdot \widehat{\mathbf{P}}_{\sigma(t)} \cdot \mathbf{e}(t)$, we thus complete the proof of the statement in Item 2.

### D.2.3 PROOF OF STATEMENT IN ITEM 3

With the consideration of $\mathbf{T}^{-1} = \mathbf{V}$, according to Lemma C.1, the condition in Equation (29) implies:

$$\widehat{\mathbf{Q}}_{\sigma(t)} - \widehat{\mathbf{R}}_{\sigma(t)}^\top \cdot \mathbf{T}^{-1} \cdot \widehat{\mathbf{R}}_{\sigma(t)} = \widehat{\mathbf{Q}}_{\sigma(t)} - \widehat{\mathbf{R}}_{\sigma(t)}^\top \cdot \mathbf{V} \cdot \widehat{\mathbf{R}}_{\sigma(t)} \succ 0. \tag{51}$$

Substituting $\widehat{\mathbf{F}}_{\sigma(t)} \cdot \widehat{\mathbf{Q}}_{\sigma(t)} = \widehat{\mathbf{R}}_{\sigma(t)}$ into Equation (51) leads to

$$\widehat{\mathbf{Q}}_{\sigma(t)} - (\widehat{\mathbf{F}}_{\sigma(t)} \cdot \widehat{\mathbf{Q}}_{\sigma(t)})^\top \cdot \mathbf{V} \cdot (\widehat{\mathbf{F}}_{\sigma(t)} \cdot \widehat{\mathbf{Q}}_{\sigma(t)}) \succ 0. \tag{52}$$

multiplying both left-hand and right-hand sides of which by $\widehat{\mathbf{Q}}_{\sigma(k)}^{-1}$ yields:

$$\widehat{\mathbf{Q}}_{\sigma(t)}^{-1} - \widehat{\mathbf{F}}_{\sigma(t)}^{\top} \cdot \mathbf{V} \cdot \widehat{\mathbf{Q}}_{\sigma(t)} \succ 0,$$

from which we thus have

$$\mathbf{e}^{\top}(t) \cdot \widehat{\mathbf{Q}}_{\sigma(t)}^{-1} \cdot \mathbf{e}(t) - \mathbf{e}^{\top}(t) \cdot \widehat{\mathbf{F}}_{\sigma(t)}^{\top} \cdot \mathbf{V} \cdot \widehat{\mathbf{F}}_{\sigma(t)} \cdot \mathbf{e}(t)$$

$$= \mathbf{e}^{\top}(t) \cdot \widehat{\mathbf{P}}_{\sigma(t)} \cdot \mathbf{e}(t) - \mathbf{a}_{\text{HA}}^{\top}(t) \cdot \mathbf{V} \cdot \mathbf{a}_{\text{HA}}(t) > 0, \tag{53}$$

which is obtained via considering $\widehat{\mathbf{P}}_{\sigma(t)} = \widehat{\mathbf{Q}}_{\sigma(t)}^{-1}$, and Equation (12) with $\mathbf{e}(t) = \mathbf{s}(t) - \chi \cdot \widehat{\mathbf{s}}_{\sigma(t)}$.

We let $\mathbf{e} = \mathbf{s} - \chi \cdot \widehat{\mathbf{s}}_{\sigma(k)}$. The patch definition in Equation (11) can re-expressed as

$$\Psi_{\sigma(k)} \triangleq \left\{ \mathbf{e} \mid \mathbf{e}^{\top} \cdot \widehat{\mathbf{P}}_{\sigma(k)} \cdot \mathbf{e} \leq (1 - \chi)^2 \cdot \mathbf{s}^{\top}(k) \cdot \widehat{\mathbf{P}}_{\sigma(k)} \cdot \mathbf{s}(k), \right.$$

$$\left. \text{with } \mathbf{s}(k) \text{ subject to Equation (7), and } \widehat{\mathbf{P}}_{\sigma(k)} \succ 0 \right\}. \tag{54}$$

The inequality in Equation (53) can be expressed as $\mathbf{e}^{\top}(t) \cdot \widehat{\mathbf{P}}_{\sigma(t)} \cdot \mathbf{e}(t) > \mathbf{a}_{\text{HA}}^{\top}(t) \cdot \mathbf{V} \cdot \mathbf{a}_{\text{HA}}(t)$. Based on Equation (41), we can conclude that if $\mathbf{e}(t) \in \Theta_3$, meaning it satisfies $\mathbf{e}^{\top}(t) \cdot \widehat{\mathbf{P}}_{\sigma(t)} \cdot \mathbf{e}(t) < 1$, then $\mathbf{a}_{\text{HA}}^{\top}(t) \cdot \mathbf{V} \cdot \mathbf{a}_{\text{HA}}(t) < 1$. Additionally, considering Equation (43) and Equation (54), if $\mathbf{e}(t) \in \Psi_{\sigma(k)}$, then $\mathbf{a}_{\text{HA}}^{\top}(t) \cdot \mathbf{V} \cdot \mathbf{a}_{\text{HA}}(t) < 1$. It's important to note that $t \in \{k, \dots, k+\tau\} = \mathbb{T}_{\sigma(k)}$, and $k$ represents the triggering time of HA-Teacher.

Upon verification from Equation (13), it becomes evident that $\mathbf{e}(k) = \mathbf{s}(k) - \chi \cdot \widehat{\mathbf{s}}_{\sigma(k)} = \mathbf{s}(k) - \chi \cdot \widehat{\mathbf{s}}(k)$, where $\mathbf{e}(k)$ lies on the boundary of the patch: $\mathbf{e}(k)^{\top} \cdot \widehat{\mathbf{P}}_{\sigma(k)} \cdot \mathbf{e}(k) = (1 - \chi)^2 \cdot \mathbf{s}^{\top}(k) \cdot \widehat{\mathbf{P}}_{\sigma(k)} \cdot \mathbf{s}(k)$. Furthermore, as per the second statement in Item 2, i.e., $\mathbf{e}^{\top}(t+1) \cdot \widehat{\mathbf{P}}_{\sigma(t)} \cdot \mathbf{e}(t+1) \leq \alpha \cdot \mathbf{e}^{\top}(t) \cdot \widehat{\mathbf{P}}_{\sigma(t)} \cdot \mathbf{e}(t)$ for time $t \in \mathbb{T}_{\sigma(k)}$, we can infer that $\mathbf{e}(t)$ never exits the patch during the active time of HA-Teacher initiated at time $k$. Hence, we can conclude that $\mathbf{a}_{\text{HA}}^{\top}(t) \cdot \mathbf{V} \cdot \mathbf{a}_{\text{HA}}(t) < 1$ holds for any time $t \in \mathbb{T}_{\sigma(k)}$.

Finally, taking into account Equation (23) and Lemma C.3 in Appendix C, we can establish that $\mathbf{a}_{\text{HA}}(t) \in \mathbb{A}$ for any time $t \in \mathbb{T}_{\sigma(k)}$, thus completing the proof.

# E   GUIDANCE FOR CORRECTION HORIZON AND DWELL TIME

Upon reviewing Figure 1 and Equations (8) to (10), we can conclude that $\tau$ serves as both the correction horizon for the unsafe actions of HP-Student and the dwell time for HA-Teacher. The value of $\tau$ significantly influences HP-Student's runtime learning in achieving a high-performance policy. If $\tau$ is small, the patch center fails to attract the system states to the envelope inside, resulting in HA-Teacher dominating the learning process, solely ensuring safety. Conversely, if $\tau$ is very large, HP-Student is unable to effectively and swiftly self-learn to achieve his goal. Thus, these considerations should guide the selection of $\tau$. This guidance is based on the results from Theorem 6.4, as presented in the following corollary.

**Corollary E.1.** *If the correction horizon and the dwell time of HA-Teacher, denoted as $\tau$, satisfy:*

$$\tau = \left\lceil \frac{\ln(\delta \cdot \mu) - \ln(\mathbf{e}^\top (k) \cdot \mathbf{P} \cdot \mathbf{e}(k))}{\ln \alpha} \right\rceil, \tag{55}$$

*we have $\mathbf{e}^\top (k + \tau) \cdot \widehat{\mathbf{P}}_{\sigma(k)} \cdot \mathbf{e}(k + \tau) \leq \delta$, where $k$ denotes the triggering time of HA-Teacher.*

*Proof.* We obtain from Item 2 that

$$\mathbf{e}^\top (t) \cdot \widehat{\mathbf{P}}_{\sigma(t)} \cdot \mathbf{e}(t) \leq \alpha^{t-k} \cdot \mathbf{e}^\top (k) \cdot \widehat{\mathbf{P}}_{\sigma(k)} \cdot \mathbf{e}(k), \ \ t \in \mathbb{T}_{\sigma(k)}. \tag{56}$$

Considering $0 < \alpha < 1$, we can verify from Equation (56) that $\alpha^{t-k} \cdot \mathbf{e}^\top (k) \cdot \widehat{\mathbf{P}}_{\sigma(k)} \cdot \mathbf{e}(k) \leq \delta$ is equivalent to

$$\tau = t - k \geq \frac{\ln \delta - \ln(\mathbf{e}^\top (k) \cdot \widehat{\mathbf{P}}_{\sigma(k)} \cdot \mathbf{e}(k))}{\ln \alpha}. \tag{57}$$

In addition, considering $\widehat{\mathbf{P}}_{\sigma(k)} = \widehat{\mathbf{Q}}_{\sigma(k)}^{-1}$ and $\mu > 0$, the condition in Equation (27) used for designing real-time patch in Theorem 6.4 is equivalent to

$$\frac{1}{\mu} \cdot \mathbf{P} \succ \widehat{\mathbf{P}}_{\sigma(k)},$$

which, in conjunction with $0 < \alpha < 1$, leads to

$$\begin{aligned}
\frac{\ln \delta - \ln(\mathbf{e}^\top (k) \cdot \widehat{\mathbf{P}}_{\sigma(k)} \cdot \mathbf{e}(k))}{\ln \alpha} &\leq \frac{\ln \delta - \ln(\frac{1}{\mu} \cdot \mathbf{e}^\top (k) \cdot \mathbf{P} \cdot \mathbf{e}(k))}{\ln \alpha} \\
&= \frac{\ln(\delta \cdot \mu) - \ln(\mathbf{e}^\top (k) \cdot \mathbf{P} \cdot \mathbf{e}(k))}{\ln \alpha} \\
&\leq \left\lceil \frac{\ln(\delta \cdot \mu) - \ln(\mathbf{e}^\top (k) \cdot \mathbf{P} \cdot \mathbf{e}(k))}{\ln \alpha} \right\rceil.
\end{aligned} \tag{58}$$

Based on Equation (58), we can conclude that if the condition in Equation (55) is satisfied, then the inequality in Equation (57) also holds. Consequently, we have $\alpha^{t-k} \cdot \mathbf{e}^\top (k) \cdot \widehat{\mathbf{P}}_{\sigma(k)} \cdot \mathbf{e}(k) \leq \delta$. This, together with Equation (56), implies $\mathbf{e}^\top (t) \cdot \widehat{\mathbf{P}}_{\sigma(t)} \cdot \mathbf{e}(t) \leq \delta$. Furthermore, if we consider $\sigma(k)$ as a piece-wise signal (i.e., $\sigma(m) = \sigma(k)$ for $m \in \mathbb{T}_{\sigma(k)} = \{k, k+1, \ldots, k+\tau\}$) and $\tau = t - k$, then $\mathbf{e}^\top (t) \cdot \widehat{\mathbf{P}}_{\sigma(t)} \cdot \mathbf{e}(t) \leq \delta$ can be rewritten as $\mathbf{e}^\top (k + \tau) \cdot \widehat{\mathbf{P}}_{\sigma(k)} \cdot \mathbf{e}(k + \tau) \leq \delta$. This completes the proof. □

The real-time tracking error $\mathbf{e}(t)$ represents the distance to the patch center $\chi \cdot \widehat{\mathbf{s}}_{\sigma(k)}$. Therefore, $\mathbf{e}^\top (t) \cdot \widehat{\mathbf{P}}_{\sigma(t)} \cdot \mathbf{e}(t)$ serves as a measurement metric for proximity to the patch center. Additionally, $\mathbf{e}^\top (k + \tau) \cdot \widehat{\mathbf{P}}_{\sigma(k)} \cdot \mathbf{e}(k + \tau) \leq \delta$ can be interpreted as a safety criterion for returning to HP-Student. Consequently, Corollary E.1 implies that $\tau$ is computed to ensure that the real plant, under the control of HA-Teacher, satisfies the preset safety criteria rather than infinitely approaching the patch center.

# F  UNKNOWN UNKNOWN: RANDOMIZED BETA DISTRIBUTION

In the real world, plants can encounter a multitude of unknown variables, each with unique characteristics. To tackle this challenge, we propose utilizing a variant of the Beta distribution Johnson et al. (1995) to effectively model one type of these unknowns. This approach holds promise in mathematically defining and addressing these uncertainties.

**Definition F.1** (Randomized Beta Distribution). The disturbance, noise, or fault, denoted by $\mathbf{d}(k)$, is considered to be a bounded unknown if (i) $\mathbf{d}(k) \sim Beta(\alpha(k), \beta(k), c, a)$, and (ii) $\alpha(k)$ and $\beta(k)$ are random parameters. In other words, the disturbance $\mathbf{d}(k)$ is within the range of [a, c], and its probability density function (pdf) is given by

$$f\left(\mathbf{d}(k); \ \alpha(k), \ \beta(k), \ a, \ c\right) = \frac{(\mathbf{d}(k) - a)^{\alpha-1}(c - \mathbf{d}(k))^{\alpha(k)-1}\Gamma\left(\alpha(k) + \beta(k)\right)}{(c - a)^{\alpha(k)+\beta(k)-1}\Gamma\left(\alpha(k)\right)\Gamma\left(\beta(k)\right)}, \quad (59)$$

where $\Gamma\left(\alpha(k)\right) = \int_0^\infty t^{\alpha(k)-1}e^{-t}dt$, $\mathrm{Re}\left(\alpha(k)\right) > 0$, $\alpha(k)$ and $\beta(k)$ are randomly given at every $k$.

The randomized Beta distribution defined in Definition F.1 is crucial for describing a certain type of unknown unknown. This is due to two critical reasons. First, the characteristics of unknown unknowns involve minimal historical data and unpredictable time and distributions. This leads to unavailable models for scientific discoveries and understanding.

In the example shown in Figure 8, the parameters $\alpha$ and $\beta$ directly influence the probability density function (pdf) of the distribution, and consequently, the mean and variance. Suppose $\alpha$ and $\beta$ are randomized (expressed as $\alpha(k)$ and $\beta(k)$). In that case, the distribution of $\mathbf{d}(k)$ can take the form of a uniform distribution, exponential distribution, truncated Gaussian distribution, or a combination of these. However, the specific distribution is unknown. Therefore, the randomized $\alpha(k)$ and $\beta(k)$, which result in a randomized Beta distribution, can effectively capture the characteristics of "unavailable model" and "unforeseen" traits associated with unknown unknowns in both time and distribution. Furthermore, the randomized Beta distributions are bounded, with the bounds denoted as $a$ and $c$. This is motivated by the fact that, in general, there are no probabilistic solutions for handling unbounded unknowns, such as earthquakes and volcanic eruptions.

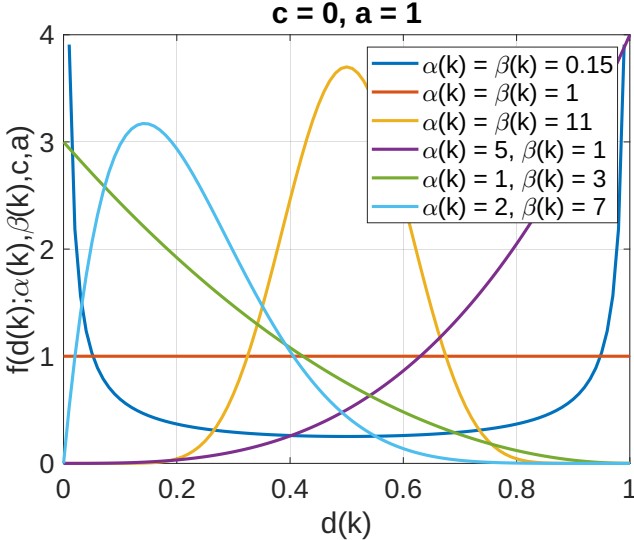

Figure 8: $\alpha(k)$ and $\beta(k)$ control the robability density the function of the distribution.

# G  EXPERIMENT: CART-POLE SYSTEM

## G.1  CONFIGURATIONS OF PRE-TRAINING AND RUNTIME LEARNING

The pre-training configurations for HP-Student, other DRL agents, and the runtime learning mimicking real plants are all the same to ensure fair comparisons. Specifically, we utilize the DDPG algorithm Lillicrap et al. (2016) to pre-train DRL and Phy-DRL models and to support runtime learning. The actor and critic networks are implemented as Multi-Layer Perceptrons (MLPs) with four fully connected layers. The output dimensions of the critic and actor networks are 256, 128, 64, and 1, respectively. The activation functions of the first three neural layers are ReLU, while the output of the last layer is the Tanh function for the actor network and Linear for the critic network. The input of the critic network is $[\mathbf{s}; \mathbf{a}]$, while the input of the actor network is $\mathbf{s}$. In more detail, we set the discount factor $\gamma = 0.9$ and the learning rates of the critic and actor networks to be the same at 0.0003. We set the batch size to 200. The episode consists of 1000 steps, and the sampling frequency is 30 Hz.

## G.2  SYSTEM DYNAMICS

The physics knowledge about the dynamics of cart-pole systems used by HP-Student and HA-Teacher for their designs is from the following dynamics model in Florian (2005):

$$\ddot{\theta} = \frac{g \sin\theta + \cos\theta \left( \frac{-F - m_p l \dot{\theta}^2 \sin\theta}{m_c + m_p} \right)}{l \left( \frac{4}{3} - \frac{m_p \cos^2\theta}{m_c + m_p} \right)}, \tag{60a}$$

$$\ddot{x} = \frac{F + m_p l \left( \dot{\theta}^2 \sin\theta - \ddot{\theta} \cos\theta \right)}{m_c + m_p}, \tag{60b}$$

whose $g$ represents the gravitational acceleration, with $m_c = 0.94$ kg, $m_p = 0.29$ kg, $l = 0.32$ m, and $F$ as the actuator input.

## G.3  HP-STUDENT DESIGN

As Phy-DRL allows us to simplify the nonlinear dynamics model in Equation (60) to an analyzable linear model:

$$\dot{\mathbf{s}} = \widehat{\mathbf{A}} \cdot \mathbf{s} + \widehat{\mathbf{B}} \cdot \mathbf{a}, \tag{61}$$

where $\mathbf{s} = [x, v, \theta, \omega]^\top$. To obtain $\widehat{\mathbf{A}}$ and $\widehat{\mathbf{B}}$ from equation Equation (60), we approximate $\cos\theta$ as 1, $\sin\theta$ as $\theta$, and $\omega^2 \sin\theta$ as 0. Additionally, the sampling technique converts the continuous-time model in Equation (61) to a discrete-time model:

$$\mathbf{s}(k+1) = \mathbf{A} \cdot \mathbf{s}(k) + \mathbf{B} \cdot \mathbf{a}(k), \text{ with } \mathbf{A} = \mathbf{I}_4 + T \cdot \widehat{\mathbf{A}}, \ \mathbf{B} = T \cdot \widehat{\mathbf{A}},$$

where we have

$$\mathbf{A} = \begin{bmatrix} 1 & 0.0333 & 0 & 0 \\ 0 & 1 & -0.0565 & 0 \\ 0 & 0 & 1 & 0.0333 \\ 0 & 0 & 0.8980 & 1 \end{bmatrix}, \quad \mathbf{B} = [0 \ \ 0.0334 \ \ 0 \ \ -0.0783]^\top. \tag{62}$$

Considering the safety set

$$\overline{\mathbb{X}} = \left\{ \mathbf{s} \in \mathbb{R}^4 \big| -0.8 \le x \le 0.8, -0.7 < \theta < 0.7, -4 \le \dot{x} \le 4, -4 \le \dot{\theta} \le 4 \right\},$$

the model knowledge $(\mathbf{A}, \mathbf{B})$ in Equation (62), and according to the design of Phy-DRL in Cao et al. (2024), we have

$$\mathbf{P} = \begin{bmatrix} 54.1134178606985 & 26.2600592637275 & 61.7975412804215 & 12.9959418258126 \\ 26.2600592637275 & 14.3613985149923 & 34.6710819094179 & 7.27321583818861 \\ 61.7975412804215 & 34.6710819094179 & 88.7394386456256 & 18.0856894519164 \\ 12.9959418258126 & 7.27321583818861 & 18.0856894519164 & 3.83961074325448 \end{bmatrix},$$

$$\mathbf{F} = \begin{bmatrix} 46.1347017672011 & 31.4100347880721 & 106.033772085368 & 19.9606055711095 \end{bmatrix},$$

With which, and letting $w(\mathbf{s}(k), \mathbf{a}_{\text{HP}}(k)) = -\mathbf{a}_{\text{drl}}^2(k)$, the residual action policy in Equation (4) and the safety-embedded reward in Equation (5) are then ready for HP-Student, i.e., Phy-DRL agent.

### G.4 HA-TEACHER DESIGN

#### G.4.1 MODEL KNOWLEDGE

Compared to HP-Student, HA-Teacher possesses a more comprehensive understanding of system dynamics in physics, directly and equivalently derived from Equation (60) as

$$
\frac{d}{dt}\underbrace{\begin{bmatrix} x \\ \dot{x} \\ \theta \\ \dot{\theta} \end{bmatrix}}_{\mathbf{s}} = \underbrace{\begin{bmatrix} 0 & 1 & 0 & 0 \\ 0 & 0 & \frac{-m_p g \sin\theta\cos\theta}{\theta[\frac{4}{3}(m_c+m_p)-m_p\cos^2\theta]} & \frac{\frac{4}{3}m_p l \sin\theta\dot\theta}{\frac{4}{3}(m_c+m_p)-m_p\cos^2\theta} \\ 0 & 0 & 0 & 1 \\ 0 & 0 & \frac{g\sin\theta(m_c+m_p)}{l\theta[\frac{4}{3}(m_c+m_p)-m_p\cos^2\theta]} & \frac{-m_p\sin\theta\cos\theta\dot\theta}{\frac{4}{3}(m_c+m_p)-m_p\cos^2\theta} \end{bmatrix}}_{\widehat{\mathbf{A}}(\mathbf{s})} \cdot \begin{bmatrix} x \\ \dot{x} \\ \theta \\ \dot{\theta} \end{bmatrix}
$$

$$
+ \underbrace{\begin{bmatrix} 0 \\ \frac{\frac{4}{3}}{\frac{4}{3}(m_c+m_p)-m_p\cos^2\theta} \\ 0 \\ \frac{-\cos\theta}{l[\frac{4}{3}(m_c+m_p)-m_p\cos^2\theta]} \end{bmatrix}}_{\widehat{\mathbf{B}}(\mathbf{s})} \cdot \underbrace{F}_{\mathbf{a}}, \quad (63)
$$

where $\widehat{\mathbf{A}}(\mathbf{s})$ and $\widehat{\mathbf{B}}(\mathbf{s})$ are known to the HA-Teacher. The sampling technique transforms the continuous-time dynamics model equation 68 to the discrete-time one:

$$
\mathbf{s}(k+1) = (\mathbf{I}_4 + T \cdot \widehat{\mathbf{A}}(\mathbf{s})) \cdot \mathbf{s}(k) + T \cdot \widehat{\mathbf{B}}(\mathbf{s}) \cdot \mathbf{a}(k),
$$

from which we obtain the model knowledge $\mathbf{A}(\widehat{\mathbf{s}}_{\sigma(k)})$ and $\mathbf{B}(\widehat{\mathbf{s}}_{\sigma(k)})$ in Equation (15) as

$$
\mathbf{A}(\widehat{\mathbf{s}}_{\sigma(k)}) = \mathbf{I}_4 + T \cdot \widehat{\mathbf{A}}(\widehat{\mathbf{s}}_{\sigma(k)}), \qquad \mathbf{B}(\overline{\mathbf{s}}^*) = T \cdot \widehat{\mathbf{B}}(\widehat{\mathbf{s}}_{\sigma(k)}), \quad (64)
$$

where $T = \frac{1}{30}$ second, i.e., the sampling frequency is 30 Hz.

#### G.4.2 PARAMETERS FOR REAL-TIME PATCH COMPUTING

Right now, we have $\mathbf{A}(\widehat{\mathbf{s}}_{\sigma(k)})$ and $\mathbf{B}(\widehat{\mathbf{s}}_{\sigma(k)})$ in Equation (77). To satisfy Assumption 6.3, we let $\kappa = 0.01$. For inequalities in Equations (27) to (30), we set $\alpha = 0.999$, $\chi = 0.3$, $\gamma_1 = 1$, $\gamma_2 = 0.1$, and $\tau = 10$. Finally, according to the given safety set $\mathbb{X} = \{\mathbf{s} \in | |x| \le 1, |\theta| < 0.8\}$ and the action space of HA-Teacher $\mathbb{A} = \{\mathbf{a}_{HA} \in \mathbb{R}| |\mathbf{a}_{HA}| \le 20\}$, we obtain following knowledge for the inequalities in Equations (18) and (23):

We currently have $\mathbf{A}(\widehat{\mathbf{s}}_{\sigma(k)})$ and $\mathbf{B}(\widehat{\mathbf{s}}_{\sigma(k)})$ in Equation (77). To satisfy Assumption 6.3, we set $\kappa$ to be 0.01. For the inequalities in Equations (27) to (30), we assign the values $\alpha = 0.99$, $\chi = 0.3$, $\gamma_1 = 1$, and $\gamma_2 = 0.1$. Finally, based on the given safety set $\mathbb{X} = \{\mathbf{s} \in \mathbb{R}^n | |x| \le 1, |\theta| < 0.8\}$ and the action space of HA-Teacher $\mathbb{A} = \{\mathbf{a}_{HA} \in \mathbb{R} | |\mathbf{a}_{HA}| \le 40\}$, we obtain the following information for the inequalities in Equations (18) and (23):

$$
\mathbf{D} = \begin{bmatrix} 1 & 0 & 0 & 0 \\ 0 & 0 & 1/0.8 & 0 \end{bmatrix}, \quad \mathbf{C} = 1/40.
$$

### G.5 ADDITIONAL EXPERIMENTS

The section presents additional experimental results to bolster the claims of our proposed runtime learning machine.

#### G.5.1 RUNTIME LEARNING MACHINE V.S. FAULT-TOLERANT DRL AND SAFE DRL

We have included additional phase plots to illustrate further the key feature of our runtime learning machine: ensuring lifetime safety, meaning safety assurance at any stage of the learning process. To achieve this, we conducted 5 learning episodes where we observed the system behavior under the control of the runtime learning machine at episodes 5, 10, 15, and 20. Additionally, for each episode, we tested the models using three random initial conditions. The phase plots of position

and angle, including those of the four compared models, can be found in Figures 9 to 12. Upon observing Figures 9 to 12, we conclude that our runtime learning machine can consistently guarantee the safety of real plants across all sampled learning episodes, in the presence of unknown factors and the Sim2Real gap. This level of safety cannot be achieved by the current state-of-the-art safe DRL and fault-tolerant DRL methods.

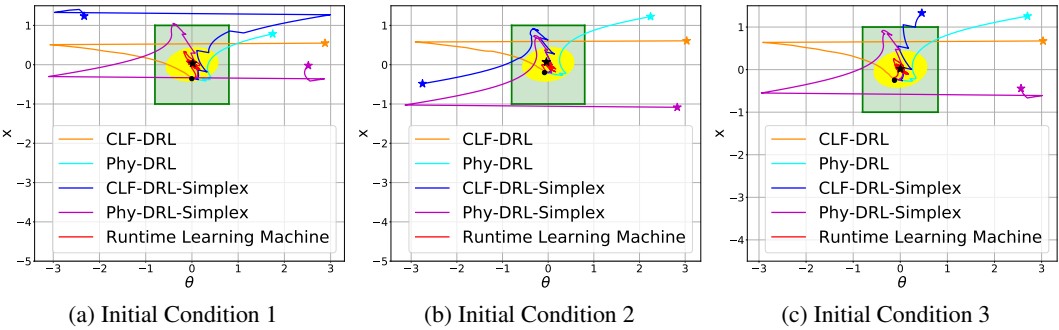

(a) Initial Condition 1        (b) Initial Condition 2        (c) Initial Condition 3

Figure 9: **Episode 5**. Phase plots, given the same initial condition. The black dot and star denote the initial condition and final location, respectively.

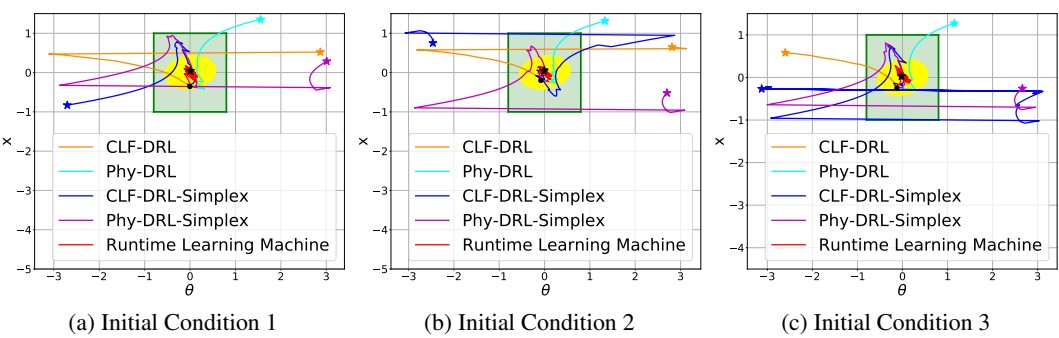

(a) Initial Condition 1        (b) Initial Condition 2        (c) Initial Condition 3

Figure 10: **Episode 10.** Phase plots, given the same initial condition. The black dot and star denote the initial condition and final location, respectively.

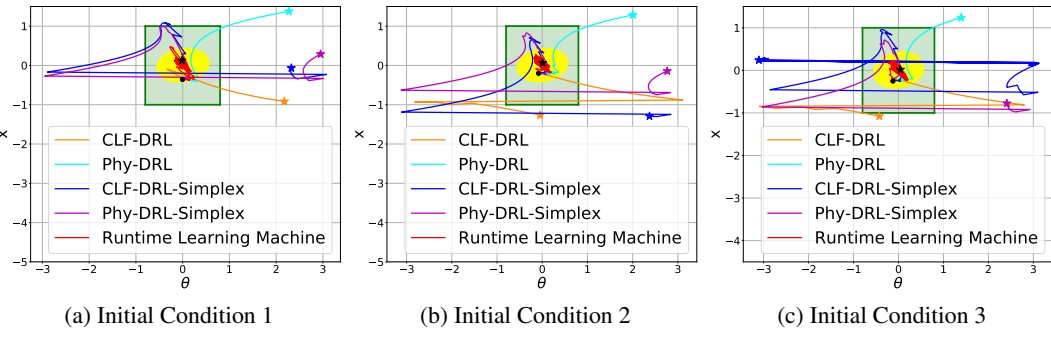

(a) Initial Condition 1        (b) Initial Condition 2        (c) Initial Condition 3

Figure 11: **Episode 15**. Phase plots, given the same initial condition. The black dot and star denote the initial condition and final location, respectively.

### G.5.2 AUTOMATIC HIERARCHY LEARNING MECHANISM

The trajectories of system states under the control of the HP-Student are shown in Figure 13 for ten random initial conditions. The HP-Student engages in runtime learning. After reviewing Figure 13,

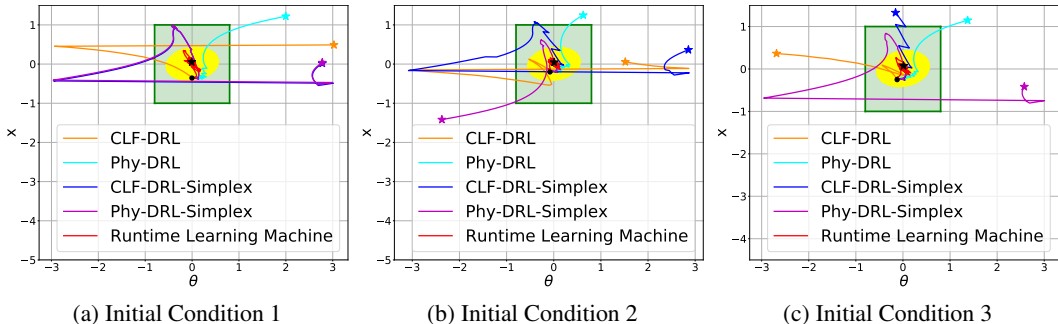

(a) Initial Condition 1      (b) Initial Condition 2      (c) Initial Condition 3

Figure 12: **Episode 20**. Phase plots, given the same initial condition. The black dot and star denote the initial condition and final location, respectively.

we can conclude that the action policy of the HP-Student in episode 20 demonstrates higher mission performance compared to episode 5. It is much closer to the mission goal and remains stable.

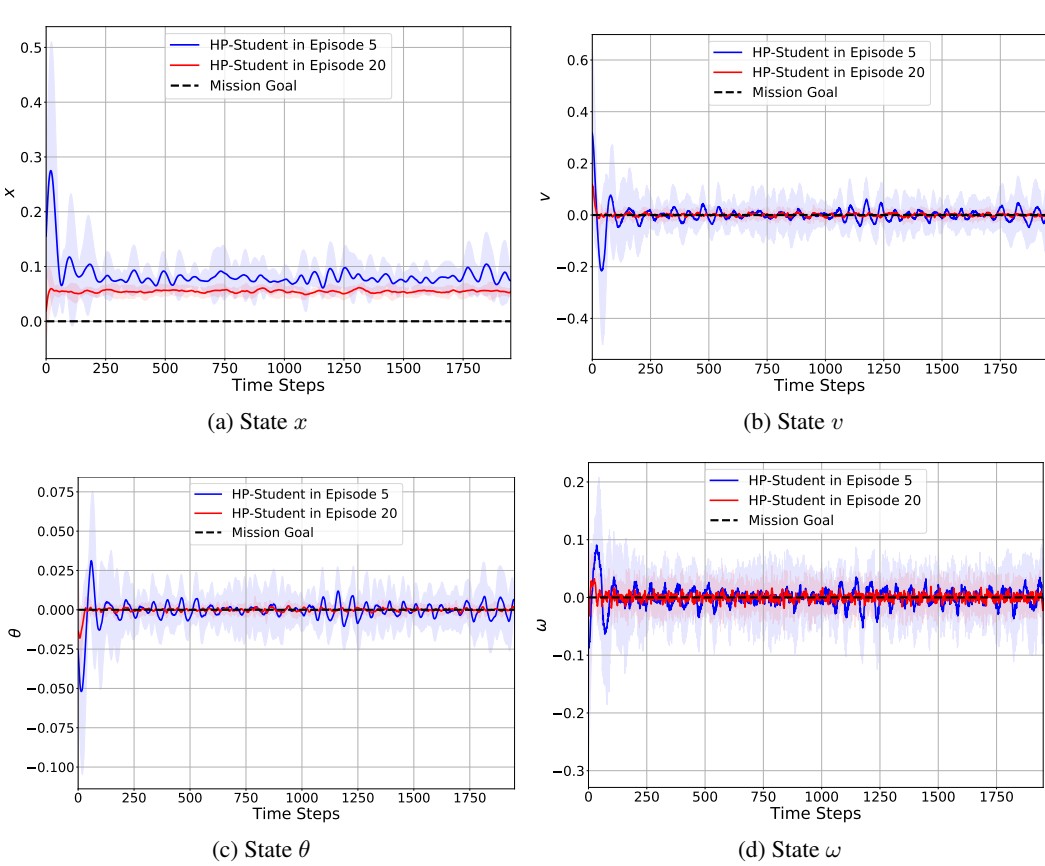

(a) State $x$             (b) State $v$

(c) State $\theta$             (d) State $\omega$

Figure 13: Sole HP-Student controls the real plant: Trajectories of the system for ten random initial conditions in episodes 5 and 20 (95% CI).

### G.5.3 ACTIVATION RATIO OF HA-TEACHER

To demonstrate HA-Teacher's contributions to the claimed automatic hierarchy learning Mechanism, we first define the metric of HA-Teacher's activation ratio:

$$\text{HA-Teacher's activation ratio} = \frac{\text{HA-Teacher's total dwell/activation time in one episode}}{\text{one episode length}} \in (0,1),$$

where a ratio of 0 means HA-Teacher is never activated throughout the entire episode of learning, while a ratio of 1 means HA-Teacher completely dominates HP-Student for the entire episode.

The graph in Figure 14 illustrates the activation ratio trajectories over the episode steps during runtime learning for three different episode lengths and five random seeds. From the graph, we can conclude that HA-Teacher is rarely activated to correct the unsafe learning of HP-Student and support the safety of real plants after 15 episode-steps of runtime learning. This also indicates that HP-Student has learned safety from HA-Teacher.

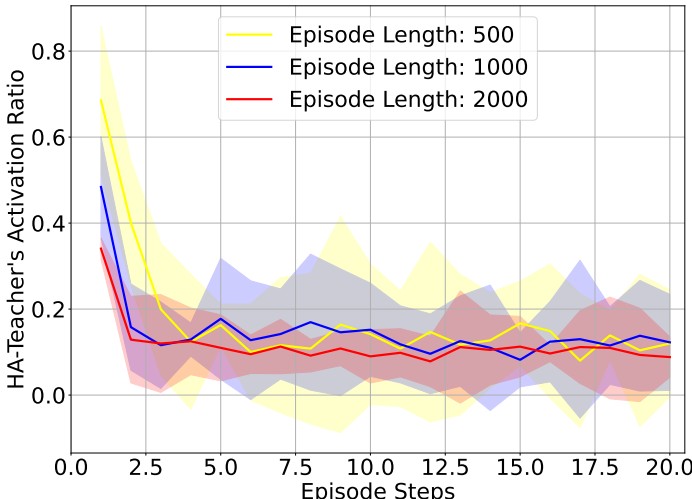

Figure 14: HA-Teacher's activation ratio over 20 episodes, five random seeds (%95 CI).

### G.5.4 HA-TEACHER'S UNSAFE LEARNING CORRECTION

We define "bad" learning data as data that leads to HP-Student's unsafe actions, resulting in the unsafe behavior of real plants. In our runtime learning machine, when such data is detected according to the condition in Equation (7), HA-Teacher steps in to ensure the safety of the real plants and correct the problematic learning data. The corrected data is then stored in HP-Student's replay buffer. This highlights HA-Teacher's role in providing a safe physical learning environment and delivering corrected data to HP-Student. Ultimately, HA-Teacher's correction of unsafe learning, as described in Equations (6) and (10), will contribute to HP-Student's fast and stable learning with larger reward values. To demonstrate this, we conduct an ablation experiment, in which we disable HA-Teacher's mission of unsafe learning correction, resulting in a runtime learning machine "without unsafe-learning correction." So, the compact runtime learning machine is the one "with unsafe-learning correction."

The trajectories of HP-Student's learning reward are shown in Figure 15 for the two learning machines: one with unsafe-learning correction and one without. These trajectories were generated using the same four random initial conditions and ten seeds. Figure 15 emphasizes the important role of HA-Teacher's unsafe learning correction in contributing to HP-Student's fast and stable learning, with larger reward values.

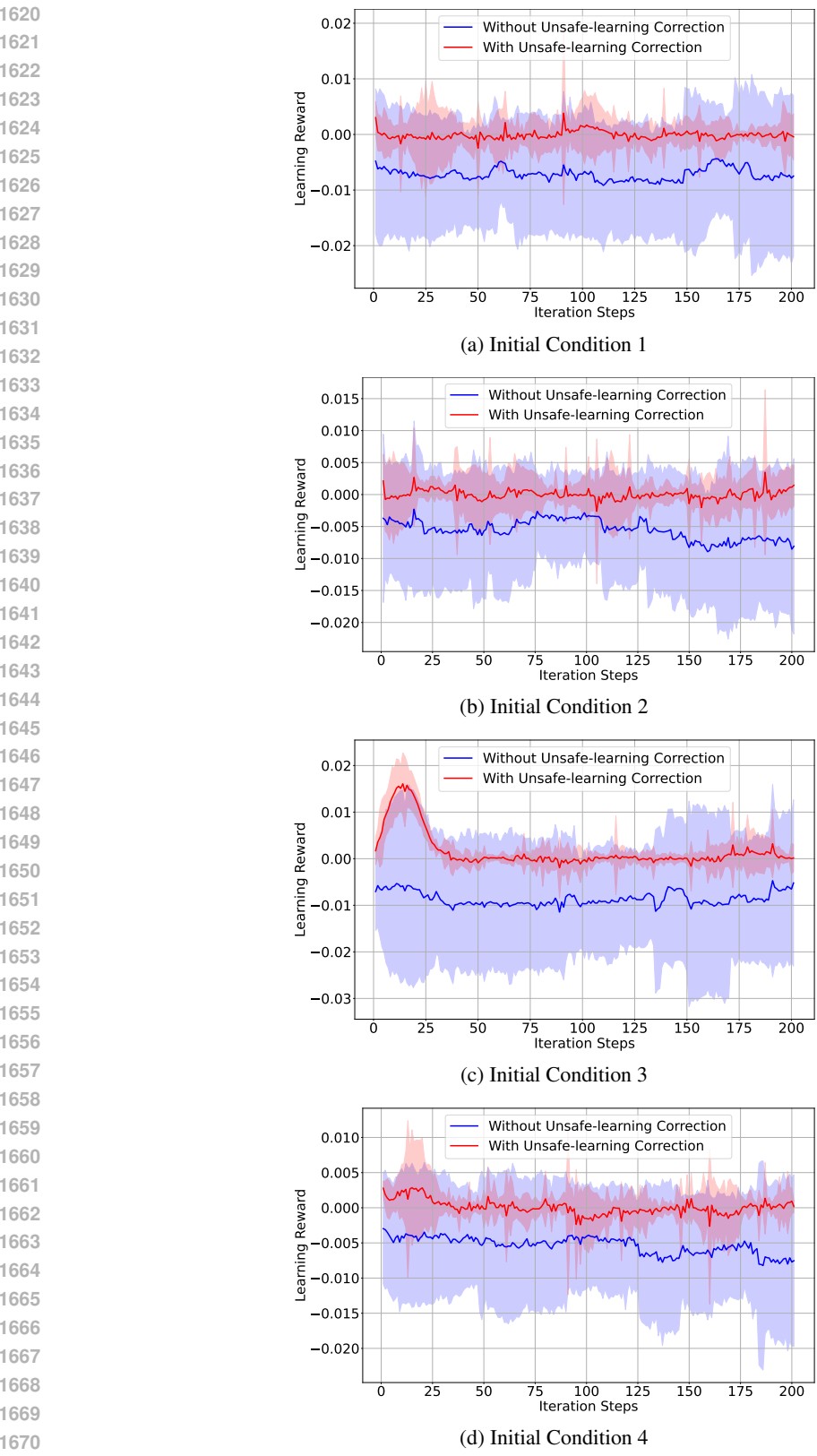

Figure 15: HP-Student's learning rewards for two runtime learning machines: one with HA-Teacher's unsafe learning correction and one without. Trajectories: four random initial conditions and ten seeds (%95 CI).

## H  EXPERIMENT: REAL QUADRUPED ROBOT

In the real quadruped robot experiment, we utilized a Python-based framework designed for the Unitree A1 robot, which was released on GitHub by Yang. This framework consists of a Pybullet-based simulation, an interface for direct simulation-to-real transfer, and an implementation of the Convex Model Predictive Controller for fundamental motion control.

### H.1  POLICY LEARNING

The runtime learning machine and Phy-DRL are designed to achieve the safe mission described in Section 7.2. The policy observation consists of a 10-dimensional tracking error vector between the robot's state vector and the mission vector. Both systems are based on the DDPG algorithm Lillicrap et al. (2016). The actor and critic networks are implemented as Multi-Layer Perceptrons (MLPs) with four fully connected layers. The output dimensions of the critic network are 256, 128, 64, and 1, while the output dimensions of the actor network are 256, 128, 64, and 6. The input for the critic-network consists of the tracking error vector and the action vector, while the input for the actor network is the tracking error vector. The activation functions for the first three neural layers are ReLU, and the output of the last layer is the Tanh function for the actor network and Linear for the critic network. Additionally, the discount factor $\gamma$ is set to 0.9, and the learning rates for the critic and actor networks are both 0.0003. Finally, the batch size is set to 512.

### H.2  SYSTEM DYNAMICS

The robot's physics knowledge used by HP-Student and HA-Teacher for their designs is based on the dynamics model of the robot, which involves a single rigid body subject to forces at the contact patches Di Carlo et al. (2018). Our robot dynamics include the position of the body's center of mass (CoM) height (h), the CoM velocity (v) represented as a 3D vector [CoM x-velocity, CoM y-velocity, CoM z-velocity], the Euler angles (e) described as a 3D vector [roll, pitch, yaw], and the angular velocity in world coordinates (w). According to Di Carlo et al. (2018), this model can describe the body dynamics of quadruped robots.

$$\frac{\mathbf{d}}{\mathbf{d}t}\underbrace{\begin{bmatrix} h \\ \widetilde{\mathbf{e}} \\ \mathbf{v} \\ \mathbf{w} \end{bmatrix}}_{\triangleq\,\widehat{\mathbf{s}}} = \underbrace{\begin{bmatrix} \mathbf{O}_{1\times1} & \mathbf{O}_{1\times5} & 1 & \mathbf{O}_{1\times3} \\ \mathbf{O}_{3\times3} & \mathbf{O}_{3\times3} & \mathbf{O}_{3\times3} & \mathbf{R}(\phi,\theta,\psi) \\ \mathbf{O}_{3\times3} & \mathbf{O}_{3\times3} & \mathbf{O}_{3\times3} & \mathbf{O}_{3\times3} \\ \mathbf{O}_{3\times3} & \mathbf{O}_{3\times3} & \mathbf{O}_{3\times3} & \mathbf{O}_{3\times3} \end{bmatrix}}_{\triangleq\,\widehat{\mathbf{A}}(\phi,\theta,\psi)} \cdot \begin{bmatrix} h \\ \widetilde{\mathbf{e}} \\ \mathbf{v} \\ \mathbf{w} \end{bmatrix} + \widehat{\mathbf{B}}\cdot\widehat{a} + \begin{bmatrix} 0 \\ \mathbf{O}_{3\times1} \\ \mathbf{O}_{3\times1} \\ \widetilde{\mathbf{g}} \end{bmatrix}$$

$$+\ \mathbf{f}(\widehat{\mathbf{s}}), \quad (65)$$

where $\widetilde{\mathbf{g}} = [0; 0; -g] \in \mathbb{R}^3$, with $g$ being the gravitational acceleration. $\mathbf{f}(\widehat{\mathbf{s}})$ denotes model mismatch, $\widehat{\mathbf{B}} = [\mathbf{O}_{4\times6}; \mathbf{I}_6]^\top$, and $\mathbf{R}(\phi,\theta,\psi) = \mathbf{R}_z(\psi) \cdot \mathbf{R}_y(\theta) \cdot \mathbf{R}_x(\phi) \in \mathbb{R}^{3\times3}$ is the rotation matrix, with

$$\mathbf{R}_x(\phi) = \begin{bmatrix} 1 & 0 & 0 \\ 0 & \cos\phi & -\sin\phi \\ 0 & \sin\phi & \cos\phi \end{bmatrix}, \mathbf{R}_y(\theta) = \begin{bmatrix} \cos\theta & 0 & \sin\theta \\ 0 & 1 & 0 \\ -\sin\theta & 0 & \cos\theta \end{bmatrix}, \mathbf{R}_z(\psi) = \begin{bmatrix} \cos\psi & -\sin\psi & 0 \\ \sin\psi & \cos\psi & 0 \\ 0 & 0 & 1 \end{bmatrix}.$$

### H.3  HP-STUDENT: PHYSICS KNOWLEDGE AND DESIGN

In order to represent the model knowledge denoted by $(\mathbf{A}, \ \mathbf{B})$ for robot dynamics in Equation (65), we simplify by setting $\mathbf{R}(\phi,\theta,\psi) = \mathbf{I}_3$, which is achieved by setting the roll, pitch, and yaw angles to zero, i.e., $\phi = \theta = \psi = 0$. By referring to Equation (65) and disregarding any unknown model mismatch, we can derive a simplified linear model for robot dynamics, that is the one below.

$$\frac{\mathbf{d}}{\mathbf{d}t}\underbrace{\begin{bmatrix} \widetilde{h} \\ \widetilde{\mathbf{e}} \\ \widetilde{\mathbf{v}} \\ \widetilde{\mathbf{w}} \end{bmatrix}}_{\triangleq\,\widetilde{\mathbf{s}}} = \underbrace{\begin{bmatrix} \mathbf{O}_{1\times1} & \mathbf{O}_{1\times3} & 1 & \mathbf{O}_{1\times5} \\ \mathbf{O}_{3\times3} & \mathbf{O}_{3\times3} & \mathbf{O}_{3\times3} & \mathbf{R}(\phi,\theta,\psi) \\ \mathbf{O}_{3\times3} & \mathbf{O}_{3\times3} & \mathbf{O}_{3\times3} & \mathbf{O}_{3\times3} \\ \mathbf{O}_{3\times3} & \mathbf{O}_{3\times3} & \mathbf{O}_{3\times3} & \mathbf{O}_{3\times3} \end{bmatrix}}_{\triangleq\,\widetilde{\mathbf{A}}} \cdot \begin{bmatrix} \widetilde{h} \\ \widetilde{\mathbf{e}} \\ \widetilde{\mathbf{v}} \\ \widetilde{\mathbf{w}} \end{bmatrix} + \widehat{\mathbf{B}}\cdot\widetilde{a} \qquad (66)$$

Given the equilibrium point (or control goal) $\mathbf{s}^*$ and $\widetilde{\mathbf{s}} \triangleq \chi \cdot \widehat{\mathbf{s}}_{\sigma(k)}$, we define $\mathbf{s} \triangleq \widetilde{\mathbf{s}} - \mathbf{s}^*$. It is then straightforward to obtain a dynamics from Equation (66) as $\dot{\mathbf{s}} = \widetilde{\mathbf{A}} \cdot \mathbf{s} + \widehat{\mathbf{B}} \cdot \widetilde{\mathbf{a}}$, which transforms to a discrete-time model via sampling technique:

$$\mathbf{s}(k+1) = \mathbf{A} \cdot \mathbf{s}(k) + \mathbf{B} \cdot \widetilde{\mathbf{a}}(k), \text{ with } \mathbf{A} = \mathbf{I}_{10} + T \cdot \widetilde{\mathbf{A}} \text{ and } \mathbf{B} = T \cdot \widehat{\mathbf{B}}, \tag{67}$$

where $T$ is the sampling period.

Given the model knowledge $(\mathbf{A}, \mathbf{B})$ in Equation (67), and according to the design of Phy-DRL in Cao et al. (2024), we have

$$\mathbf{P} = \begin{bmatrix} 122.1647861 & 0 & 0 & 0 & 2.487166 & 0 & 0 \\ 0 & 1.5\mathrm{e}-6 & 0 & 0 & 0 & 0 & 0 \\ 0 & 0 & 1.5\mathrm{e}-6 & 0 & 0 & 0 & 0 \\ 0 & 0 & 0 & 480.6210753 & 0 & 0 & 0 \\ 2.487166 & 0 & 0 & 0 & 3.2176033 & 0 & 0 \\ 0 & 0 & 0 & 0 & 0 & 1.3\mathrm{e}-6 & 0 \\ 0 & 0 & 0 & 0 & 0 & 0 & 1.2\mathrm{e}-6 \\ 0 & 9\mathrm{e}-7 & 0 & 0 & 0 & 0 & 0 \\ 0 & 0 & 9\mathrm{e}-7 & 0 & -0 & 0 & 0 \\ 0 & 0 & 0 & 155.2954559 & 0 & 0 \end{bmatrix}$$

$$\begin{bmatrix} 0 & 0 & 0 \\ 9\mathrm{e}-7 & 0 & 0 \\ 0 & 9\mathrm{e}-7 & 0 \\ 0 & 0 & 155.2954559 \\ 0 & 0 & 0 \\ 0 & 0 & 0 \\ 0 & 0 & 0 \\ 7\mathrm{e}-7 & 0 & 0 \\ 0 & 7\mathrm{e}-7 & 0 \\ 0 & 0 & -0, 156.3068079 \end{bmatrix},$$

$$\mathbf{F} = \begin{bmatrix} 0 & 0 & 0 & 0 & -23.65 & 0 & 0 & 0 & 0 & 0 \\ 0 & 0 & 0 & 0 & 0 & -20 & 0 & 0 & 0 & 0 \\ -63.11 & 0 & 0 & 0 & 0 & 0 & -20 & 0 & 0 & 0 \\ 0 & -32.51 & 0 & 0 & 0 & 0 & 0 & -21.88 & 0 & 0 \\ 0 & 0 & -32.51 & 0 & 0 & 0 & 0 & 0 & -21.88 & 0 \\ 0 & 0 & 0 & -30.95 & 0 & 0 & 0 & 0 & 0 & -22.28 \end{bmatrix},$$

with which and matrices $\mathbf{A}$ and $\mathbf{B}$ in Equation (67), we are able to deliver the residual action policy in Equation (4) and safety-embedded reward in Equation (5).

### H.4 HA-TEACHER: REAL-TIME PATCH

Compared to HP-Student, HA-Teacher possesses a deeper understanding of system dynamics, which is directly and equivalently derived from Equation (65) as

$$\frac{d}{dt} \underbrace{\begin{bmatrix} h \\ \widetilde{\mathbf{e}} \\ \mathbf{v} \\ \mathbf{w} \end{bmatrix}}_{\mathbf{s}} = \underbrace{\begin{bmatrix} \mathbf{O}_{1\times1} & \mathbf{O}_{1\times3} & 1 & \mathbf{O}_{1\times5} \\ \mathbf{O}_{3\times3} & \mathbf{O}_{3\times3} & \mathbf{O}_{3\times3} & \mathbf{R}(\phi,\theta,\psi) \\ \mathbf{O}_{3\times3} & \mathbf{O}_{3\times3} & \mathbf{O}_{3\times3} & \mathbf{O}_{3\times3} \\ \mathbf{O}_{3\times3} & \mathbf{O}_{3\times3} & \mathbf{O}_{3\times3} & \mathbf{O}_{3\times3} \end{bmatrix}}_{\widehat{\mathbf{A}}(\mathbf{s})} \cdot \begin{bmatrix} h \\ \widetilde{\mathbf{e}} \\ \mathbf{v} \\ \mathbf{w} \end{bmatrix} + \underbrace{\begin{bmatrix} \mathbf{O}_3 & \mathbf{O}_3 & \mathbf{O}_3 & \mathbf{O}_3 \\ \mathbf{O}_3 & \mathbf{O}_3 & \mathbf{O}_3 & \mathbf{O}_3 \\ \mathbf{O}_3 & \mathbf{O}_3 & \mathbf{I}_3 & \mathbf{O}_3 \\ \mathbf{O}_3 & \mathbf{O}_3 & \mathbf{O}_3 & \mathbf{I}_3 \end{bmatrix}}_{\widehat{\mathbf{B}}(\mathbf{s})} \cdot \mathbf{a}$$

$$+ \mathbf{g}(\mathbf{s}), \tag{68}$$

where $\widehat{\mathbf{A}}(\mathbf{s})$ and $\widehat{\mathbf{B}}(\mathbf{s})$ are known to the HA-Teacher. The sampling technique transforms the continuous-time dynamics model in Equation (68) to the discrete-time one:

$$\mathbf{s}(k+1) = (\mathbf{I}_4 + T \cdot \widehat{\mathbf{A}}(\mathbf{s})) \cdot \mathbf{s}(k) + T \cdot \widehat{\mathbf{B}}(\mathbf{s}) \cdot \mathbf{a}(k) + T \cdot \mathbf{g}(\mathbf{s}),$$

from which we obtain the knowledge of $\mathbf{A}(\widehat{\mathbf{s}}_{\sigma(k)})$ and $\mathbf{B}(\widehat{\mathbf{s}}_{\sigma(k)})$ in Equation (15) as

$$\mathbf{A}(\widehat{\mathbf{s}}_{\sigma(k)}) = \mathbf{I}_4 + T \cdot \widehat{\mathbf{A}}(\widehat{\mathbf{s}}_{\sigma(k)}) \text{ and } \mathbf{B}(\widehat{\mathbf{s}}_{\sigma(k)}) = T \cdot \widehat{\mathbf{B}}(\widehat{\mathbf{s}}_{\sigma(k)}). \tag{69}$$

Meanwhile, for the patch in Equation (11), the model mismatch in Assumption 6.3, and the dwell time in Equation (8), we let $\chi = 0.25$, $\kappa = 0.01$, and $\tau = 100$. For LMIs in Equations (27) to (30), we let $\alpha = 0.9$, $\gamma_1 = 1$, and $\gamma_2 = 0.45$. Finally, according to the given safety set $\mathbb{X} = \{\mathbf{s} \mid |\text{CoM x-velocity} - 0.3 \text{ m/s}| \le 0.3 \text{ m/s}, |\text{CoM z-height} - 0.24 \text{ m}| \le 0.15 \text{ m}\}$ and the action space of HA-Teacher $\mathbb{A} = \{\mathbf{a}_{\text{HA}} \mid |\mathbf{a}_{\text{HA}}| \le [30, 30, 30, 60, 60, 60]^\top\}$, we obtain following knowledge for the LMIs in Equations (18) and (23):

$$\mathbf{D} = \begin{bmatrix} 1/0.15 & 0 & 0 & 0 & 0 & 0 & 0 & 0 & 0 & 0 \\ 0 & 0 & 0 & 0 & 1/0.3 & 0 & 0 & 0 & 0 & 0 \end{bmatrix},$$

$$\mathbf{C} = \begin{bmatrix} 1/30 & 0 & 0 & 0 & 0 & 0 \\ 0 & 1/30 & 0 & 0 & 0 & 0 \\ 0 & 0 & 1/30 & 0 & 0 & 0 \\ 0 & 0 & 0 & 1/60 & 0 & 0 \\ 0 & 0 & 0 & 0 & 1/60 & 0 \\ 0 & 0 & 0 & 0 & 0 & 1/60 \end{bmatrix}.$$

### H.5 ADDITIONAL EXPERIMENTAL RESULTS

#### H.5.1 TRAJECTORIES IN DIFFERENT EPISODES

The real robot's trajectories of CoM height and CoM x-velocities under the control of runtime learning machine in episodes 5, 10, 15, and 20 are shown in Figure 16. The trajectories straightforwardly depict that the runtime learning machine guarantees the safety of real robots in all picked episodes.

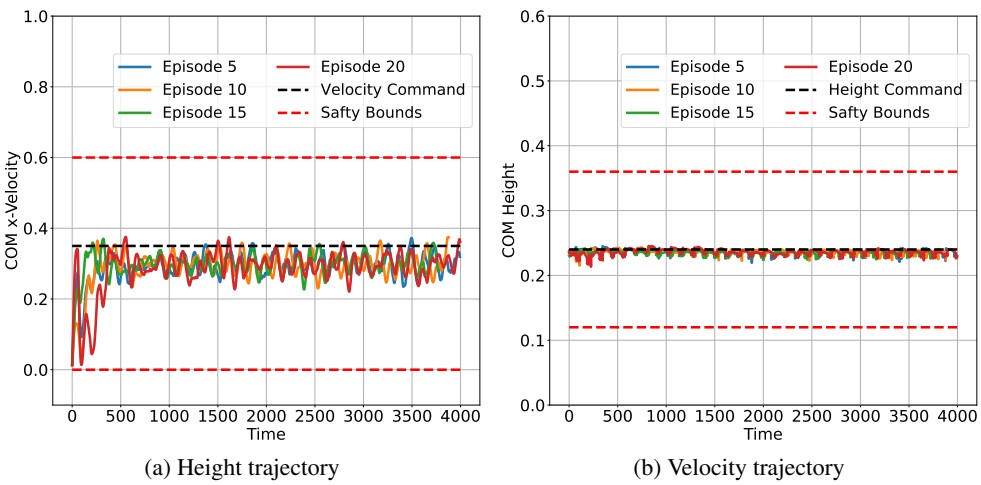

(a) Height trajectory     (b) Velocity trajectory

Figure 16: Robot's trajectories of CoM height and x-velocity under control of runtime learning machine in the episodes 5, 10, 15, and 20.

#### H.5.2 TRAJECTORIES IN FACE OF UNKNOWN UNKNOWNS

Figure 17 presents the trajectories of CoM height and CoM x-velocity of real robot under control of runtime learning machines, in the face of three unknown unknowns: i) 'Beta + PD,' ii) 'Beta + DoS + Kick,' and iii) 'Beta + PS.' Figure 17 shows that the two states are successfully constrained into safety set, i.e., never exceed safety bounds.

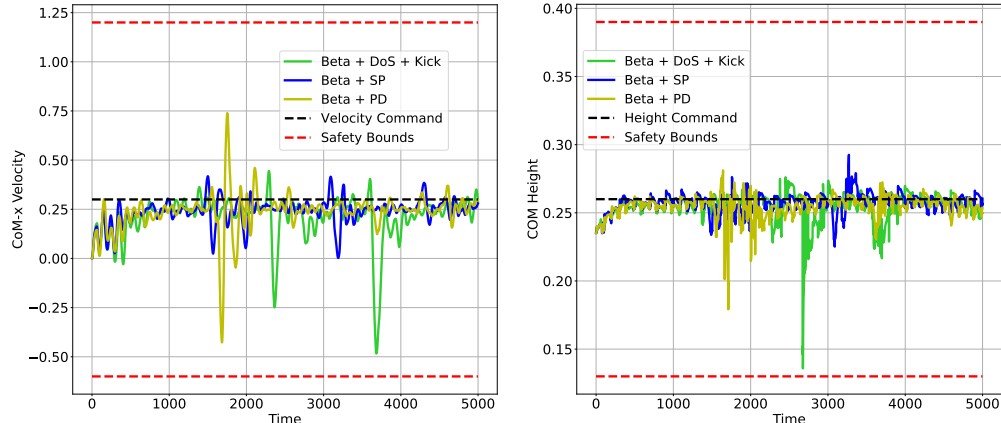

Figure 17: Trajectories in the presence of unknown unknowns.

# I EXPERIMENT: GO2 QUADRUPED ROBOT AND 2D QUADROTOR

The section presents experiments on two new benchmarks: a go2 quadruped robot and a 2D quadrotor.

## I.1 GO2 QUADRUPED ROBOT

Many safety-critical autonomous systems, such as quadruped robots, drones, UAVs, and autonomous vehicles, interact dynamically with their environments. For instance, the movement dynamics on a sandy road will be different from those on a surface covered in freezing rain. As a result, the operating environment plays a crucial role in introducing real-time unknown unknowns, Sim2Real gap, and domain gap. So, this subsection's experiment aims to demonstrate the safety assurance of our runtime learning machine in challenging environments.

To do so, we initially pre-trained HP-Student for the A1 robot in the PyBullet simulator, using a flat terrain environment, as the same one in Section 7.2. After this pre-training, we directly deployed HP-Student to the Go2 robot. We utilized NVIDIA Isaac Sim to create an operating environment for showcasing the Go2 robot's runtime learning capabilities. This environment transitions from flat terrain to unstructured and uneven ground, further complicated by ice from unforseen freezing rain. We here can conclude that Go2 robot's operating environment are non-stationary and unforeseen, and never occur in the pre-training stage. Besides, A1 and Go2 robots are very different in their motors, weights, heights, mass, etc.

For the Go2 robot, its safety set is

$$\mathbb{X} = \left\{ \mathbf{s} \mid |\text{CoM x-velocity} - r_{v_x}| \leq 0.4 \text{ m/s}, |\text{CoM z-height} - r_h| \leq 0.15 \text{ m} \right\}. \quad (70)$$

All other designs are the same as those of A1 quadruped robot, presented in Appendix H.

### I.1.1 NON-STATIONARY, UNSTRUCTURED, UNEVEN, AND UNFORESEEN ENVIRONMENTS

In the challenging real-time operating environments, our first mission command sent to the robot is *walking forward at velocity 0.7 m/s (i.e., $r_{v_x}$ = 0.7 m/s) and maintaining CoM height at $r_h$ = 0.3 m, while constraining them to the safety set in Equation* (70). When we disable HA-Teacher's real-time patch and unsafe learning correction, our runtime learning machine degrades to the recently runtime assurance Chen et al. (2022); Brat & Pai (2023); Sifakis & Harel (2023), which is also proposed to support runtime learning in real plants. When HA-Teacher is completely disabled for backing up safety, our runtime learning machine further degrade to pure Phy-DRL Cao et al. (2024; 2023).

The demonstration video of the well-pretrained HP-Student (Phy-DRL) in PyBullet, along with the execution of mission command by our runtime learning machine and runtime assurance, is available at Go2 Forward [anonymous hosting and browsing]. Besides, Figure 18 shows the robot's trajectories of CoM height and CoM x-velocity. We also set the second mission command: *walking backward at velocity 0.7 m/s (i.e., $r_{v_x}$ = -0.7 m/s) while maintaining CoM height at $r_h$ = 0.3 m,*

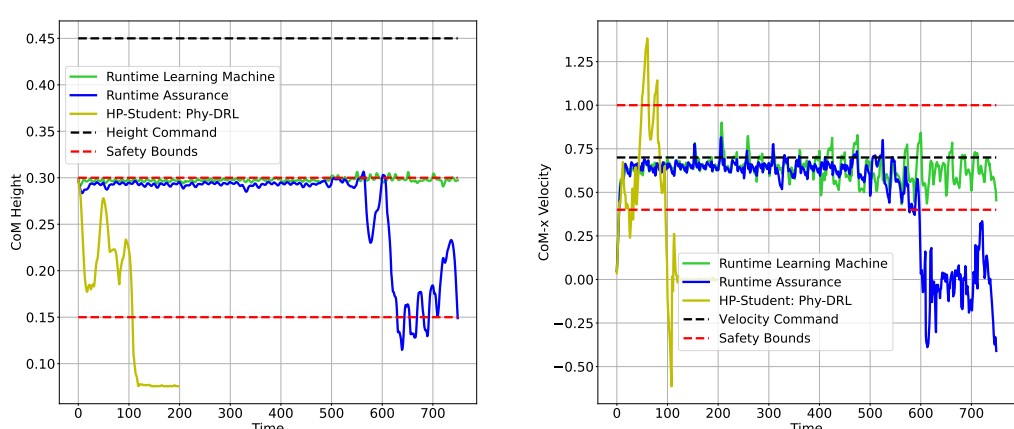

Figure 18: Trajectories of CoM height and CoM velocity in non-stationary, unstructured, uneven, and unforeseen environments, given the command of walking forward at 0.7 m/s.

*while constraining them to the safety set in Equation* (70). The demonstration video is available at Go2 Backward [anonymous hosting and browsing]. Figure 19 shows the robot's trajectories of CoM height and CoM x-velocity for the second mission command.

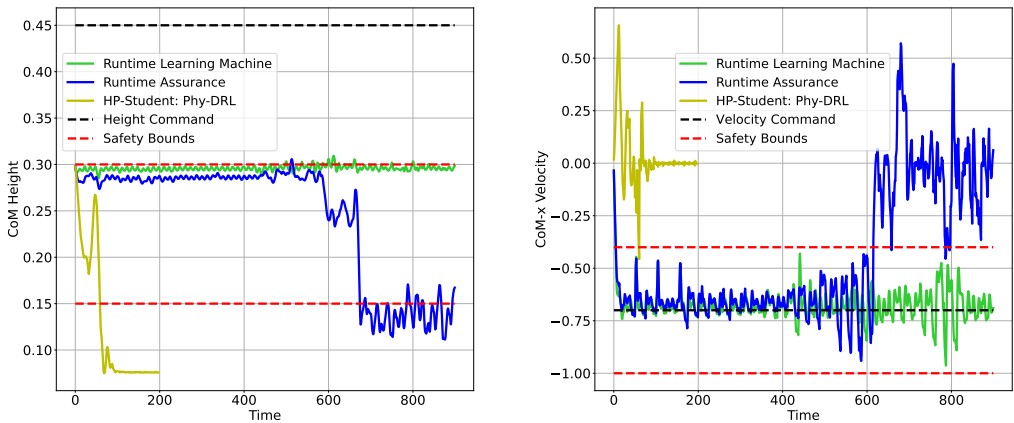

Figure 19: Trajectories of CoM height and CoM velocity in non-stationary, unstructured, uneven, and unforeseen environments, given the command of walking backward at 0.7 m/s.

Upon watching the demonstration videos and observing Figures 18 and 19, we concluded that our runtime learning machine can guarantee the safety of the Go2 robot operating in non-stationary, unstructured, uneven, and unforeseen environments. In contrast, runtime assurance and continual learning by sole Phy-DRl cannot achieve this.

### I.1.2 RANDOM KICKING AND NOISY STATE SAMPLING

In addition to the non-stationary, unstructured, and uneven operating environments, we inject unknown-unknown noise into state samplings and randomly kick the Go2 robot to demonstrate the machine's capability of assuring safety. Following the method of inducing action noise in Section 7, the unknown-unknown noise for state samplings are generated by a randomized Beta distribution. Appendix F explains why the randomized Beta distribution generate one kind of unknown

unknown. In presence of the two additional unknown unknowns, the demonstration video of robot's execution of the two mission commands (given in Appendix I.1.1) by our runtime learning machine is available at Go2: Kick–Sensor [anonymous hosting and browsing]. Meanwhile, the phase plots of CoM height and CoM x-velocity are shown in Figure 20. They well demonstrate the significantly enhanced safety assurance by our runtime learning machine in the complex setting.

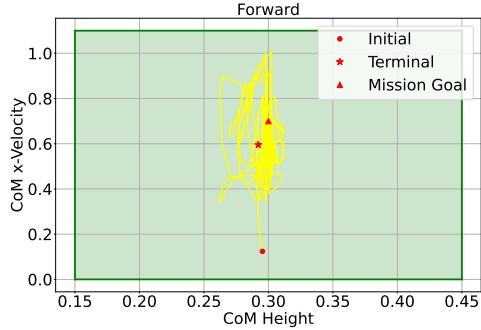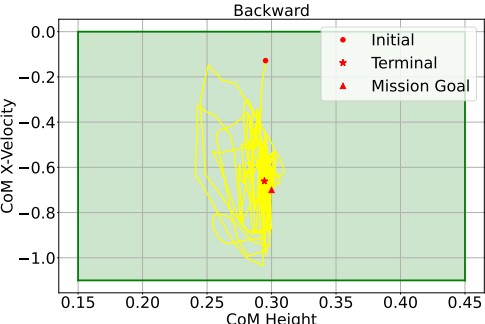

Figure 20: Phase plots of CoM height and CoM x-velocity in non-stationary, unstructured, and uneven operating environments, with unknown-unknown noise injected into state samplings and randomly kicking on robot.

## I.2  2D QUADROTOR

We take the 2D quadrotor simulator provided in Safe-Control-Gym Yuan et al. (2022) as an experimental system to demonstrate of the mechanism of automatic hierarchy learning. It is characterized by $(x, z)$ – the translation position of the CoM of the quadrotor in the $xz$-plane, $\theta$ – the pitch angle, and their velocities $v_x = \dot{x}$, $v_z = \dot{z}$, and $v_\theta = \dot{\theta}$. The mission of the action policy is to stabilize the quadrotor at the waypoint $(r_x, r_z, r_\theta)$ under safety constraints:

$$|x - r_x| \leq 0.5 \text{ m}, \ |z - r_z| \leq 0.8 \text{ m}, \ |\theta - r_\theta| \leq 0.8 \text{ rad}. \tag{71}$$

### I.2.1  POLICY LEARNING

The actor and critic networks are implemented as a Multi-Layer Perceptron (MLP) with four fully connected layers. The output dimensions of critic and actor networks are 256, 128, 64, and 1, respectively. The activation functions of the first three neural layers are ReLU, while the output of the last layer is the Tanh function for the actor-network and Linear for the critic network. The input of the critic network is $[\mathbf{s}; \mathbf{a}]$, while the input of the actor-network is $\mathbf{s}$. In more detail, we let discount factor $\gamma = 0.9$, and the learning rates of critic and actor networks are the same as 0.0003. We set the batch size to 200.

### I.2.2  HP-STUDENT: PHY-DRL DESIGN

According to Safe-Control-Gym Yuan et al. (2022), the dynamics model of 2D Quadrotor is

$$\ddot{x} = \frac{T_1 + T_2}{m} \cdot \sin(\theta) \tag{72a}$$

$$\ddot{z} = \frac{T_1 + T_2}{m} \cdot \cos(\theta) - g \tag{72b}$$

$$\ddot{\theta} = \frac{\sqrt{2}}{2} \cdot l \cdot \frac{(T_2 - T_1)}{I_{yy}}, \tag{72c}$$

where $(x, z)$ is the translation position of the CoM of the quadrotor in the $xz$-plane, $\theta$ is the pitch angle, $T_1$ and $T_2$ are the thrusts generated by two pairs of motors (one on each side of the body's $y$-axis), $m = 0.027$ is the mass of quadrotor, $g = 9.8$ m/$s^2$ is the gravitational acceleration, $l = 0.0397$ m is the arm length of the quadrotor, and $I_{yy} = 1.4e^{-5}$ is the moment of inertia about the $y$-axis.

To have the model for Phy-DRL, we let $\sin(\theta) \approx \theta$, $\cos(\theta) \approx 1$, and $\ddot{z} \approx 0$. In this way, the dynamics model in Equation (72) transforms to

$$\ddot{x} = g \cdot \theta \tag{73a}$$

$$\ddot{z} = \frac{T_1 + T_2}{m} - g \tag{73b}$$

$$\ddot{\theta} = \frac{\sqrt{2}}{2} \cdot l \cdot \frac{(T_2 - T_1)}{I_{yy}}, \tag{73c}$$

The state vector $s \in \mathbf{R}^6$ is $[x, \dot{x}, z, \dot{z}, \theta, \dot{\theta}]$. The action vector $a \in \mathbf{R}^2$ is $[T_1, T_2]$. We let $v_x = \dot{x}$, $v_z = \dot{z}$, and $v_\theta = \dot{\theta}$. The linear state space model is transformed from Equation (73) as

$$\frac{d}{dt} \underbrace{\begin{bmatrix} x \\ v_x \\ z \\ v_z \\ \theta \\ v_\theta \end{bmatrix}}_{\triangleq \, \widehat{\mathbf{s}}} = \underbrace{\begin{bmatrix} 0 & 1 & 0 & 0 & 0 & 0 \\ 0 & 0 & 0 & 0 & g & 0 \\ 0 & 0 & 0 & 1 & 0 & 0 \\ 0 & 0 & 0 & 0 & 0 & 0 \\ 0 & 0 & 0 & 0 & 0 & 1 \\ 0 & 0 & 0 & 0 & 0 & 0 \end{bmatrix}}_{\triangleq \, \widehat{\mathbf{A}}} \cdot \begin{bmatrix} x \\ v_x \\ z \\ v_z \\ \theta \\ v_\theta \end{bmatrix} + \underbrace{\begin{bmatrix} 0 & 0 \\ 0 & 0 \\ 0 & 0 \\ \frac{1}{m} & \frac{1}{m} \\ 0 & 0 \\ \frac{-\sqrt{2}l}{2 \cdot I_{yy}} & \frac{\sqrt{2}l}{2 \cdot I_{yy}} \end{bmatrix}}_{\triangleq \, \widehat{\mathbf{B}}} \cdot \underbrace{\begin{bmatrix} T_1 \\ T_2 \end{bmatrix}}_{\triangleq \, \mathbf{a}} + \underbrace{\begin{bmatrix} 0 \\ 0 \\ 0 \\ -g \\ 0 \\ 0 \end{bmatrix}}_{\triangleq \, \mathbf{f}}. \tag{74}$$

For trajectory tracking tasks, the set point varies depending on the pre-computed set points, which is denoted by $\mathbf{s}^* = [x_r, 0, z_r, 0, 0, 0]^\top$, and define $\mathbf{s} = \widehat{\mathbf{s}} - \mathbf{s}^*$. We then consider the digital sampling technique, which yields a discrete-time model of tracking error from Equation (74)

$$\mathbf{s}(k+1) = \mathbf{A} \cdot \mathbf{s}(k) + \mathbf{B} \cdot \mathbf{u}(k) + T \cdot \mathbf{f}, \tag{75}$$

where $T = \frac{1}{50}$ sec is the sampling period, and

$$\mathbf{A} = \mathbf{I}_6 + T \cdot \widehat{\mathbf{A}}, \quad \mathbf{B} = T \cdot \widehat{\mathbf{B}}, \tag{76}$$

Considering equation 71, we have

$$\mathbf{D} = \begin{bmatrix} 2 & 0 & 0 & 0 & 0 & 0 \\ 0 & 0 & 1.25 & 0 & 0 & 0 \\ 0 & 0 & 0 & 0 & 1.25 & 0 \end{bmatrix},$$

with which, $\alpha = 0.85$, and the knowledge of the model $(\mathbf{A}, \mathbf{B})$ in Equation (76), by solving the analytic centering problem via PYCVX toolbox, we have

$$\mathbf{P} = \begin{bmatrix} 17.7508 & 5.9194 & -0.0000 & -0.0000 & 8.1374 & 0.0810 \\ 5.9194 & 2.5987 & -0.0000 & -0.0000 & 3.6321 & 0.0369 \\ -0.0000 & -0.0000 & 1.9843 & 0.1046 & -0.0000 & -0.0000 \\ -0.0000 & -0.0000 & 0.1046 & 0.0259 & -0.0000 & -0.0000 \\ 8.1374 & 3.6321 & -0.0000 & -0.0000 & 7.3610 & 0.0764 \\ 0.0810 & 0.0369 & -0.0000 & -0.0000 & 0.0764 & 0.0014 \end{bmatrix},$$

$$\mathbf{F} = \begin{bmatrix} 0.3282 & 0.1500 & -0.2075 & -0.0249 & 0.3275 & 0.0066 \\ -0.3282 & -0.1500 & -0.2075 & -0.0249 & -0.3275 & -0.0066 \end{bmatrix}.$$

Now, with $\mathbf{P}, \mathbf{F}, \mathbf{A}$, and $\mathbf{B}$ at hand, we can deliver the safety-embedded reward in Equation (5), and the model-based policy in for the residual policy diagram Equation (4).

### I.2.3   HA-TEACHER DESIGN

The model in Equation (72) is used for HA-Teacher to have real-time, state-dependent dynamics model. The dynamics model can equivalently transform to

$$\frac{d}{dt} \underbrace{\begin{bmatrix} x \\ v_x \\ z \\ v_z \\ \theta \\ v_\theta \end{bmatrix}}_{\triangleq \, \widehat{\mathbf{s}}} = \underbrace{\begin{bmatrix} 0 & 1 & 0 & 0 & 0 & 0 \\ 0 & 0 & 0 & 0 & g & 0 \\ 0 & 0 & 0 & 1 & 0 & 0 \\ 0 & 0 & 0 & 0 & 0 & 0 \\ 0 & 0 & 0 & 0 & 0 & 1 \\ 0 & 0 & 0 & 0 & 0 & 0 \end{bmatrix}}_{\triangleq \, \widehat{\mathbf{A}}} \cdot \begin{bmatrix} x \\ v_x \\ z \\ v_z \\ \theta \\ v_\theta \end{bmatrix} + \underbrace{\begin{bmatrix} 0 & 0 \\ \frac{\sin(\theta)}{m} & \frac{\sin(\theta)}{m} \\ 0 & 0 \\ \frac{\cos(\theta)}{m} & \frac{\cos(\theta)}{m} \\ 0 & 0 \\ \frac{-\sqrt{2}l}{2 \cdot I_{yy}} & \frac{\sqrt{2}l}{2 \cdot I_{yy}} \end{bmatrix}}_{\triangleq \, \widehat{\mathbf{B}}(\widehat{\mathbf{s}})} \cdot \underbrace{\begin{bmatrix} T_1 \\ T_2 \end{bmatrix}}_{\triangleq \, \mathbf{a}} + \underbrace{\begin{bmatrix} 0 \\ 0 \\ 0 \\ -g \\ 0 \\ 0 \end{bmatrix}}_{\triangleq \, \mathbf{f}},$$

from which, with the consideration of sampling technique, we can obtain the knowledge of $\mathbf{A}(\widehat{\mathbf{s}}_{\sigma(k)})$ and $\mathbf{B}(\widehat{\mathbf{s}}_{\sigma(k)})$ in Equation (15) as

$$\mathbf{A}(\widehat{\mathbf{s}}_{\sigma(k)}) = \mathbf{I}_4 + T \cdot \widehat{\mathbf{A}} \ \text{ and } \ \mathbf{B}(\widehat{\mathbf{s}}_{\sigma(k)}) = T \cdot \widehat{\mathbf{B}}(\widehat{\mathbf{s}}_{\sigma(k)}). \tag{77}$$

Meanwhile, for the patch in Equation (11), the model mismatch in Assumption 6.3, and the dwell time in Equation (8), we let $\chi = 0.25$, $\kappa = 0.01$, and $\tau = 30$. For LMIs in Equations (27) to (30), we let $\alpha = 0.9$, $\gamma_1 = 1$, and $\gamma_2 = 0.45$. The the actions' space are set as $[-0.15, 0.15]$.

### I.2.4 Automatic Hierarchy Learning and Lifetime Safety: Learning from Scratch

We now present the experimental results to demonstrate the automatic hierarchy learning (i.e., safety-first learning, and then high-performance learning for high-performance action policy) of our runtime learning machine. We note the runtime learning can be understood as safe continual learning (if having after pre-training of HP-Student) for a high-performance action policy in real plants – using real-time sensor data generated from real-time physical environments. The pre-training can help reduce the workload of runtime learning or test the operational mechanisms, it is not mandatory for the runtime learning process. If the pre-training is removed, the our runtime learning will be learning from scratch in real plants. All the previous experiment of cart-pole system, A1 and Go2 robots have the pre-training of HP-Student. We now consider the case, where the HP-Student does not have pre-training. The episode length is 1500 steps.

The phase plots illustrating tracking errors of the x and z positions across episodes 1 to 8 are presented in Figure 21. In these plots, red dots represent the states of the system controlled by HA-Teacher, while blue dots represent the system controlled by HP-Student. Initially, the HA-Teacher was frequently activated because the HP-Student had not yet learned how to ensure safety. However, as the HP-Student spent more time learning, his ability to maintain safety improved. This progress is particularly evident in episodes 1 to 5, during which the frequency of HA-Teacher activation decreased. By episode 6, the HP-Student had mastered the capability to ensure safety, and thereafter, his continued runtime learning no longer required assistance from the HA-Teacher. This advancement enabled him to autonomously develop a high-performance action policy, resulting in an end state close to the goal (i.e., the center of the ellipse safety envelope).

Figure 21 also demonstrate the distinguished feature – lifetime safety: safety guarantee (i.e., system states never leave the green safety set) in any learning episode.

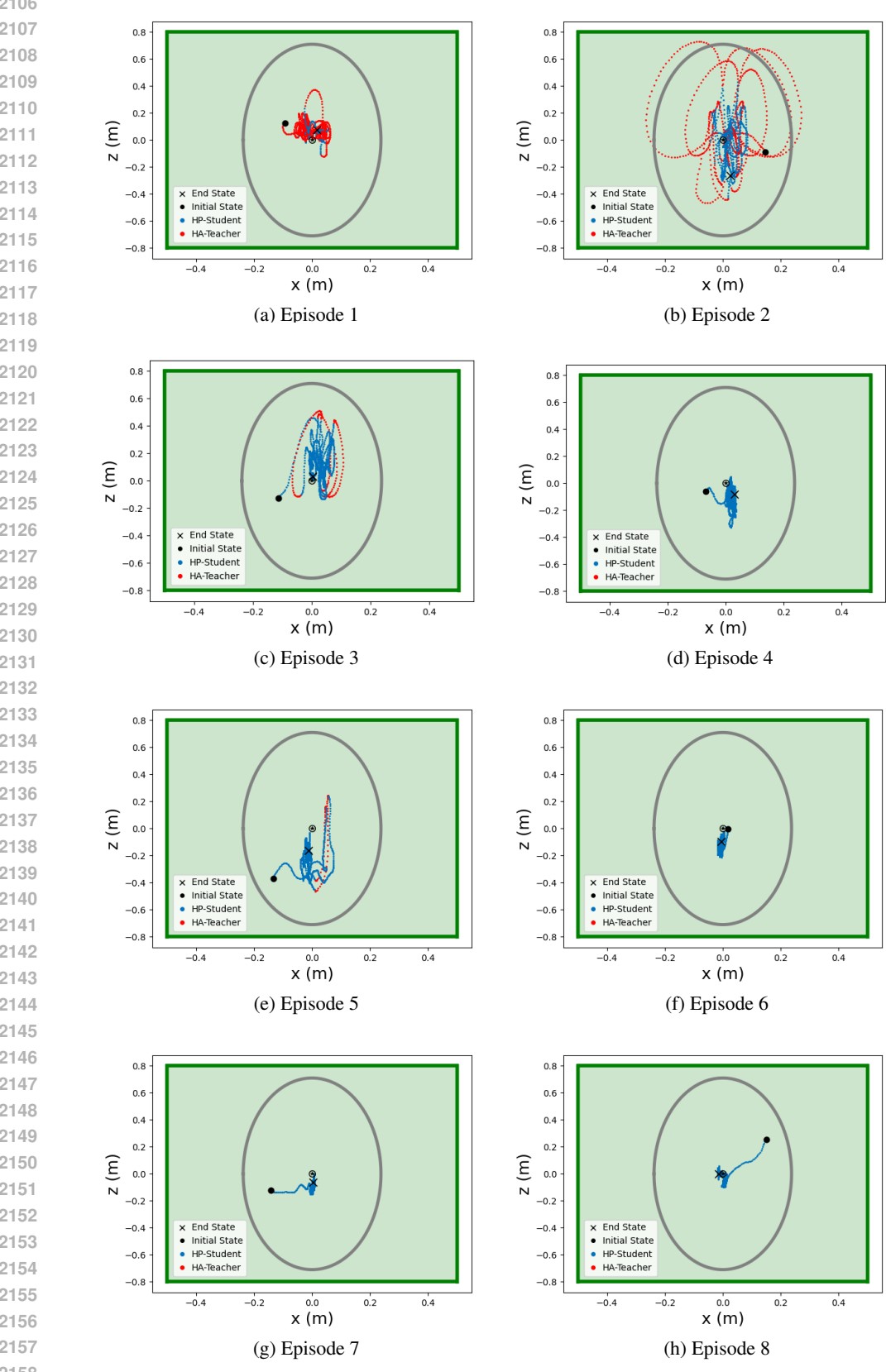

Figure 21: Phase plots in episodes 1–8.

## J IMPLEMENTATION PROBLEMS AND SOLUTIONS

This section addresses the problems of implementing the runtime learning machine in the cart-pole system and the real quadruped robot, along with our solutions.

### CVXPY TOOLBOX

The runtime learning machine depends on obtaining feasible solutions from Linear Matrix Inequalities (LMIs) to allow the HA-Teacher to perform safe actions. We found that the default CVX solver, SCS, exhibited instability and inconsistent accuracy across different computing platforms. Therefore, in our experiment, we chose to use the CVXOPT solver Andersen et al. (2013) to achieve more stable and accurate results.

While LMIs can generally be solved in real-time, the operating frequency on our platform still varies between 10 and 50 Hz. Although this fluctuation does not adversely affect overall performance in simulations, it does pose challenges when implementing the framework on a physical platform. To ensure efficient operation on hardware that requires a high frequency, such as a quadruped robot, we designed the HA-Teacher with an additional process that runs in parallel with the central control process. This design, however, adds complexity, such as the need for multi-process synchronization, and increases the demand for computational resources.

### HARDWARE REAL-TIME EFFICIENCY

The runtime learning machine was tested by sending remote control commands to the quadruped robot indoors. All computations were executed on a desktop equipped with a 12th Gen Intel® Core™ i9-12900K 16-core processor. The onboard computing platform of the Unitree A1 utilizes an Intel® Atom™ x5-Z8350 CPU, which operates at 1.44 GHz with 6 cores. To enhance development efficiency, Python was chosen as the programming language for implementing this framework.

Given that the runtime learning process requires significant interaction between the CPU and GPU, and considering the current architecture includes a high-frequency Model Predictive Control (MPC) module operating at 500 Hz for the quadruped robot, achieving comparable optimal real-time performance with the existing onboard hardware may be challenging. However, we believe that this computational limitation can be addressed or significantly alleviated by either adding extra computing resources—such as an external mini-PC, as suggested in Yang et al. (2022b)—or by restructuring the code framework. This restructuring could involve encapsulating the current implementation in C++ to enhance real-time performance, as proposed in Chen & Nguyen (2024).

## K  COMPUTATION RESOURCES

In all case studies, we trained and tested the deep reinforcement learning (DRL) algorithm on a desktop computer running Ubuntu 22.04. The desktop was equipped with a 12th Gen Intel(R) Core(TM) i9-12900K 16-core processor, 64 GB of RAM, and an NVIDIA GeForce GTX 3090 GPU. The DRL algorithm was implemented in Python using the TensorFlow framework. We utilized the open-source Python CVX solver to solve LMI (Linear Matrix Inequalities) problems.

In our system architecture, the computation of $\widehat{\mathbf{F}}_{\sigma(k)}$ and $\widehat{\mathbf{P}}_{\sigma(k)}$ for the HA-Teacher at each patch needs to be performed when the Safety Coordinator is triggered. To ensure real-time computation of CVX and interaction with the environment, we have implemented a multi-processing pipeline to control the robot and solve LMIs in parallel in real-time. For solving LMIs, we always allow the solver to use the most recent state so that when the safety coordinator is triggered, the latest $\widehat{\mathbf{F}}$ and $\widehat{\mathbf{P}}$ are readily available. We have taken into account the delay issue and formulated it in the LMI problems.

We observed that the MATLAB-based CVX solver consistently solved the LMIs problem better than the Python-based solver, providing more reliable solutions. However, transferring data between MATLAB and Python could cause additional delays when updating $\widehat{\mathbf{F}}_{\sigma(k)}$ and $\widehat{\mathbf{P}}_{\sigma(k)}$ for HA-Teacher. Additionally, implementing multiprocessing in both MATLAB and Python posed technical challenges due to software compatibility issues. As a result, we opted for the Python-based CVX solver for real-time real-world experiments, while recommending the MATLAB-based solver for less time-sensitive applications.

