# OpenReview forum: "Runtime Learning Machine"
_ICLR.cc/2025/Conference — Submitted to ICLR 2025_

### Official Review · Reviewer_xKGE · 2024-10-31

**Soundness:** 3
**Presentation:** 2
**Contribution:** 3
**Rating:** 8
**Confidence:** 3

**Summary:**

This paper introduces a runtime learning machine for safety-critical autonomous systems, featuring an interactive HP-Student, HA-Teacher, and Coordinator. It ensures lifetime safety by addressing unknowns and the Sim2Real gap, enabling real-time learning for safe, high-performance policies. Experimental results validate its effectiveness on cart-pole and quadruped robot systems.

**Strengths:**

- The proposed method combines a High Assurance Teacher with a Performance optimizing student to yield strong real word robustness results on a challenging quadruped robot.
- The experiments show scalable computation that can work fast enough for real-time learning using matrix toolboxes and related libraries.

**Weaknesses:**

- The setting considered assumes knowledge of the environment dynamics by the HA-Teacher. System identification is proposed but does not appear to have been experimented with.
- The presentation is cluttered at times and causes some difficulty in parsing. For example, some of the numerous assumptions, remarks, characteristics, and definitions should be consolidated if possible. Section 1.3 is almost entirely italicized.

**Questions:**

1. How is this approach placed in the context of other shielding approaches [1,2]?
2. The triggering condition in Eq 7 is entirely causal (i.e., depends only on the current time and prior to the current time). Why is this better than using the environment model to predict for a future failure from the current state and input?
3. How does the approach generalize to different safety sets (e.g., velocity < 1.5 vs velocity < 3)? Will the entire training process be needed for each change? How long would the training time be for each minor change?
4. Are there any experiments or examples with System Identification as mentioned in Remark 6.5 or Appendix C?

### References:
[1] Safe Reinforcement Learning with Nonlinear Dynamics via Model Predictive Shielding, Bastani, ACC 2020

[2] Dynamic Model Predictive Shielding for Provably Safe Reinforcement Learning, Banerjee et al, arXiv 2024

---

> ### Author Response · Authors · 2024-11-17
> **Response to Reviewer xKGE: 1/4**
>
> **W1: The setting considered assumes knowledge of the environment dynamics by the HA-Teacher. System identification is proposed but does not appear to have been experimented with.**
>
> **Answer:** The dynamics of safety-critical autonomous systems have been extensively and well studied, allowing us to access the nonlinear dynamics models of many autonomous systems. For instance, dynamics models of quadruped robots, drones, and autonomous vehicles can be found in the works of Carlo et al. (2018), Beard (2008), and Rajamani (2011), respectively. System identification could be the solution in rare cases where a dynamics model cannot be obtained offline. However, this extreme scenario did not arise in our studies. Consequently, our original paper on HA-Teacher did not use system identification.
>
> During the rebuttal, we tested system identification in a 2D quadrotor. The biggest problem is that the obtained model is very sensitive to spurious correlations, leading to violations of physics laws and, consequently, unreliable HA-Teacher. So, through the experiment, we confirmed that the HA-Teacher in our runtime learning machine will always consider the dynamics models that are well-validated in mechanics. To address the potential confusion, we have removed the discussion of system identification in the remark and appendix.
>
> -----
>
> **W2: The presentation is cluttered at times and causes some difficulty in parsing. For example, some of the numerous assumptions, remarks, characteristics, and definitions should be consolidated if possible. Section 1.3 is almost entirely italicized.**
>
> **Answer:** We appreciate the reviewer's valuable suggestion, which greatly enhanced the paper's readability. We have made the necessary revisions accordingly. In the updated paper, we have merged two remarks, retained the assumption, and reverted the italicized fonts in Sections 1.3 and 6 to standard fonts.
>
> -----
>
> **Q1: How is this approach placed in the context of other shielding approaches [1,2]?**
>
> **Answer:** We appreciate the reviewer's insightful question and the valuable references that enhance the literature review of the revised paper.
>
> At a very high level, viewing from the critical modules **only**, we can conclude that our proposed runtime learning machine, the innovative model predictive shielding [NeurIPS 2024, ACC 2020], and the recently developed runtime assurance all represent novel variants of Simplex [8, 9], which is a fault-tolerant software architecture. Simplex consists of three main components: a high-performance module (HPM) that is not fully verified, a high-assurance module (HAM) which is verified and serves as a backup, and a monitor. We outline several vital historical developments in the field. The Simplex approach was proposed before 1999 and has been established to handle safety-critical systems. For instance, Rockwell Collins Inc. demonstrated that the Simplex principle could reduce the model checking time for a dual redundant flight control system from over 35 hours to less than 30 seconds. Today, machine learning (ML) has been integrated into many safety-critical autonomous systems; however, it introduces new safety challenges, such as ML hallucinations. Simplex provides a solution to these challenges and has been further developed into neural Simplex [10] and runtime assurance [11] in recent years. A notable recent advancement is the extension of the Simplex architecture for trustworthy autonomous systems, created by Turing laureate Joseph Sifakis [12].

---

> ### Author Response · Authors · 2024-11-18
> **Response to Reviewer xKGE: 2/4**
>
> Our runtime learning machine differs significantly from the model predictive shielding (MPS) frameworks [NeurIPS 2024, ACC 2020] due to varying motivations, safety concerns, and operating environments. Below, we detail the critical differences.
>
> 1. **Time-Critical Physical Operating Environment: Fast Response Needed.** The physical environments in which we operate our considered safety-critical autonomous systems—such as quadruped robots, drones, UAVs, and autonomous vehicles—are inherently **time-critical**. In these situations, even a millisecond can make a significant difference. When the system's real-time state $\mathbf{s}(k)$ approaches a safety boundary, we must make a safe decision immediately. Failing to do so could lead to a catastrophic outcome. For example, in the A1 demonstration video available at [A1 demo video (anonymous link)](https://www.dropbox.com/scl/fi/mjwi9u6ng72oghbm4wkx7/sim2real.mp4?rlkey=2lg4fayh2mrpj2m6nqvx02tox&st=ono7z5na&dl=0), we see that when a robot's real-time CoM height nears its safety lower bound, we need to activate the real-time HA-Teacher intervention to prevent the robot from falling, which could result in damage. Another real-life instance that highlights this critical nature is the 2022 US Central Illinois Freezing Rain incident. Over just 2.5 hours, more than 87 car crashes and spin-offs were reported, along with 29 rescue calls (not including slide-offs) made to first responders in Champaign City alone (https://www.wcia.com/news/state-police-urge-caution-as-freezing-rain-leads-to-several-crashes/). The current MPC used in MPS may encounter issues when a fast, time-critical response is required. Specifically, the MPC's nonlinear dynamics model can result in significant simulation time, leading to an un-ignorable delay in response. This delay can result in a real-time safety violation. Drawing inspiration from the lecture on LQR and MPC [13], one potential solution is to update the nonlinear MPC with a **real-time** and **state-dependent** linear MPC. This approach would eliminate the need for simulation during action planning/computing, allowing the CVXPY toolbox to be utilized directly for real-time recovery actions. In our real robot experiment, updating the real-time model and computing a real-time action policy can be done within 0.01 sec. Our future research direction will be the investigation of the real-time and state-dependent linear MPC.
>
> 2. **Non-Stationary Dynamics-Environment Interactions: Real-Time and State-Dependent Dynamics Model.** Our considered safety-critical autonomous systems, such as quadruped robots, drones, UAVs, and autonomous vehicles, have dynamics-environment interactions. This means the system dynamics can vary significantly between different environments, such as a sandy road, compared to an environment with freezing rain. One of the fundamental assumptions of model-based policies is the availability of a relatively accurate model of the underlying dynamics in consideration. While for the systems having system-environment interaction dynamics, a single off-line-built dynamics model cannot capture the differences in the dynamics induced by environmental variations. Moreover, in an unforeseen operating environment,  it is unreasonable to expect that the off-line-built models are sufficient to describe system dynamics accurately. Intuitively, when unforeseen environments cause safety violations in a time-critical environment, it is crucial to promptly update the dynamics models, action plans, and mission goals to ensure safe and effective responses in real time. We here conclude that for Simplex, runtime assurance, MPS, and our runtime learning machine, if their dynamics models for HAM are stationary, they will face the challenge of safety guarantee in non-stationary physical operating environments. These considerations inspire us to develop HA-Teacher, whose model knowledge, action policy, and mission goal are dynamic,  real-time, and state-dependent.
>
> 3. **Algorithm and Implementation.** Safety has two aspects: 1) the algorithm guarantees the satisfaction of all the applicable safety constraints, and 2) the verification of its implementation software. MPS is an innovative approach that can significantly improve the safety of RL algorithms. Our approach complements the MPS.   We compute a safety envelope for the systems using the verifiable policy.  A safety envelope is defined in the form of a Lyapunov function (a hyper ellipsoid) that satisfies all the applicable safety constraints.  If the system state approaches the envelope boundary, HA-Teacher will be activated.  We facilitate the learning process by embedding the Lyapunov function into the reward function and HA-Teacher's correction of unsafe actions and associated reward values in the replay buffer. The software of HA-Teacher is also verifiable. Thus, our runtime learning machine addresses both the algorithmic and implementation safety of real plants.

---

> ### Author Response · Authors · 2024-11-18
> **Response to Reviewer xKGE: 3/4**
>
> 4. **Verifiable Safety Requirements.** In the paper, we specify three essential requirements for designing a verifiably safe HA-Teacher. If these requirements are not met, we cannot trust HA-Teacher to correct unsafe learning of HP-Student and back up the safety of the real plants.  The requirements are 1) conformity with safety regulations,  2) conformity with operation regulations, and 3) attracting toward the safety envelope of HP-Student.
>
> **References**
>
> [8] Sha, L. (2001). Using simplicity to control complexity. IEEE Software, 18(4), 20-28.
>
> [9] Seto, D., \& Sha, L. (1999). An engineering method for safety region development. Carnegie Mellon University, Software Engineering Institute.
>
> [10] Phan, D. T., Grosu, R., Jansen, N., Paoletti, N., Smolka, S. A., & Stoller, S. D. (2020). Neural Simplex architecture. In NASA Formal Methods: 12th International Symposium, Moffett Field, CA, USA, May 11–15, 2020.
>
> [11] Brat, G., \& Pai, G. Runtime assurance of aeronautical products: preliminary recommendations. NTRS - NASA Technical Reports Server, 2023.
>
> [12] Sifakis, J., \& Harel, D. (2023). Trustworthy autonomous system development. ACM Transactions on Embedded Computing Systems, 22(3), 1-24.
>
> [13] LQR and MPC: https://people.smp.uq.edu.au/YoniNazarathy/Control4406/material/SlidesUnit6.pdf.
>
> -----
>
> **Q2: The triggering condition in Eq 7 is entirely causal (i.e., depends only on the current time and prior to the current time). Why is this better than using the environment model to predict for a future failure from the current state and input?**
>
> **Answer:** We thank the reviewer for the critical question. We would like to recall the partial **Answer** to **Q1** to answer it. Incorporating the prediction of future failure by the environment model will enable our runtime learning machine with an early warning function. We are actively seeking to achieve this. However, we face challenges due to the real-time unknown unknowns. Non-stationary and unforeseen physical operating environments are typical examples of unknown unknowns. So, we use the environment example to explain why we consider the **real-time** and **causal** triggering condition.
>
> Our considered safety-critical autonomous systems have dynamics-environment interactions. This means the system dynamics can vary significantly between different environments, such as a sandy road, compared to an environment with freezing rain. In non-stationary and unforeseen operating environments, it is unreasonable to expect that the offline-built environment models describe system dynamics accurately for trajectory predictions. The environment models are usually obtained by learning dynamics via DNN to make accurate predictions. However, an implicit assumption here is that the environments for generating the data of dynamics learning are stationary, and the data is sufficient. We can infer the assumption does not hold in non-stationary and unforeseen environments or when a real-time unknown occurs. The reason is rooted in characteristics of unknown unknowns: almost zero historical data, unpredictable timing, and distributions, resulting in the unavailability of models for scientific discoveries and understanding. In other words, the real-time unknown unknowns (including non-stationary and unforeseen environments) will create an imbalance and the out-of-distribution of training data.
>
> Obtaining or learning an accurate environment model in the presence of real-time unknown unknowns is a significant and open challenge. Solving this issue is essential for enabling our runtime learning machine to predict future failures and address safety concerns related to unknown unknowns.
>
> -----
>
> **Q3: How does the approach generalize to different safety sets (e.g., velocity < 1.5 vs velocity < 3)? Will the entire training process be needed for each change? How long would the training time be for each minor change?**
>
> **Answer:** We appreciate the reviewer's insightful questions. For a more straightforward explanation, let's define two mission goals for the robot: a velocity of 1.5 m/s and 3 m/s. Safety regulations establish the maximum allowable deviation from these mission goals. We only adjust the mission goals while keeping the maximum permissible deviation fixed. In that case, the HP-Student does not need to make any update. In contrast,  HA-Teacher's action policy is dynamic, state-dependent, and updated in response to real-time unknown unknowns and the Sim2Real gap. However, if the maximum allowable deviation changes, the HP-Student will recall the CVXPY toolbox to update the safety envelope according to the new safety regulations. This updated safety envelope will be shared immediately with the HA-Teacher for real-time adjustments to the patch design.

---

> ### Author Response · Authors · 2024-11-18
> **Response to Reviewer xKGE: 4/4**
>
> To effectively demonstrate this, we designed a challenging experiment. Initially, we pre-trained HP-Student for the A1 robot in the PyBullet simulator, using a flat terrain environment with a mission goal of 0.6 m/s. After this pre-training, we directly deployed HP-Student to the Go2 robot. We utilized NVIDIA Isaac Sim to create an operating environment for showcasing the Go2 robot's runtime learning capabilities. This environment transitions from flat terrain to uneven ground, further complicated by ice from unexpected freezing rain. We can summarize that the Go2 robot's operating environment differs significantly from the A1 robot's pre-training environment. Besides, as shown in the figure of A1 and Go 2 robots available at [A1Go2 (anonymous link)](https://www.dropbox.com/scl/fi/uqz37dibqfv4z4kosfth6/A1Go2.pdf?rlkey=mqhmg29zo9ht8z1bmsui66rzj&st=liwphjtb&dl=0), the two application systems are very different in their motors, weights, heights, etc. Lastly, we set very different mission goals for Go2's runtime learning machine: forward at velocity = 0.7 $m/s$ and backward at velocity = -0.7 $m/s$. The demonstration videos are available on [Go2 Forward (anonymous link)](https://www.dropbox.com/scl/fi/b1l7uskmx5wg2uw6n8tzz/forward.mp4?rlkey=6k4mfs1f8symff3ilquajsd3o&st=a17uvfo9&dl=0) and [Go2 Backward (anonymous link)](https://www.dropbox.com/scl/fi/fclb0npdyatq93qzq2i72/backward.mp4?rlkey=7u9bcmw3sz30zk6njqu79vtel&st=yje3jfwt&dl=0).
>
> Finally, we note that in our runtime learning machine, we do not have **training**; we only have **pre-training** (in a simulator and another domain) and **runtime learning**.  Runtime learning can be understood as safe continual learning (after pre-training) for a high-performance action policy in **real** plants  -- using **real**-time sensor data generated from **real**-time physical environments. While pre-training can help reduce the workload of runtime learning or test the operational mechanisms, it is not mandatory for the runtime learning process. If the pre-training is removed, the our runtime learning will be learning from scratch in the real plant. The experiment of the 2D quadrotor in Appendix I.2 of the updated paper is such an example.
>
> -----
>
> **Q4: Are there any experiments or examples with System Identification as mentioned in Remark 6.5 or Appendix C?**
>
> **Answer:** One example is the work in [14]. The dynamics of safety-critical autonomous systems have been extensively studied in the mechanical area, allowing us to access the dynamics models of many autonomous systems. For instance, dynamics models of quadruped robots, drones, and autonomous vehicles can be found in the works of Carlo et al. (2018), Beard (2008), and Rajamani (2011), respectively. In rare cases where a dynamics model cannot be obtained offline, system identification techniques could be a solution, but it depends on system complexity. However, this extreme scenario did not arise in our studies. Consequently, our original paper on HA-Teacher did not use system identification.
>
> During the rebuttal, we tested system identification in a 2D quadrotor. The biggest problem is that the obtained model is very sensitive to spurious correlations, leading to violations of physics laws and, consequently, unreliable HA-Teachet.  So, through this experiment, we confirmed that the HA-Teacher in our runtime learning machine will always consider the dynamics models that are well-validated in mechanics. To address the potential confusion, we have removed the discussion of system identification in the remark and appendix.
>
>
>
>
> Reference:
>
> [14] Mao, Y., et al. "SL1-Simplex: safe velocity regulation of self-driving vehicles in dynamic and unforeseen environments." ACM Transactions on Cyber-Physical Systems 7.1 (2023): 1-24.
>
> -----
>
> **-END-**

---

> > ### Comment · Reviewer_xKGE · 2024-11-26
> > **Thank you for the detailed response**
> >
> > My questions were answered and I especially appreciate the improvements to the clarity of the paper. I am inclined to increase my score but it is still a little difficult for me to clearly quantify the merits of the proposed method over related work on robustness to unknown environment factors and learning robust control of robots such as quadruped models [1].
> >
> > [1] Learning Quadruped Locomotion Using Differentiable Simulation, Song et al, CoRL 2024

---

> ### Author Response · Authors · 2024-11-22
> **Discussion deadline approaching**
>
> Dear Reviewer xKGE
>
> Thank you for your thoughtful reviews. We have addressed all of your comments in detail and updated the paper as necessary.
>
> The discussion period will conclude in a few days. Could you please review our responses and the updated paper? We would appreciate any follow-up questions you may have, and we are happy to provide timely answers.
>
> Kind regards,
>
> the Authors

---

> > ### Comment · Area_Chair_pxBR · 2024-11-24
> > **Please respond to the rebuttal ASAP**
> >
> > Dear reviewer,
> > The process only works if we engage in discussion. Can you please respond to the rebuttal provided by the authors ASAP?

---

> ### Author Response · Authors · 2024-11-25
> **The deadline for paper revisions is in 36 hours!**
>
> Dear Reviewer xKGE,
>
> We hope this message finds you well. The author-reviewer discussion phase has been extended, but only for discussions on minor issues. The deadline for revising the paper is within 36 hours. This means we cannot update the paper after 36 hours. Your timely feedback is thus crucial to us now.
>
> Could you please review our responses and the updated paper? We are always eager to hear your valuable feedback and provide further responses.
>
>
> Warm regards,
>
> Authors

---

> ### Author Response · Authors · 2024-11-26
> **Thank you for your encouraging feedback!**
>
> Dear Reviewer xKGE,
>
> We appreciate your encouraging feedback for us! The experiments in the paper demonstrate the robustness of our proposed runtime learning machine. However, we do not formally claim it is a notable feature in the current paper. There are several critical reasons:
>
> 1. The paper has claimed and succesfully demonstrated four notable features of proposed runtime learning machine: 1) assuring lifetime safety, 2) tolerating real-time unknown unknowns, 3) addressing the Sim2Real gap, and 4) an automatic hierarchy learning mechanism. All of them are centered around the motivations, identified safety challenges, and design aims of the paper. However, robustness is not very directly related to the paper's research problems.
>
> 2. We plan to schedule the robustness studies as our future in-depth and comprehensive studies. This is because there are many open and significant problems here, so it is necessary to develop a second paper. For example, do conflicts exist between safety assurance, mission and operation performance, and robustness? If so, how should the conflicts be managed? Can we design the HA-Teacher to have verifiable robustness (utilizing the verified adaptive and robust controller, e.g., L1 adaptive controller) to teach the DRL agent to learn to gain a safe, robust, and high-performance action policy quickly? We believe our nearby paper will answer the problems formally.
>
> 3. The current paper is quite long, **over 42 pages**. If it includes theoretical studies and experiments related to robustness, they must be moved to the new appendices.
>
> Finally, we thank the reviewer for recommending the reference related to robustness and quadruped robots, which will enrich the paper's Conclusion and Discussion section about future robustness studies.
>
>
> Warm regards,
>
> Authors

---

### Official Review · Reviewer_Z8i5 · 2024-11-01

**Soundness:** 3
**Presentation:** 2
**Contribution:** 2
**Rating:** 5
**Confidence:** 3

**Summary:**

The paper presents a novel approach to runtime learning for agents, guided by a verified teacher during runtime to ensure safe operation. This teacher-student framework leverages physics-regulated deep reinforcement learning (PhyRL) and aims to address safety concerns arising from unknown variables and Sim2Real gaps. The validation of the approach is conducted on two benchmarks, including a real quadruped robot, which adds to the practical value of this research.

**Strengths:**

1. The paper addresses a important safety issue in learning-based systems, particularly focusing on challenges like unknown unknowns and Sim2Real gap.
2. The paper presents solid theorem.
3. The evalution is performed on a real robot, which exhibits the practicality of this work.

**Weaknesses:**

Though the results look impressive, I found some parts of the paper needs further clarification and a few more discussion about related works are expected.
1. The paper lacks details on the verifiability of the PhyRL structure, especially regarding how the teacher's safety is guaranteed in unknown environments. In Equation 2, the system safety is only related to its state. Does this mean that this paper assumes such simple constraint and not consider how the environment change impacts on the states?
2. The paper should discuss whether switching back to the teacher affects operational performance, as such switches may potentially degrade it. Additionally, the evaluation would benefit from presenting metrics on how frequently the teacher’s guidance is triggered.
3. The work currently assumes a single-step safety model, where each action ensures safety only for the immediate next state. However, a broader perspective on end-to-end safety—considering the effects over several steps or even entire trajectories—is the cases that usually happen in reality.
4. There are lots of verified learning works that are not cited/discussed by this paper.  For example, Neurosymbolic reinforcement learning with formally verified exploration. Neurips 2020.
5. There are only two benchmarks presented. Though one of them is the real robot, it would be necessary to evaluate on a more diverse set of benchmarks.

**Questions:**

1. How often does the teacher’s intervention occur during runtime, and is there an impact on the agent's performance with frequent switches?
2. If the teacher is adjusted or modified during runtime, how is its performance or reliability guaranteed in changing conditions?

**Details Of Ethics Concerns:**

/

---

> ### Author Response · Authors · 2024-11-15
> **Response to Reviewer Z8i5: 1/4**
>
> **W1: The paper lacks details on the verifiability of the PhyRL structure, especially regarding how the teacher's safety is guaranteed in unknown environments. In Equation 2, the system safety is only related to its state. Does this mean that this paper assumes such simple constraint and not consider how the environment change impacts on the states?**
>
> **Answer:** We appreciate the reviewer's questions. Many safety-critical autonomous systems, such as quadruped robots, drones, UAVs, and autonomous vehicles, have dynamics-environment interactions. This means that the system dynamics can vary significantly between different environments, such as a sandy road compared to an asphalt road. For instance, in the case of quadruped robots and autonomous vehicles, road friction forces directly affect the acceleration of their movements, which in turn influences their velocities and positions. More details on this topic can be found in the book [Rajamani (2011), Vehicle Dynamics and Control]. In addition, the real plants serve as our controlling objects, and we cannot manipulate external environments. Real-time states provide direct observations of whether the system is safe. For instance, the ratio of a vehicle's wheel velocity to its longitudinal velocity can indicate whether it is slipping or sliding on icy roads. Similarly, a robot's center of mass (CoM) height can reveal whether it is about to fall. These explain that our safety set in Equation (2) and the safety definition are expressed as the system states.
>
> Lastly, the HA-Teacher’s dynamics knowledge, represented by $(\mathbf{A}(\mathbf{s}(k)), \mathbf{B}(\mathbf{s}(k)))$, is dependent on the real-time state samplings. As a result, the HA-Teacher's action policy is also real-time in response to real-time and dynamic environments. We can conclude that our resulting runtime learning machine is adaptive to the operating environments to ensure safety. To convincingly demonstrate this. We designed a challenging experiment. Specifically, we first pre-trained Phy-DRL well for the A1 robot in the **PyBullet** simulator, and its environment is a plat terrain with high friction. We then directly employed the well-pre-trained Phy-DRL for the A1 robot as the HP-Student in the runtime learning machine for the Go2 robot. We use the **NVIDIA Isaac Sim** to build an operating environment for the Go2 robot. The operating environment transitions from flat to uneven and unstructured terrain, covered with ice caused by unforeseen freezing rain. We conclude that the Go2 robot's operating environment differs significantly from the A1 robot's training and testing environments. Besides, as shown in the figure available at [A1Go2 (anonymous link)](https://www.dropbox.com/scl/fi/uqz37dibqfv4z4kosfth6/A1Go2.pdf?rlkey=mqhmg29zo9ht8z1bmsui66rzj&st=liwphjtb&dl=0), A1 and Go robots are very different in their motors, weights, heights, etc. The demonstration videos are available at [Go2 Forward (anonymous link)](https://www.dropbox.com/scl/fi/b1l7uskmx5wg2uw6n8tzz/forward.mp4?rlkey=6k4mfs1f8symff3ilquajsd3o&st=a17uvfo9&dl=0),  [Go2 Backward (anonymous link)](https://www.dropbox.com/scl/fi/fclb0npdyatq93qzq2i72/backward.mp4?rlkey=7u9bcmw3sz30zk6njqu79vtel&st=yje3jfwt&dl=0), and [Go2: Kick--Sensor (anonymous link)](https://www.dropbox.com/scl/fi/glqv1o9viltxvp083vmdt/sensorkick.mp4?rlkey=cj06mdbrdfnmcdqr688jjn38k&st=u0cs6312&dl=0). The video links have been added to the revised paper.
>
> -----
>
> **W2: The paper should discuss whether switching back to the teacher affects operational performance, as such switches may potentially degrade it. Additionally, the evaluation would benefit from presenting metrics on how frequently the teacher’s guidance is triggered.**
>
> **Answer:** We want to thank the reviewer for the questions. We think the reviewer's questions are about **Mission Performance v.s. HA-Teacher**. Our paper defines operation performance as conformity with operation regulations (i.e., confining the real-time actions within a physically feasible bounded action space). In our runtime learning machine, HP-Student's capability of achieving this is innate through the sigmoid and clipper functions. The HA-Teacher adheres to operation regulations through its design, mathematically expressed in Item 3 of Theorem 6.7 in the paper. The formal proof can be found in Appendix D. 2.3.

---

> ### Author Response · Authors · 2024-11-16
> **Response to Reviewer Z8i5: 2/4**
>
> Our paper includes various such studies, with evaluation metrics and experimental results. For instance, Appendix E offers theoretical guidance on selecting the teacher's dwell time. In Appendix G 5.3, we define the evaluation metric as  $\text{HA-Teacher's activation ratio} = \frac{\text{HA-Teacher's total dwell/activation time in one episode}}{\text{one episode length}} \in (0, 1)$, which is used to demonstrate the automatic hierarchy learning mechanism and describe the HA-Teacher's switching behavior. Appendix G 5.3 also includes the experimental results. However, due to the page limit, the 10-page content mainly demonstrates the mechanism of automatic hierarchy learning. Others were in the appendices, which impressed the reviewer that the paper lacked such work. We address the issue by highlighting the connection sentences in the teal color in the revised paper.
>
> In our runtime learning machine, HP-Student and HA-Teacher are highly interactive. HP-Student shares his safety regulations and envelope with the HA-Teacher for his real-time patch design. HA-Teacher corrects HP-Student's unsafe learning and backs up the safety of the real plants. Meanwhile, the Coordinator monitors the real-time safety status and facilitates interactions between students and teachers. Specifically, when the real-time safety status of the real plant being controlled by HP-Student approaches the safety boundary, the Coordinator **automatically** prompts the HA-Teacher to intervene, assure the safety of the real plant, and correct HP-Student's unsafe learning. When the real-time states return to a safe region, the Coordinator **automatically** triggers the switch back to HP-Student and terminates the learning correction.
>
> We can conclude that the frequency of triggering HA-Teacher depends on HP-Student's capability of assuring safety in the face of the Sim2Real gap and unknown unknowns. Taking addressing the Sim2Real gap as one example, HA-Teacher is triggered frequently in the initial stage. After runtime learning in the real-time physical environment for a while, HP-Student will gain the capabilities of guaranteeing safety and addressing the Sim2Real gap. His continual runtime learning seldom triggers HA-Teacher and automatically self-learns for a high-performance action policy. These have been witnessed by the trajectories of HA-Teacher's activation ratio and the system's phase plots available at [HA-Teacher (anonymous link)](https://www.dropbox.com/scl/fi/ewgbx0m6hc1xuawiv7jo4/HA-Teacher.pdf?rlkey=ha509lx9grd5d5qvm30ok5prm&st=7lbozvq5&dl=0) (also, Figures 4 and 14 in the paper).  Another phase plot available at [HA-Teacher v.s. HP-Student (anonymous link)](https://www.dropbox.com/scl/fi/j70n27wtyufmjgrzi4tn8/2DHAHP.pdf?rlkey=si9ifb43ac6d6z3s8whyjq45g&st=bdjmfqgu&dl=0) (also, Figure 21 in the paper) can also well illustrate this.
>
>
> -----
>
> **W3: The work currently assumes a single-step safety model, where each action ensures safety only for the immediate next state. However, a broader perspective on end-to-end safety—considering the effects over several steps or even entire trajectories—is the cases that usually happen in reality.**
>
> **Answer:** We appreciate the reviewer's critical question. We would like to clarify that our definition of safety is **real-time** (or state step-wise) and **deterministic**. Specifically, the Coordinator in our runtime learning machine monitors the real-time state, denoted as $\mathbf{s}(k)$. When the real-time states of the physical system being controlled by HP-Student approach or exceed the boundary of safety envelope (i.e., the $\mathbf{s}^T(k-1) \cdot \mathbf{P} \cdot \mathbf{s}(k-1) \leq 1$ and $\mathbf{s}^\top(k) \cdot \mathbf{P} \cdot \mathbf{s}(k) > 1$ occur), the Coordinator immediately activates the HA-Teacher to back up safety and correct unsafe learning conditions of the HP-Student.  Furthermore, if there is any instance $k$ at which the real-time state $\mathbf{s}(k)$ falls outside the safety set $\mathbb{X}$ (i.e., $\mathbf{s}(k) \notin \mathbb{X}$), the system is considered unsafe according to our safety definition.

---

> ### Author Response · Authors · 2024-11-16
> **Response to Reviewer Z8i5: 3/4**
>
> We impose higher requirements or constraints on system states than end-to-end safety definitions (considering the effects over several steps or even entire trajectories). We require the real-time system states to be constrained into the safety set at every step rather than probabilistic safety with high confidence. The critical motivation behind our higher requirements is that many safety-critical autonomous systems, such as quadruped robots, drones, UAVs, and autonomous vehicles, are also **time-critical**. For such systems, a millisecond can make a difference. Once the real-time state $\mathbf{s}(k)$ approaches the safety boundary, we must immediately make our safe decision. Otherwise, a disaster can happen.  This can be viewed from the A1 demonstration video available at [A1 demo video (anonymous link)](https://www.dropbox.com/scl/fi/mjwi9u6ng72oghbm4wkx7/sim2real.mp4?rlkey=2lg4fayh2mrpj2m6nqvx02tox&st=ono7z5na&dl=0). When the robot's real-time CoM height approaches its safety lower-bound, if we cannot immediately trigger real-time HA-Teacher to back up safety, the robot will drop, causing damage.
>
> Another real-life example that underlines the underlying time-critical characteristic is the 2022 US Central Illinois Freezing Rain: in the 2.5 hours,  more than 87 car crashes and spin-offs were reported, and 29 rescue calls (not including slide-offs) were called into the first responders in the Champaign City only [8].
>
>
> **Reference**
>
> [8] C. Smith, “State police urge caution as freezing rain leads to several crashes,” WCIA. Available: https://www.wcia.com/news/state-police-urge-caution-as-freezing-rain-leads-to-several-crashes/.
>
> -----
>
>
>
> **W4: There are lots of verified learning works that are not cited/discussed by this paper.  For example, Neurosymbolic reinforcement learning with formally verified exploration. Neurips 2020.**
>
> **Answer:** We agree with the reviewer that significant efforts have been devoted to DRL with verifiable safety. However, different approaches often have varying motivations, research focuses, and definitions of safety. In our Related Work section of the paper, the subsection dedicated to Safe DRL provides a literature review of **12** papers. Our selection criteria focused on the definition of safety, which, like our approach, is real-time and deterministic. The subsection on fault-tolerant DRL reviews **6** papers, with selection criteria based on their ability to support runtime learning and safety verification in real-world applications while ensuring safety. However, we could not compare all of these studies in our experiments due to the page limitation. Instead, we focused on state-of-the-art work. Specifically, we compared CLF-DRL (**CoRL 2022**) and Phy-DRL (**ICLR 2024**) in safe DRL. For fault-tolerant DRL, we compared Neural Simplex (**NASA Formal Methods Symposium, 2020**) and runtime assurance (**ICRA 2022; NASA Technical Reports, 2023**). Additionally, we incorporated the approaches of delay randomization (**ICLR 2022**) and domain randomization into the comparison models for addressing Sim2Real and domain gaps.
>
> We thank the reviewer for recommending the seminal work in [Neurosymbolic reinforcement learning with formally verified exploration. Neurips 2020]. However, our differing motivations have led us to develop distinct frameworks. To the best of our knowledge, the aforementioned work does not yet address the following open problems: the motivations behind developing our runtime learning machine.
>
> 1. The Sim2Real challenge raises significant safety concerns for inference models used in real plants. Can the developed safety verification procedure be applied in safety-critical settings to address the Sim2Real gap? If so, does the data-driven verification procedure require exhaustive samplings to obtain a verified safe model?
>
> 2. Due to DNN's colossal parameter space, intractable activation functions, and random learning factors, DRL's behavior or functions are hard to analyze and verify. DRL's real-time hallucinations are thus unavoidable. During verification in real plants, how can we guarantee the safety of real plants once a hallucination occurs?
>
> 3. Many safety-critical autonomous systems operate in outdoor, dynamic, and unpredictable environments. How can the safety of real plants be ensured when safety verification is performed in such real environments?
>
> The work presented in [Neurosymbolic reinforcement learning with formally verified exploration. Neurips 2020] offers valuable insights for the further development of our runtime learning machine. For instance, the proposed reachability through worst-case dynamics could serve as a potential solution for incorporating an early warning function into our runtime learning machine. We have included this discussion in the revised paper.
>
> -----

---

> ### Author Response · Authors · 2024-11-17
> **Response to Reviewer Z8i5: 4/4**
>
> **W5: There are only two benchmarks presented. Though one of them is the real robot, it would be necessary to evaluate on a more diverse set of benchmarks.**
>
> **Answer:** To address safety challenges posed by unknown unknowns and the Sim2Real gap, we have designed our runtime learning machine to achieve the **Prospect:** Runtime learning for an effective action policy in real environments—utilizing real-time sensor data generated from actual physical conditions while prioritizing safety.  Meanwhile, the proposed runtime learning machine notably features i) assuring lifetime safety (i.e., safety guarantee in any runtime learning stage, regardless of HP-Student's success), ii) tolerating real-time unknown unknowns, iii) addressing Sim2Real gap, and iv) automatic hierarchy learning (i.e., safety-first learning, and then high-performance learning for high-performance action policy). Initially, we were able to provide more benchmarks; however, we need to demonstrate that we are achieving the prospect along with all four claimed features, which requires additional pages. Due to the page limit, the benchmarks in our original paper were reduced to a cart-pole system and the real A1 quadruped robot. In this way, partial experimental results still need to be moved to the appendices in the original paper.
>
> Following the reviewer's suggestion, we added two benchmarks to Appendix I in the revised paper. The new benchmarks include waypoint tracking of a 2D quadrotor and locomotion of the Go2 quadruped robot in dynamic, unforeseen, uneven, and unstructured environments.
>
> -----
>
>
> **Q1: How often does the teacher’s intervention occur during runtime, and is there an impact on the agent's performance with frequent switches?**
>
> **Answer:** Coordinator monitors the real-time safety status and facilitates interactions between students and teachers. Specifically, when the real-time safety status of the real plant controlled by the HP-Student approaches the safety boundary, the Coordinator automatically prompts the HA Teacher to intervene, ensuring the safety of the real plant and correcting HP-Student's unsafe learning. When the real-time states return to a safe region, the Coordinator automatically triggers the switch back to HP-Student and terminates the learning correction. We can conclude that HP-Student's capability of assuring safety (in the face of the real-time Sim2Real gap and unknown unknowns) controls the frequency of triggering HA-Teacher.
>
> Taking addressing the Sim2Real gap as one example, when HP-Student was first deployed to a real plant, HA-Teacher was triggered frequently in the initial stage due to the significant Sim2Real gap. After some time spent (runtime) learning in a real-time physical environment, the gap is being addressed, and the HP-Student's ability to ensure safety is improving. As he masters this capability, his continuous runtime learning rarely activates the HA-Teacher, allowing him to automatically self-learn a high-performance action policy. These can be viewed from the trajectories of HA-Teacher's activation ratio and the system's phase plots available at [HA-Teacher (anonymous link)](https://www.dropbox.com/scl/fi/ewgbx0m6hc1xuawiv7jo4/HA-Teacher.pdf?rlkey=ha509lx9grd5d5qvm30ok5prm&st=7lbozvq5&dl=0) (also, Figures 4 and 14 in the paper). The phase plot available at [HA-Teacher v.s. HP-Student (anonymous link)](https://www.dropbox.com/scl/fi/j70n27wtyufmjgrzi4tn8/2DHAHP.pdf?rlkey=si9ifb43ac6d6z3s8whyjq45g&st=bdjmfqgu&dl=0) (Figure 21 in the paper) is also an illustration experiment.
>
> -----
>
> **Q2: If the teacher is adjusted or modified during runtime, how is its performance or reliability guaranteed in changing conditions?**
>
> **Answer:** This question is very critical. HA-Teacher's designs, including his dynamics model, patch, action policy, and mission/control goal, are not static or fixed. They are dynamic, real-time, and state-dependent. Below is a summary of the motivation.
>
> Enabling runtime learning in real plants is straightforward in addressing the Sim2Real gap, but not so for unknown unknowns because unknown unknowns lack historical data and cannot be predicted in time and distribution. When an unknown unknown creates safety issues in a time-critical environment, it is crucial to promptly update the dynamics models, action plans, and mission goals to ensure safe and effective responses in real time. The insight inspires us to develop the real-time patch as the HA-Teacher. Its model knowledge, action policy, and mission goal are dynamic and real-time.
>
> The designs of HA-Teacher are formally outlined in Theorem 6.7 of the paper. In Appendix D, we provide a mathematical proof that demonstrates HA-Teacher meets three crucial requirements: 1) compliance with safety regulations, 2) adherence to operational regulations, and 3) alignment with the safety envelope of HP-Student. This guarantees that the actions taken by HA-Teacher are verifiably safe.
>
> -----
>
> **-END-**

---

> ### Author Response · Authors · 2024-11-22
> **Discussion deadline approaching**
>
> Dear Reviewer Z8i5
>
> Thank you for your thoughtful reviews. We have addressed all of your comments in detail and updated the paper as necessary.
>
> The discussion period will conclude in a few days. Could you please review our responses and the updated paper? We would appreciate any follow-up questions you may have, and we are happy to provide timely answers.
>
> Kind regards,
>
> the Authors

---

> > ### Comment · Area_Chair_pxBR · 2024-11-24
> > **Please respond to rebuttal ASAP**
> >
> > Dear reviewer,
> > The process only works if we engage in discussion. Can you please respond to the rebuttal provided by the authors ASAP?

---

> ### Author Response · Authors · 2024-11-25
> **The deadline for paper revisions is in 36 hours!**
>
> Dear Reviewer Z8i5,
>
> We hope this message finds you well. The author-reviewer discussion phase has been extended, but only for discussions on minor issues. The deadline for revising the paper is within 36 hours. This means we cannot update the paper after 36 hours. Your timely feedback is thus crucial to us now.
>
> Could you please review our responses and the updated paper? We are always eager to hear your valuable feedback and provide further responses.
>
>
> Warm regards,
>
> Authors

---

> > ### Comment · Reviewer_Z8i5 · 2024-11-26
> > **Response**
> >
> > Thank you for the detailed explanation and additional experiments.
> >
> > I appreciate the discussion about work motivation: 'Many safety-critical autonomous systems operate in outdoor, dynamic, and unpredictable environments. How can the safety of real plants be ensured when safety verification is performed in such real environments?'
> >
> > - I am still confused about how this work's teacher's performance is theoretically assured and verified.
> > - The state step-wise safety is weak from the reviewer's point of view. How the long-term performance & safety is theoretically influenced by the step-wise state safety is still not clear.

---

> ### Author Response · Authors · 2024-11-26
> **Response To reviewer Z8i5's new question**
>
> We thank the reviewer for the new questions. We note such outdoor systems usually have dynamics-environment interactions. When dynamic and unpredictable environments create a safety violation in a time-critical environment. As a real-time response, HA-Teacher is proposed to immediately use the most recent sensor data to update the dynamics model, and the updated model will be immediately used to compute a real-time patch and action policy.
>
>
> -----
>
> **Verifiable Safety:** We first recall that the patch in the paper is the designed invariant state space of HA-Teacher's action policy under the assumption of local Lipschitz on model mismatch:
>
> **Assumption:** The model mismatch $\mathbf{h}(\cdot)$ is locally Lipschitz in the patch $\Psi_k$, i.e.,
>
> $(\mathbf{h}(\bf{e}_1) - \bf{h}(e_2))^T \cdot \widehat{\mathbf{P}}_k \cdot (\mathbf{h}(\bf{e}_1) - \bf{h}(e_2)) \le \kappa  \cdot (\bf{e}_1 - e_2)^T \cdot \widehat{\mathbf{P}}_k \cdot (\bf{e}_1 - e_2)$ holds for any $\bf{e}_1,  \bf{e}_2 \in \Psi_k.$
>
> Our Theorem 6.4 in the updated paper is the theoretical support or foundation of the HA-Teacher design. In summary, if the assumption is verified to hold in the real plant, i.e., the computed real-time action policy by Theorem 6.4 can **always** constrain the real-time states into the real-time patch. **Furthermore, the real-time patch is always a subset of a safety set**. So, according to Theorem 6.4, system systems are always contained to the safety set, such that safety is guaranteed. We note the derived verifiable safety, or Theorem 6.4, is under the assumption of local Lipschitz. So, to have a verified safe action policy, we have two steps.
>
> **Step 1:** Before the design, we must verify that given a Lipschitz upper bound  $\kappa$, if the assumption of local Lipschitz holds by using the system's dynamics model:
>
> $\mathbf{e}(k+1) = \mathbf{A}(\mathbf{s}(k)) \cdot \mathbf{e}(k) + \mathbf{B}(\mathbf{s}(k)) \cdot \mathbf{a}_{\text{HA}}(k) + \mathbf{h}(\mathbf{e}(k)),$
>
> where  $\mathbf{e}(k+1)$, $\mathbf{e}(k)$, $\mathbf{B}(\mathbf{s}(k))$, $\mathbf{a}_{\text{HA}}(k)$, $\mathbf{B}(\mathbf{s}(k))$ are known, so that model mismatch $\mathbf{h}(\mathbf{e}(k))$ can be computed by the model.
>
> **Step 2:** If the assumption holds, we output the Lipschitz upper bound to Theorem 6.4 for computing an action policy. The action policy thus has verified safety according to our theory. We note that **if the assumption is verified not to hold, Theorem 6.4 can still compute an action policy for us but does not have verifiable safety by our theory.**
>
> -----
>
> **Step-wise Safety and Long-term Safety:** The authors may have misunderstood the reviewer's previous question. We may also have different definitions of **step-wise**. We regard **step-wise** as **real-time**. For example, a real plant's step-wise state s(k) is the state sampling at real-time k.   If the reviewer's definition of **long-term** is the whole trajectory. Our step-wise safety is also long-term safety. To explain this, we recall our formal safety definition from the paper:
>
> **Lifetime Safety:** Consider the safety set $\mathbb{X}$. The real plant is said to have lifetime safety if given any $\mathbf{s}(1) \in \mathbb{X}$, the $\mathbf{s}(k) \in  \mathbb{X}$ holds at any time $k \in \mathbb{N}$, regardless of HP-Student's failure.
>
> The ``any time $k \in \mathbb{N}$" can be understood as long-term or the entire trajectory because if safety can be guaranteed at any step-wise $k$, the safety of the whole trajectory is assured. So, according to our definition, long-term safety is also not assured if step-wise safety is not guaranteed.
>
> If long-term safety needs trajectory prediction in the face of unknown unknowns, it will exceed the scope of this paper. The reason is rooted in characteristics of unknown unknowns: almost zero historical data, unpredictable timing, and distributions, resulting in the unavailability of prediction models.

---

> > ### Comment · Reviewer_Z8i5 · 2024-11-26
> > **Response**
> >
> > Thank you for your explanation about the local Lipschitz assumption. My concern about the teacher's performance is addressed.
> >
> > However, I am still concerned about the limitation of the step-wise safety. Thus, I will keep my score unchanged.

---

> ### Author Response · Authors · 2024-11-26
> **To Reviewer Z8i5's question. Lifetime safety, not step-wise safety.**
>
> Dear Reviewer Z8i5,
>
> To address your last concern, we want to clarify that **step-wise safety** is not our safety definition and design aim. Ours is **life-time safety!**
>
> First of all, our runtime learning machine needs to monitor the real-time safety metric $\mathbf{s}^\top(k) \cdot \mathbf{P} \cdot \mathbf{s}(k)$ to decide if trigger HA-Teacher for backup safety. This condition impressed you that our theoretical safety guarantee or definition is ``step-wise", i.e., guaranteeing safety for only one step. **Definition 2.1 and Theorem 6.4 of the paper clearly and formally state this is untrue!** To demonstrate this, we recall them from paper.
>
> **Definition 2.1 (Lifetime Safety).** Consider the safety set $\mathbb{X}$. The real plant is said to have lifetime safety, if given any $\mathbf{s}(1) \in \mathbb{X}$, the $\mathbf{s}(k) \in  \mathbb{X}$ holds at any time $k \in \mathbb{N}$, regardless of HP-Student's failure.
>
> **Statement 1 in Theorem 6.4.** The real-time patch ${\Psi_{\sigma(k)}} \subseteq {\mathbb{X}}$ holds for any time $k$.
>
> Definition 2.1 specifies one of our design aims: rendering the given safety set $\mathbb{X}$ invariant by our learning machine, that is, under the control of our runtime learning machine, operating from the safety set, system states will never leave the safety set at any current or future time. This design aim is mathematically described by 'given any $\mathbf{s}(1) \in \mathbb{X}$, the $\mathbf{s}(k) \in  \mathbb{X}$ holds at any time $k \in \mathbb{N}$' in Definition 2.1. Lifetime safety is achieved, as indicated by Statement 1 in Theorem 6.4. We note that 'holds at any time $k \in \mathbb{N}$' in Definition 2.1 and Statement 1 in Theorem 6.4 explicitly state that our safety definition is lifetime, not step-wise.
>
> During real operation, we must constantly monitor the real-time system states to decide if to trigger HA-Teacher for backup safety. The triggering condition is $\mathbf{s}^\top(k-1) \cdot \mathbf{P} \cdot \mathbf{s}(k-1) \leq 1$ and $\mathbf{s}^\top(k) \cdot \mathbf{P} \cdot \mathbf{s}(k) > 1$. The condition is only used to trigger the switching.  If we only consider safety assurance, i.e., not caring about mission performance, we can always use HA-Teacher. The real-time monitor can be removed because its design ensures safety for a **lifetime, not step-wise**, as indicated by Theorem 6.4 of the revised paper.
>
> In conclusion, our safety definition is **lifetime** and **deterministic**, which has been a standard design aim for many other safety- and time-critical autonomous systems, such as stated by Definition 2.1 in [R1], Definitions 2.3 and 2.4 in [R2], Section 2 in [R3], Definition 1 in [R4].
>
> Importantly, lifetime safety is our claimed notable feature of runtime learning machine, which has been well demonstrated by experimental results in **Figures 3 (a) and (b)**, **Figures 9 to 12**, and **Figure 21** (or the plot available at [HA-Teacher v.s. HP-Student (anonymous link)](https://www.dropbox.com/scl/fi/j70n27wtyufmjgrzi4tn8/2DHAHP.pdf?rlkey=si9ifb43ac6d6z3s8whyjq45g&st=bdjmfqgu&dl=0)) of the updated paper.
>
> **Lastly, we believe the reviewer’s concern can be further addressed by the (safety) demonstration videos available at [A1 Robot: unknown unknowns (anonymous link)](https://www.dropbox.com/scl/fi/jjxlym8xtdxqgr7d5hclp/unk.mp4?rlkey=kh76ojwhspqgslzolmowuuu0l&st=igbjnji3&dl=0), [A1 Robot: Sim2Real gap (anonymous link)](https://www.dropbox.com/scl/fi/mjwi9u6ng72oghbm4wkx7/sim2real.mp4?rlkey=2lg4fayh2mrpj2m6nqvx02tox&st=ono7z5na&dl=0), [Go2 Forward (anonymous link)](https://www.dropbox.com/scl/fi/b1l7uskmx5wg2uw6n8tzz/forward.mp4?rlkey=6k4mfs1f8symff3ilquajsd3o&st=a17uvfo9&dl=0),  [Go2 Backward (anonymous link)](https://www.dropbox.com/scl/fi/fclb0npdyatq93qzq2i72/backward.mp4?rlkey=7u9bcmw3sz30zk6njqu79vtel&st=yje3jfwt&dl=0), and [Go2: Kick--Sensor (anonymous link)](https://www.dropbox.com/scl/fi/glqv1o9viltxvp083vmdt/sensorkick.mp4?rlkey=cj06mdbrdfnmcdqr688jjn38k&st=u0cs6312&dl=0).**
>
> **References**
>
> [R1] Cao et al. Physics-Regulated Deep Reinforcement Learning: Invariant Embeddings. In The Twelfth International Conference on Learning Representations, 2024.
>
> [R2] Seto, D., \& Sha, L. An engineering method for safety region development. Carnegie Mellon University, Software Engineering Institute.
>
> [R3] Bak et al. Real-time reachability for verified simplex design. In 2014 IEEE Real-Time Systems Symposium. IEEE.
>
> [R4] Tabas, D., \& Zhang, B. Computationally efficient safe reinforcement learning for power systems. In 2022 American Control Conference.

---

> ### Author Response · Authors · 2024-12-01
> **Updated response to the last concern.**
>
> Dear Reviewer Z8i5,
>
> We hope you are having a lovely weekend.
>
> We have updated our response to your last concern. Could you please take a moment to review it and reconsider the rating?
>
> Regards,
>
> The Authors

---

### Official Review · Reviewer_mpnQ · 2024-11-06

**Soundness:** 3
**Presentation:** 3
**Contribution:** 2
**Rating:** 6
**Confidence:** 3

**Summary:**

This paper introduces a "Runtime Learning Machine" designed for autonomous systems operating in safety-critical environments. The machine combines three main components: an HP-Student (high-performance learner), an HA-Teacher (high-assurance safety monitor), and a Coordinator. The HP-Student learns in real-time from physical environments, with the HA-Teacher correcting unsafe actions and enforcing safety. This design aims to address critical challenges in deep reinforcement learning, including tolerating unknown unknowns and bridging the Sim2Real gap.

**Strengths:**

The paper presents an innovative hierarchical safety mechanism where the HA-Teacher manages safety while the HP-Student prioritizes performance optimization within safety constraints. This interactive approach is practical and well-suited to the paper’s focus on real-world, safety-critical applications. The experimental setup, involving both a cart-pole system and a quadruped robot, is comprehensive and shows the method’s robustness across different unknowns. The results suggest that the system can maintain stability through real-time patching by the HA-Teacher, showing practical value in environments where safety and performance trade-offs are essential. Additionally, the framework addresses the Sim2Real gap, which remains a significant challenge in deploying DRL systems in physical settings.

**Weaknesses:**

The design details for the teacher, a core element of this framework, could be expanded, particularly in scenarios where unknowns extend beyond simple disturbances to include hardware or sensor faults. Additionally, the theoretical justification for the teacher’s resilience against unknown unknowns could be strengthened, as the current analysis lacks formal rigor or quantitative measures. While the selected experimental domains are relevant, further discussion on generalizability to applications like autonomous driving or complex industrial robots would broaden the framework’s impact. Although comparisons with other DRL methods are made, the analysis could be enhanced by discussing specific scenarios where other methods may outperform this framework or identifying potential limitations.

**Questions:**

Could you provide more insights into the teacher's handling of complex unknowns, such as hardware failures or significant sensor errors?

What theoretical guarantees exist for the teacher's performance in completely unanticipated scenarios?

How might this system perform in applications outside of the tested environments, such as in higher dimensional or more complex environments?

---

> ### Author Response · Authors · 2024-11-13
> **Response to Reviewer mpnQ: 1/5**
>
> **W1: The design details for the teacher, a core element of this framework, could be expanded, particularly in scenarios where unknowns extend beyond simple disturbances to include hardware or sensor faults.**
>
> **Answer:** We appreciate the valuable suggestion from the reviewer. Our system's sensors generate two types of data: samplings of the system state (including velocities, positions, roll, yaw, pitch, etc.) and samplings of control commands. In our experiments with the cart-pole system and the real A1 quadruped robot, we introduced faults in the controller sensors by injecting unknown-unknown noise. This noise was generated according to a randomized Beta distribution, explained in detail in Appendix F of the paper as one method for representing unknown unknowns. Initially, we did not intentionally induce faults in the system-state sensors because the sensor readings are inherently noisy. Besides, the real robots' center of mass (CoM) positions are estimated using the Kalman filter that relies on velocity samplings, making estimation errors unavoidable.
>
> In the new experiment of the Go2 robot, we introduced system-state sensor faults by injecting unknown-unknown noise. The demonstration video showcasing the runtime learning system in the presence of noisy state samplings and random kicking is available at [Go2: Kick--Sensor (anonymous link)](https://www.dropbox.com/scl/fi/glqv1o9viltxvp083vmdt/sensorkick.mp4?rlkey=cj06mdbrdfnmcdqr688jjn38k&st=u0cs6312&dl=0).
>
>
>
> Our unknowns in the experiment with the real A1 quadruped robot go beyond simple disturbances, which include:
> 1. **Beta**: Disturbances injected into HP-Student's action/control commands.
> 2. **PD**: Random and sudden payload (around 4 lbs) drops on the robot's back.
> 3. **Kick**: Random and sudden kick by a human.
> 4. **DoS**: A real denial-of-service fault of the platform, which can be caused by task scheduling latency, communication delay, communication block, etc., is unknown to us.
> 5. **SP**:  A sudden side push.
>
> To be more convincing, we consider three combinations of these unknown unknowns applied to the runtime learning stage: i) **Beta + PD**, ii) **Beta + DoS + Kick**, and iii) **Beta + SP**. The demonstration video is available at [A1 Robot: unknown unknowns (anonymous link)](https://www.dropbox.com/scl/fi/jjxlym8xtdxqgr7d5hclp/unk.mp4?rlkey=kh76ojwhspqgslzolmowuuu0l&st=igbjnji3&dl=0). To clarify this critical information, we highlighted the five different unknowns in bold black font in the revised paper.
>
> In addition, Appendix I of the revised paper includes an experiment demonstrating safety assurance in the face of unknwn-unknown environments. Specifically, we initially pre-trained HP-Student for the A1 robot in the PyBullet simulator, using a flat terrain environment. After this pre-training, we directly deployed HP-Student to the Go2 robot. We utilized NVIDIA Isaac Sim to create an operating environment for showcasing the Go2 robot's runtime learning capabilities. This environment transitions from flat terrain to unstructured and uneven ground, further complicated by ice from unforseen freezing rain. We here can conclude that Go2 robot's operating environment are non-stationary and unforeseen, and never occur in the pre-training stage. The demonstration videos are available on [Go2 Forward (anonymous link)](https://www.dropbox.com/scl/fi/b1l7uskmx5wg2uw6n8tzz/forward.mp4?rlkey=6k4mfs1f8symff3ilquajsd3o&st=a17uvfo9&dl=0) and [Go2 Backward (anonymous link)](https://www.dropbox.com/scl/fi/fclb0npdyatq93qzq2i72/backward.mp4?rlkey=7u9bcmw3sz30zk6njqu79vtel&st=yje3jfwt&dl=0).
>
>
> -----
>
> **W2: Additionally, the theoretical justification for the teacher’s resilience against unknown unknowns could be strengthened, as the current analysis lacks formal rigor or quantitative measures.**
>
> **Answer:** We appreciate the reviewer’s insightful question. Our runtime learning machine has a rigorous and verifiable design. We detail them below.
>
> One of the primary motivations for this is the Sim2Real gap, as prevalent DRL typically involves training within a simulator. Additionally, there are challenges posed by unknown unknowns, the unpredictability of DNNs, and the difficulty in verifying their functions due to the vast parameter space and nonlinear activations. These make it nearly impossible to enable DRL with verified safety in safety-critical systems that operate in time-sensitive physical environments.

---

> ### Author Response · Authors · 2024-11-13
> **Response to Reviewer mpnQ: 2/5**
>
> These challenges inspire us to develop HA-Teacher, a verified yet simplified design focused exclusively on safety-critical functions. As a complement to DRL, HA-Teacher has two main objectives: i) to correct any unsafe learning conducted by HP-Student (the DRL agent), and ii) to back up the safety of the real plants. We conclude that the design of the HA-Teacher must be formal and rigorous regarding safety assurance. The actions of HA-Teacher must have verifiable safety, with rigorous theoretical foundations and proof. To achieve this, we propose three essential requirements (outlined in Requirements 1-3 of the paper) that the designed HA-Teacher must simultaneously fulfill. If these requirements are not met, we cannot trust HA-Teacher to correct unsafe learning or ensure safety effectively.  We will now summarize these three requirements.
>
> **Requirement 1: Conformity with Safety Regulations.** It requires that the HA-Teacher's real-time patches be subsets of the safety set. Otherwise, the HA-Teacher will not be able to ensure safety.
>
> **Requirement 2: Conformity with Operation Regulations.** It is necessary to confine the real-time actions of HA-Teacher within a physically-feasible bounded action space.
>
> **Requirement 3: Attracting Toward Safety Envelope.** The center of the HA-Teacher's patch must be located within the safety envelope. If it is not, the system's state may become trapped within the patch. This situation can cause the HA-Teacher to dominate the machine during runtime learning, preventing the HP-Student (the DRL agent) from independently learning a high-performance action policy.
>
> In the paper, Equations (16), (18), (23), and (28) to (31) detail the design of HA-Teacher, which is formally stated in Theorem 6.7. The resulting properties are outlined in Items 1-3 of Theorem 6.7. Meanwhile, Items 1-3 mean that Requirements 1-3 are satisfied. The theoretical proof and Equations (18), (23), and (28) to (31) are pretty lengthy, so we have moved them to Appendix D due to page limitation. This may have led to the reviewer’s impression that the design of HA-Teacher lacks formal rigor. To address this concern, we have highlighted sentences on pages 7 and 8 in teal color, clearly stating that the design of HA-Teacher simultaneously and strictly satisfies the proposed Requirements 1-3.
>
> -----
>
> **W3: While the selected experimental domains are relevant, further discussion on generalizability to applications like autonomous driving or complex industrial robots would broaden the framework’s impact.**
>
> **Answer:** We appreciate the reviewer’s suggestion. Our proposed runtime learning machine is explicitly designed for safety-critical systems. It applies to various safety-critical applications, such as quadruped robots, drones, UAVs, autonomous vehicles, humanoid robots, and cart-pole systems. The proposed learning machine holds strong generalizability because the needed additional knowledge for design is only the system dynamics model. The dynamics of safety-critical autonomous systems have been extensively and well studied in the mechanical area, allowing us to access the nonlinear dynamics models of many autonomous systems. For instance, when our application systems include quadruped robots, drones, and autonomous vehicles, we can obtain their dynamics models from relevant literature on system dynamics [1-3].
>
> Initially, we were able to provide more benchmarks to demonstrate the generalizability. Due to the page limit, the benchmarks in our original paper were reduced to a cart-pole system and a real A1 quadruped robot. However, partial experimental results still need to be moved to the appendices in the original paper. To more convincingly demonstrate the generalizability of our runtime learning machine, we added additional benchmarks to Appendix I in the revised paper. The two benchmarks are a 2D quadrotor and a Go2 quadruped robot.
>
> **References**
>
> [1] Di Carlo, J., Wensing, P. M., Katz, B., Bledt, G., \& Kim, S. (2018, October). Dynamic locomotion in the MIT Cheetah 3 through convex model-predictive control. In 2018 IEEE/RSJ international conference on intelligent robots and systems.
>
> [2] Beard, R. W. (2008). Quadrotor dynamics and control. Brigham Young University, 19(3), 46-56.
>
> [3] Rajamani, R. (2011). Vehicle dynamics and control. Springer Science \& Business Media.
>
>
>
> -----

---

> ### Author Response · Authors · 2024-11-13
> **Response to Reviewer mpnQ: 3/5**
>
> **W4: Although comparisons with other DRL methods are made, the analysis could be enhanced by discussing specific scenarios where other methods may outperform this framework or identifying potential limitations.**
>
> **Answer:** We appreciate the reviewer’s suggestion. While implementing the runtime learning machine on real robots, we encountered a few issues and limitations related to the software toolbox and hardware resources. Fortunately, we have successfully addressed these challenges.
>
> These issues and limitations do not occur in current DRL and fault-tolerant DRL frameworks because they do not address the real-time unknown unknowns and the real-time Sim2Real gap in the real plants. As a result, the real-time updating of the dynamics model and action policy, the real-time correction of unsafe learning, and runtime learning are not necessary. Below, we outline the issues and limitations we faced, along with our solutions. While this experience may have limited academic significance, it offers valuable implementation contributions that can save users considerable time and reduce hardware costs. Therefore, in the revised paper, we have included the identified issues, limitations, and solutions in **Appendix J: Implementation Problems and Solutions**.
>
> 1. **CVXPY Toolbox**:  The runtime learning machine depends on obtaining feasible solutions from Linear Matrix Inequalities (LMIs) to allow the HA-Teacher to perform safe actions. We found that the default CVX solver, SCS, exhibited instability and inconsistent accuracy across different computing platforms. Therefore, in our experiment, we used the CVXOPT solver [5] to achieve more stable and accurate results. While LMIs can generally be solved in real-time, the operating frequency on our platform still varies between 10 and 50 Hz. Although this fluctuation does not adversely affect overall performance in simulations, it does pose challenges when implementing the framework on a physical platform. To ensure efficient operation on hardware that requires a high frequency, such as a quadruped robot, we designed the HA-Teacher with an additional process that runs in parallel with the central control process. This design, however, adds complexity, such as the need for multi-process synchronization, and increases the demand for computational resources.
>
> 2. **Hardware Real-time Efficiency**: The runtime learning machine was tested by sending remote control commands to the quadruped robot indoors. All computations were executed on a desktop equipped with a 12th Gen Intel® Core™ i9-12900K 16-core processor. The onboard computing platform of the Unitree A1 utilizes an Intel® Atom™ x5-Z8350 CPU, which operates at 1.44 GHz with 6 cores. To enhance development efficiency, Python was chosen as the programming language for implementing this framework. Given that the runtime learning process requires significant interaction between the CPU and GPU, and considering the current architecture includes a high-frequency MPC module operating at 500 Hz for the quadruped robot, achieving comparable optimal real-time performance with the existing onboard hardware may be challenging. However, we believe that this computational limitation can be addressed or significantly alleviated by either adding extra computing resources—such as an external mini-PC, as suggested in [6]—or by restructuring the code framework. This restructuring could involve encapsulating the current implementation in C++ to enhance real-time performance, as proposed in [7].
>
> **References**
>
> [5] Andersen, M. S., Dahl, J., \& Vandenberghe, L. (2013). CVXOPT: A Python package for convex optimization. Available at cvxopt. org, 54.
>
> [6] Yang, Y., Zhang, T., Coumans, E., Tan, J., \& Boots, B. (2022, January). Fast and efficient locomotion via learned gait transitions. In Conference on robot learning (pp. 773-783).
>
> [7] Chen, Y., \& Nguyen, Q. (2024, May). Learning agile locomotion and adaptive behaviors via rl-augmented mpc. In 2024 IEEE International Conference on Robotics and Automation (pp. 11436-11442).
>
> -----
>
> **Q1: Could you provide more insights into the teacher's handling of complex unknowns, such as hardware failures or significant sensor errors?**
>
> **Answer:** By default, we assume that the hardware of our safety-critical systems, including mechanical components and sensors, functions correctly while some noise is permitted. Meanwhile, we assume the computing hardware and operating systems operate correctly,  which allows the ML software to work for arbitrary faults and bugs tolerated by HA-Teacher.

---

> ### Author Response · Authors · 2024-11-15
> **Response to Reviewer mpnQ: 4/5**
>
> A single HA-Teacher cannot ensure safety if the assumptions are violated, i.e.,  significant hardware or software faults occur. For example, our quadruped robot is equipped with 12 joint motors. If even one of these motors becomes frozen or damaged, the robot could fall, potentially causing bodily harm.  In our robot experiment, a damping mode is activated to perform a soft emergency stop when HA-Teacher encounters significant hardware or software faults.
>
> In summary, safety-critical systems shall include a damping mode that responds to substantial incidents that exceed the HA Teacher's capabilities. An illustrative example is the emergency landing option available to airplane pilots; it is a safeguard when a situation surpasses their handling ability.
>
> -----
>
> **Q2: What theoretical guarantees exist for the teacher's performance in completely unanticipated scenarios?**
>
> **Answer:** To answer the question, we would like first to recall the ground-truth dynamics of tracking errors of real plants below.
>
> $\mathbf{e}(k+1) = \mathbf{A}(\mathbf{s}(k)) \cdot \mathbf{e}(k) + \mathbf{B}(\mathbf{s}(k)) \cdot \mathbf{a}_{\text{HA}}(k) + \mathbf{h}(\mathbf{e}(k)),~~~~~~~~~~(1)$
> where  $\mathbf{e}(k)$ denotes the real-time tracking error, while $(\mathbf{A}(\mathbf{s}(k)), \mathbf{B}(\mathbf{s}(k)))$ represents the real-time dynamics model knowledge employed by HA-Teacher. Additionally, $\mathbf{h}(\mathbf{e}(k))$ signifies the model mismatch. Based on this model, we then introduce a practical and commonly held assumption regarding the model mismatch for HA-Teacher design.
>
> **Assumption:** The model mismatch $\mathbf{h}(\cdot)$ is locally Lipschitz in the patch $\Psi_k$, i.e.,
>
> $(\mathbf{h}(\bf{e}_1) - \bf{h}(e_2))^T \cdot \widehat{\mathbf{P}}_k \cdot (\mathbf{h}(\bf{e}_1) - \bf{h}(e_2)) \le \kappa  \cdot (\bf{e}_1 - e_2)^T \cdot \widehat{\mathbf{P}}_k \cdot (\bf{e}_1 - e_2)$ holds for any $\bf{e}_1,  \bf{e}_2 \in \Psi_k.~~~~~~~~~(2)$
>
> Based on this assumption, we have developed the design for HA-Teacher, which is formally stated in Theorem 6.7 of the paper. In Appendix D, we provide mathematical proof demonstrating that HA-Teacher fulfills three essential requirements: 1) compliance with safety regulations, 2) adherence to operational regulations, and 3) alignment with the safety envelope of HP-Student. This ensures that the actions delivered are verifiably safe.
>
> We note that the dynamics of many safety-critical autonomous systems are governed by a combination of known knowns (e.g., Newton's laws of motion), known unknowns (e.g., Gaussian noise without knowing the mean and variance), and unknown unknowns. The unknown unknowns include the hard-to-predict hallucinations of DRL,  unanticipated/unforeseen scenarios such as freezing rain, etc.
>
> The combined influences on system dynamics can be expressed through real-time sampling of the system state, denoted as $\mathbf{s}(k)$. The HA-Teacher’s dynamics knowledge, represented by $(\mathbf{A}(\mathbf{s}(k)), \mathbf{B}(\mathbf{s}(k)))$, is dependent on the real-time state samplings. As a result, the HA-Teacher's action policy considers certain unexpected scenarios in real-time. However, there are still unanticipated influences captured by the mismatch term $\mathbf{h}(\mathbf{e}(k))$. It is important to note that the design of the HA-Teacher and the accompanying mathematical proof are based on the assumption of model mismatch (i.e., locally Lipschitz in (2)). So, if the unanticipated scenarios do not result in violating condition (2), then the HA-Teacher's real-time action policy in these scenarios is considered verified and safe. However, if these scenarios do lead to a violation of condition (2), the HA-Teacher's real-time action policy may still be safe, but it cannot be verified based on Theorem 6.7. This is because the conditions outlined in Theorem 6.7 are sufficient but not necessary and sufficient.
>
> In practical operation, if the safety of the real plant under the control of our runtime learning machine still cannot be guaranteed, the system's real-time behavior of approaching the safety boundary will trigger a damping mode for operation termination (e.g., the soft emergency stop in our robot experiments).
>
> -----
>
> **Q3: How might this system perform in applications outside of the tested environments, such as in higher dimensional or more complex environments?**
>
> **Answer:** Our **Answer** to **Q2** can also partially address **Q3**. This is because safety-critical autonomous systems usually have environment-dynamics interactions. If the real-time operating environments do not result in a violation of condition (2) in Answer to Q2, HA-Teacher can offer a verified safe action policy for backing up the safety of real plants and correcting unsafe learning of HP-Student. This also means HA-Teacher is adaptive to the operating environments.

---

> ### Author Response · Authors · 2024-11-15
> **Response to Reviewer mpnQ: 5/5**
>
> To convincingly demonstrate the capability of our runtime learning machine to ensure safety in more complex environments. We designed a challenging experiment. Specifically, we first pre-trained Phy-DRL well for the A1 robot in the PyBullet simulator, and its environment is a plat terrain with high friction. We then directly employed the well-pre-trained Phy-DRL for the A1 robot as the HP-Student in the runtime learning machine for the Go2 robot. We use the NVIDIA Isaac Sim to build an operating environment for the Go2 robot. The operating environment transitions from flat to uneven and uninstructed terrain, covered with ice caused by unforeseen freezing rain. We conclude that the Go2 robot's operating environment differs significantly from the A1 robot's training and testing environments. Besides, as shown in the figure of AI and Go robots available at [A1Go2 (anonymous link)](https://www.dropbox.com/scl/fi/uqz37dibqfv4z4kosfth6/A1Go2.pdf?rlkey=mqhmg29zo9ht8z1bmsui66rzj&st=liwphjtb&dl=0), A1 and Go2 robots are very different in their motors, weights, heights, etc. The demonstration videos are available on [Go2 Forward (anonymous link)](https://www.dropbox.com/scl/fi/b1l7uskmx5wg2uw6n8tzz/forward.mp4?rlkey=6k4mfs1f8symff3ilquajsd3o&st=a17uvfo9&dl=0) and [Go2 Backward (anonymous link)](https://www.dropbox.com/scl/fi/fclb0npdyatq93qzq2i72/backward.mp4?rlkey=7u9bcmw3sz30zk6njqu79vtel&st=yje3jfwt&dl=0).
>
> -----
>
> **-END-**

---

> ### Author Response · Authors · 2024-11-22
> **Discussion deadline approaching**
>
> Dear Reviewer mpnQ,
>
>
> Thank you for your thoughtful reviews. We have addressed all of your comments in detail and updated the paper as necessary.
>
> The discussion period will conclude in a few days. Could you please review our responses and the updated paper? We would appreciate any follow-up questions you may have, and we are happy to provide timely answers.
>
>
> Kind regards,
>
> the Authors

---

> > ### Comment · Area_Chair_pxBR · 2024-11-24
> > **Please respond to the rebuttal**
> >
> > Dear reviewer,
> > The process only works if we engage in discussion. Can you please respond to the rebuttal provided by the authors ASAP?

---

> ### Author Response · Authors · 2024-11-25
> **The deadline for paper revisions is in 36 hours!**
>
> Dear Reviewer mpnQ,
>
> We hope this message finds you well. The author-reviewer discussion phase has been extended, but only for discussions on minor issues. The deadline for revising the paper is within 36 hours. This means we cannot update the paper after 36 hours. Your timely feedback is thus crucial to us now.
>
> Could you please review our responses and the updated paper? We are always eager to hear your valuable feedback and provide further responses.
>
>
> Warm regards,
>
> Authors

---

> > ### Comment · Reviewer_mpnQ · 2024-11-26
> >
> > Thank you for your response. Some of my questions have been addressed and the paper has been improved. My score will remain unchanged based on my overall evaluation of the contribution of the proposed method.

---

> ### Author Response · Authors · 2024-11-26
> **Thank you! Your remianing questions.**
>
> Dear Reviewer mpnQ,
>
> We are pleased that our previous response has addressed some of your questions. As you know, the Author-Reviewer discussion has been extended (but after Nov. 26 AoE, paper revision is not permitted). If you could share your any remaining questions or concerns, we would be happy to address them.
>
> Warm regards,
>
> Authors

---

### Author Response · Authors · 2024-11-18
**Global Response: Part II**

$\textbf{II. Automatic Triggering Frequency of HA-Teacher (Reviewer Z8i5)}$. The original paper impressed the **Reviewer Z8i5** that the paper lacks the studies of HA-Teacher's triggering frequency. The paper has such studies, which are partially presented in the appendix. We explain them in detail below.

In our runtime learning machine, when the real-time safety status of the real plant being controlled by HP-Student approaches the safety boundary, the Coordinator **automatically** prompts the HA-Teacher to intervene, ensure the safety of the real plant, and correct HP-Student's unsafe learning. When the real-time states return to a safe region, the Coordinator **automatically** triggers the switch back to HP-Student and terminates the learning correction. We can further conclude that the frequency of triggering HA-Teacher depends on HP-Student's capability of assuring safety in the face of the Sim2Real gap and unknown unknowns. This can be viewed from the phase plot available at [HA-Teacher v.s. HP-Student (anonymous link)](https://www.dropbox.com/scl/fi/j70n27wtyufmjgrzi4tn8/2DHAHP.pdf?rlkey=si9ifb43ac6d6z3s8whyjq45g&st=bdjmfqgu&dl=0) (also, Figure 21 in the paper). Specifically, the HA-Teacher was frequently activated because the HP-Student had not yet learned how to ensure safety. However, as the HP-Student spent more time learning, his ability to maintain safety improved. This progress is particularly evident in episodes 1 to 5, during which the frequency of HA-Teacher activation decreased. By episode 6, the HP-Student had mastered the capability to ensure safety, and thereafter, his continued runtime learning no longer required assistance from the HA-Teacher. This advancement enabled him to autonomously develop a high-performance action policy, resulting in an end state close to the goal (i.e., the center of the ellipse safety envelope). Our paper includes more such studies with evaluation metrics; see Appendix G 5.3 of the revised paper.

-----

$\textbf{III. Tolerating Much More Complex Unknown Unknowns (Reviewer xKGE)}$. The dynamics of real safety-critical autonomous systems are very complex so that they can face very different types of unknown unknowns.

In the original paper, the unknowns in the experiment of cart-pole systems include the unknown-unknown friction forces and noisy samplings of action/control commands.  In the experiment with the real A1 quadruped robot, the unknowns include 1) **Beta**: Disturbances injected into HP-Student's action/control commands, 2) **PD**: Random and sudden payload (around 4 lbs) drops on the robot's back, 3) **Kick**: Random and sudden kick by a human, 4) **DoS**: A real denial-of-service fault of the platform, which can be caused by task scheduling latency, communication delay, communication block, etc., is unknown to us, 5) **SP**:  A sudden side push. In our implementation, we consider three complex combinations of them: i) **Beta + PD**, ii) **Beta + DoS + Kick**, and iii) **Beta + SP**. The demonstration video is available at [A1 Robot: unknown unknowns (anonymous link)](https://www.dropbox.com/scl/fi/jjxlym8xtdxqgr7d5hclp/unk.mp4?rlkey=kh76ojwhspqgslzolmowuuu0l&st=igbjnji3&dl=0).

Many safety-critical autonomous systems, especially those operating outdoors, have dynamics-environment interactions. So, the operating environment could also induce unknown unknowns. According to comments from Reviewer xKGE, the experiment of the new benchmark Go2 robot includes such demonstrations about environments, whose demo videos are available on [Go2 Forward (anonymous link)](https://www.dropbox.com/scl/fi/b1l7uskmx5wg2uw6n8tzz/forward.mp4?rlkey=6k4mfs1f8symff3ilquajsd3o&st=a17uvfo9&dl=0),  [Go2 Backward (anonymous link)](https://www.dropbox.com/scl/fi/fclb0npdyatq93qzq2i72/backward.mp4?rlkey=7u9bcmw3sz30zk6njqu79vtel&st=yje3jfwt&dl=0), and [Go2: Kick--Sensor (anonymous link)](https://www.dropbox.com/scl/fi/glqv1o9viltxvp083vmdt/sensorkick.mp4?rlkey=cj06mdbrdfnmcdqr688jjn38k&st=u0cs6312&dl=0). The videos demonstratee our runtime learning machine's significantly enhanced safety assurance in **non-stationary, unstructured, uneven, and unforeseen operating environments (low friction due to unexpected freezing rain)**, along with very different mission commands/goals and unknown-unknown noisy samplings of state sensors.

-----

$\textbf{IV. Two New Benchmarks}$. Initially, we were able to provide more benchmarks; however, we need to demonstrate that we have achieved the proposed prospect along with all four claimed features, which requires more pages. Due to the page limit, the benchmarks in our original paper were reduced to a cart-pole system and the real A1 quadruped robot. Having only two benchmarks leads to questions about generalizability, according to **Reviewers Z8i5 and mpnQ**. We added two new benchmarks to the revised paper to address the concern. They are the 2D quadrotor and the Go2 quadruped robot.

----


**-END-**

---

### Author Response · Authors · 2024-11-21
**Global Response: Part I**

We appreciate our reviewers' supportive feedback, thorough evaluations, and insightful remarks. The paper has been updated as needed, and the updated Supplementary Material includes all the benchmark codes and **six demonstration videos**. Although our response to reviewers is lengthy, it addresses all the reviewers' comments in detail. We'd like to summarize the four significant clarifications in response to the reviewers' critical comments as a global response.

------

$\textbf{I. Lifetime Safety, not Step-wise Safety (Reviewer Z8i5)}$. In the post-rebuttal, **Reviewer Z8i5** argues that our theoretical safety guarantee or definition is step-wise, implying it only ensures safety for one step. **Definition 2.1** (page 4) and **Theorem 6.4** (page 7) of the paper clearly and formally state this is untrue! To demonstrate this, we recall them from the paper.

**Definition 2.1 (Lifetime Safety).** Consider the safety set $\mathbb{X}$. The real plant is said to have lifetime safety, if given any $\mathbf{s}(1) \in \mathbb{X}$, the $\mathbf{s}(k) \in  \mathbb{X}$ holds at any time $k \in \mathbb{N}$, regardless of HP-Student's failure.

**Statement 1 in Theorem 6.4.** The real-time patch ${\Psi_{\sigma(k)}} \subseteq {\mathbb{X}}$ holds for any time $k$.

Definition 2.1 specifies one of our design aims: rendering the given safety set $\mathbb{X}$ invariant by our learning machine, that is, under the control of our runtime learning machine, operating from the safety set, system states will never leave the safety set at any current or future time. This design aim is mathematically described by 'given any $\mathbf{s}(1) \in \mathbb{X}$, the $\mathbf{s}(k) \in  \mathbb{X}$ holds at any time $k \in \mathbb{N}$' in Definition 2.1. Lifetime safety is achieved, as indicated by Statement 1 in Theorem 6.4. We note that 'holds at any time $k \in \mathbb{N}$' in Definition 2.1 and Statement 1 in Theorem 6.4 explicitly state that our safety definition is lifetime, not step-wise.

During real operation, we must constantly monitor the real-time system states to decide if to trigger HA-Teacher for backup safety. The triggering condition is $\mathbf{s}^\top(k-1) \cdot \mathbf{P} \cdot \mathbf{s}(k-1) \leq 1$ and $\mathbf{s}^\top(k) \cdot \mathbf{P} \cdot \mathbf{s}(k) > 1$. The condition is only used to trigger the switching.  If we only consider safety assurance, i.e., not caring about mission performance, we can always use HA-Teacher. The triggering condition can be removed because its design ensures safety for a **lifetime**, **not step-wise**, as indicated by Theorem 6.4 of the revised paper.

In conclusion, our safety definition and design are **lifetime** and **deterministic**, which has been a standard design aim for many other safety- and time-critical autonomous systems, such as stated by Definition 2.1 in [R1], Definitions 2.3 and 2.4 in [R2], Section 2 in [R3], Definition 1 in [R4].

Importantly, lifetime safety is our claimed notable feature of runtime learning machine, which has been well demonstrated by experimental results in **Figures 3 (a) and (b)**, **Figures 9 to 12**, and **Figure 21** (or the plot available at [HA-Teacher v.s. HP-Student (anonymous link)](https://www.dropbox.com/scl/fi/j70n27wtyufmjgrzi4tn8/2DHAHP.pdf?rlkey=si9ifb43ac6d6z3s8whyjq45g&st=bdjmfqgu&dl=0)) of the updated paper.



**Lastly, we believe the reviewer’s concern can be further addressed by the (safety) demonstration videos available at [A1 Robot: unknown unknowns (anonymous link)](https://www.dropbox.com/scl/fi/jjxlym8xtdxqgr7d5hclp/unk.mp4?rlkey=kh76ojwhspqgslzolmowuuu0l&st=igbjnji3&dl=0), [A1 Robot: Sim2Real gap (anonymous link)](https://www.dropbox.com/scl/fi/mjwi9u6ng72oghbm4wkx7/sim2real.mp4?rlkey=2lg4fayh2mrpj2m6nqvx02tox&st=ono7z5na&dl=0), [Go2 Forward (anonymous link)](https://www.dropbox.com/scl/fi/b1l7uskmx5wg2uw6n8tzz/forward.mp4?rlkey=6k4mfs1f8symff3ilquajsd3o&st=a17uvfo9&dl=0),  [Go2 Backward (anonymous link)](https://www.dropbox.com/scl/fi/fclb0npdyatq93qzq2i72/backward.mp4?rlkey=7u9bcmw3sz30zk6njqu79vtel&st=yje3jfwt&dl=0), and [Go2: Kick--Sensor (anonymous link)](https://www.dropbox.com/scl/fi/glqv1o9viltxvp083vmdt/sensorkick.mp4?rlkey=cj06mdbrdfnmcdqr688jjn38k&st=u0cs6312&dl=0).**

**References**

[R1] Cao et al. Physics-Regulated Deep Reinforcement Learning: Invariant Embeddings. In The Twelfth International Conference on Learning Representations, 2024.

[R2] Seto, D., \& Sha, L. An engineering method for safety region development. Carnegie Mellon University, Software Engineering Institute.

[R3] Bak et al. Real-time reachability for verified simplex design. In 2014 IEEE Real-Time Systems Symposium. IEEE.

[R4] Tabas, D., \& Zhang, B. Computationally efficient safe reinforcement learning for power systems. In 2022 American Control Conference.

-----

---

### Meta-Review · Area_Chair_pxBR · 2024-12-20

**Metareview:**

The paper presents a runtime monitoring and correction system for safe deep RL that remains performant. The primary methodology introduces a student whose goal is to maximize performance, a teacher whose goal is to ensure safety and a coordinator. These act together to ensure system safety by projecting back into a safety envelope

Strength:
Important premise for any deployed system to be safe and performant
Seems like a principled and well motivated approach
Results on a real robot system

Weaknesses
The paper is written in a way that is deeply inaccessible to most DRL researchers. In it's current form, the paper will be completely unreadable by it's target audience
Secondarily, there are lots of unusual terms, acronyms and arcane terminologies used throughout the paper that make it hard to parse.
Connections to other constrained deep RL methods should also be made more clear

Overall this paper is quite borderline,  I will rate it towards a reject because despite an interesting premise, it's got too much going on to actually communicate to the target audience. The paper would be much stronger with a simplification of the writing and a focus of the system so that it is far more accessible to the broad DRL and safety audience.

**Additional Comments On Reviewer Discussion:**

The reviewers raised questions about applicability to broader domains, guarantees, presentation and requested additional benchmarks/metrics. The authors provided some of these in the rebuttal, but there are several clarity issues that remain.

---

### Decision · Program_Chairs · 2025-01-22

Reject